# Parallel processing of working memory and temporal information by distinct types of cortical projection neurons

Jung Won Bae [1,5], Huijeong Jeong [1,2,5], Young Ju Yoon[1], Chan Mee Bae [1,2], Hyeonsu Lee [3], Se-Bum Paik [3,4] & Min Whan Jung [1,2✉]

It is unclear how different types of cortical projection neurons work together to support diverse cortical functions. We examined the discharge characteristics and inactivation effects of intratelencephalic (IT) and pyramidal tract (PT) neurons—two major types of cortical excitatory neurons that project to cortical and subcortical structures, respectively—in the deep layer of the medial prefrontal cortex in mice performing a delayed response task. We found stronger target-dependent firing of IT than PT neurons during the delay period. We also found the inactivation of IT neurons, but not PT neurons, impairs behavioral performance. In contrast, PT neurons carry more temporal information than IT neurons during the delay period. Our results indicate a division of labor between IT and PT projection neurons in the prefrontal cortex for the maintenance of working memory and for tracking the passage of time, respectively.

[1] Department of Biological Sciences, Korea Advanced Institute of Science and Technology, Daejeon, Korea. [2] Center for Synaptic Brain Dysfunctions, Institute for Basic Science, Daejeon, Korea. [3] Department of Bio and Brain Engineering, Korea Advanced Institute of Science and Technology, Daejeon, Korea. [4] Program of Brain and Cognitive Engineering, Korea Advanced Institute of Science and Technology, Daejeon, Korea. [5] These authors contributed equally: Jung Won Bae, Huijeong Jeong. ✉email: mwjung@kaist.ac.kr

The cerebral cortex is populated by multiple types of excitatory and inhibitory neurons wired together in complex circuits. Cortical projection neurons can be grouped into intratelencephalic (IT), pyramidal tract (PT), and corticothalamic (CT) projection neurons, with each group differing from one another in developmental origin, morphology, laminar distribution, and input/output connectivity[1–5]. IT neurons are distributed across all layers of the cortex. They project to other cortical areas as well as bilaterally to the striatum. PT neurons are restricted to layer 5, projecting to ipsilateral subcortical structures. CT neurons are located in layer 6 and they project primarily to the thalamus. Clarifying how the different types of cortical projection neurons contribute to the diverse functions of the cortex is essential for understanding the neural circuit mechanisms that underlie those cortical functions.

Few studies have compared the functions of different types of cortical projection neurons in mice. In the primary motor cortex, PT neurons specifically encode the direction and speed of a subsequent behavior during sensory-guided navigation. In contrast, layer 2/3 IT neurons are activated by unexpected direction changes induced by visual perturbations independent of behavioral response[6]. Similarly, PT neurons in the anterior lateral motor cortex—the secondary motor cortex—selectively encode and drive contralateral action selection. IT neurons, in contrast, show mixed directional selectivity with little contralateral bias[7]. These findings suggest that, in motor cortical areas, PT neurons are more involved in the control of specific movements than IT neurons. In the primary visual cortex, PT neurons are more direction-selective and prefer faster stimuli, but exhibit broader tuning for orientation and spatial frequency compared to IT neurons[8–10]. In the primary somatosensory cortex, activity of PT neurons, but not IT neurons, is correlated with perception of whisker movement[11]. In the medial prefrontal cortex (mPFC), optogenetic silencing of CT neurons, but not corticocortical IT neurons, impairs choice flexibility[12]. These studies demonstrate functional differentiation among IT, PT, and CT neurons in different cortical regions. Not only are there few published studies on this subject, most of those that have been published targeted the sensory/motor cortices. Thus, our understanding of the ways different types of cortical projection neurons contribute to the great variety of cortical functions, especially high-order cognitive functions, remains limited.

In the present study, we investigated the roles mPFC IT and PT neurons play in working memory, targeting deep layers where different types of projection neurons are intermingled. Working memory refers to the limited-capacity memory system by which the brain temporarily holds and manipulates information[13]. Although working memory is essential for an array of complex cognitive tasks, the cortical circuit processes that support working memory remain unclear. We are particularly interested in whether and how the different types of cortical projection neurons contribute to the maintenance of task-relevant information in the absence of external sensory cues. To investigate this, we examined the discharge characteristics and inactivation effects of IT and PT neurons in the mPFC of mice performing a delayed response task. The results indicate a major contribution of IT neurons to the active maintenance of target information whereas PT neurons contribute more to keeping track of the elapse of delay.

## Results

We trained head-fixed mice in a delayed match-to-sample task (Fig. 1a, b). The mice were rewarded with water when, after a delay period, they chose the same water lick port that was presented during the sample phase (target) before the delay period (Supplementary Video 1). The mice were trained with fixed (4, 15, or 25 s) or variable (1–7 s; uniform random distribution) durations of delay. We previously compared the discharge characteristics of mPFC neurons during the fixed- and variable-delay conditions using a subset of the data[14]. In this study, we focus on comparing discharge characteristics and inactivation effects of deep-layer IT and PT neurons during the fixed-delay condition.

**mPFC inactivation impairs behavioral performance**. All mice were initially trained with a fixed delay (4 s) until they reached the performance criterion (>70% correct target choices for three consecutive daily sessions). We used wild-type (WT) mice to examine the effect of inactivating the mPFC on behavioral performance. The WT mice ($n = 6$), that were well trained to perform the task at 4-s fixed delay, showed significantly impaired behavioral performance upon bilateral infusion of muscimol into the mPFC (Fig. 1c–f; histological identification of cannula locations in Supplementary Fig. 1) with no effect on their lick rate as compared to no-infusion and artificial cerebrospinal fluid (ACSF)-infusion controls (% correct, one-way repeated measures ANOVA, $F_{(2,10)} = 107.354$, $p = 1.7 \times 10^{-7}$; Bonferroni's post hoc test, muscimol versus ACSF, $p = 5.3 \times 10^{-5}$, muscimol versus no-infusion, $p = 4.3 \times 10^{-4}$, ACSF versus no-infusion, $p = 0.485$; % error (wrong target choice), one-way repeated measures ANOVA, $F_{(2,10)} = 23.266$, $p = 1.7 \times 10^{-4}$; Bonferroni's post hoc test, muscimol versus ACSF, $p = 0.013$, muscimol versus no-infusion, $p = 0.012$, ACSF versus no-infusion, $p = 0.609$; % miss (no choice within 4 s since delay offset), Friedman test, $\chi^2(2) = 10.333$, $p = 0.109$; lick rate, one-way repeated measures ANOVA, $F_{(2,10)} = 0.474$, $p = 0.636$; Fig. 1e, f). The WT mice were further trained to perform two consecutive blocks of fixed-delay (4 s; 60–80 trials per block) and variable-delay (1–7 s; 60–80 trials per block) trials in a daily session (their order reversed across successive sessions) and, after reaching the performance criterion (>70% correct target choices under both delay conditions for three consecutive daily sessions), tested with drug infusion. They showed significantly impaired behavioral performance upon muscimol infusion under both fixed- and variable-delay conditions (Supplementary Fig. 2). These results clearly implicate the mPFC in this delayed match-to-sample task.

**Identification and classification of IT and PT neurons**. We then examined discharge characteristics of deep-layer IT and PT neurons during the delayed match-to-sample task. For this, we expressed channelrhodopsin-2 (ChR2), that was fused with enhanced yellow fluorescent protein (eYFP), in the mPFC of Rxfp3-Cre ($n = 14$) and Efr3a-Cre ($n = 12$) knock-in mice to target deep-layer IT and PT neurons, respectively[15] (Fig. 2a; see Supplementary Fig. 3 for histological verification of IT and PT neurons using separate groups of mice). Fluorescence expression was localized largely in deep layers of the mPFC in both Rxfp3-Cre and Efr3a-Cre mice (Fig. 2a). However, the distribution of fluorescence-expressing cells differed significantly between Rxfp3-Cre and Efr3a-Cre mice (Kolmogorov–Smirnov test, $p = 0.023$; Fig. 2b). Fluorescence-expressing cells were found most densely in superficial layer 5 (layer 5a) in Rxfp3-Cre ($n = 3$) mice, but in deep layer 5 (layer 5b) in Efr3a-Cre ($n = 3$) mice (two-way mixed ANOVA, main effect of Cre-line, $F_{(1,4)} = -2.7 \times 10^{-14}$, $p = 1$, main effect of layer, $F_{(4,16)} = 28.633$, $p = 4.1 \times 10^{-7}$, Cre-line × layer interaction, $F_{(4,16)} = 15$, $p = 2.8 \times 10^{-5}$; Bonferroni's post hoc test for Cre-line, layer 5a, $p = 0.001$; layer 5b, $p = 0.002$; layer 6, $p = 0.041$; Fig. 2b), which is consistent with known laminar distributions of IT and PT neurons[15,16].

We implanted in the mPFC an optic fiber for optogenetic tagging and eight movable tetrodes for recording neuronal activity.

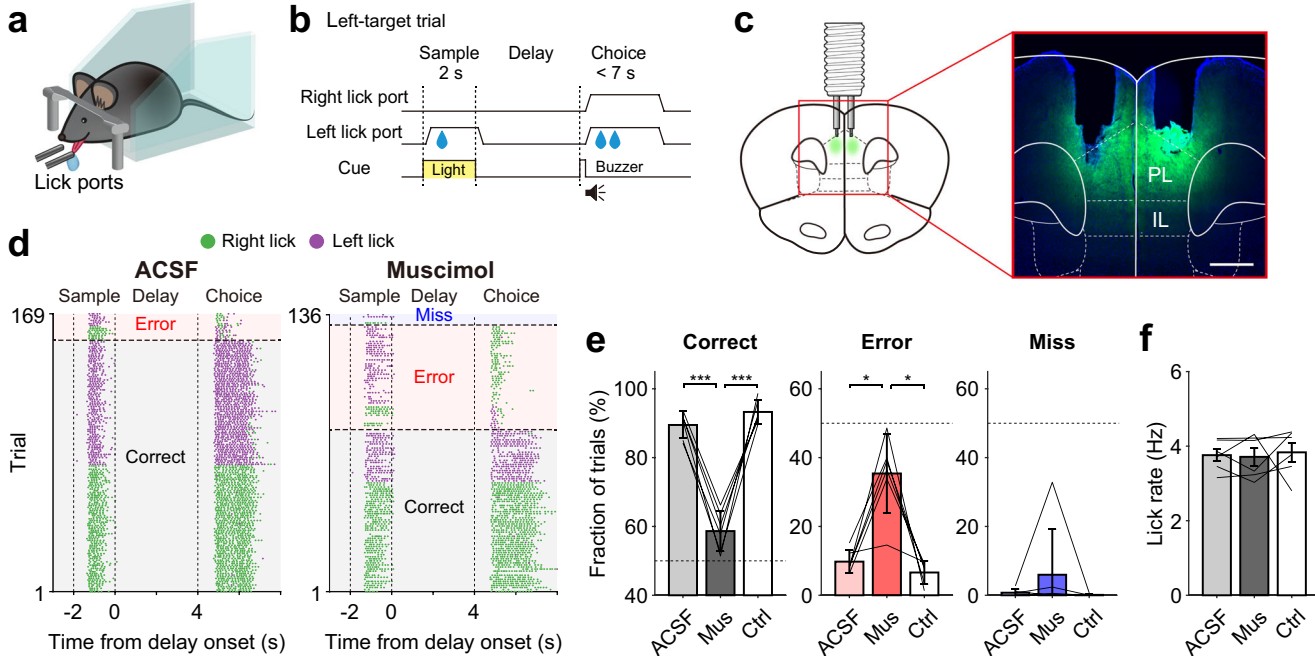

**Fig. 1 mPFC inactivation impairs behavioral performance. a** The experimental setting (custom-created image). Head-fixed mice performed a delayed match-to-sample task. **b** Schematic for a left-target trial. Mice obtained a water reward by choosing the lick port presented during the sample phase (target) before a delay period. **c** Schematic (left) and a coronal brain section (representative of six mice; right) showing cannula tracks and the spread of fluorescein (green) in the mPFC. PL prelimbic cortex, IL infralimbic cortex. Scale bar, 500 μm. **d–f** Effects of mPFC inactivation on behavior under the 4-s fixed-delay condition. **d** Licking responses during example sessions with bilateral ACSF (left) or muscimol (right) infusions into the mPFC. Each line is one trial and each dot represents a lick (green, right; purple, left). Error, wrong target choice; miss, no choice. Trials are grouped according to trial type (correct, error, and miss) and target (left versus right lick port). **e, f** Mean (±SEM across six WT mice) fractions of correct (left), error (middle), and miss (right) trials (**e**) as well as lick rates during the delay period (**f**) following ACSF, muscimol (Mus), and no-drug (Ctrl) infusions. Thin lines, individual animal data. *$p < 0.05$, ***$p < 0.001$ (one-way repeated measures ANOVA followed by Bonferroni's post hoc tests).

We recorded 860 and 713 neurons from the mPFC (prelimbic and infralimbic cortices) of 14 Rxfp3-Cre and 12 Efr3a-Cre mice, respectively, during task performance (histological identification of tetrode and optic fiber locations in Supplementary Fig. 1). Then, we classified these neurons into two groups based on their firing rates and spike waveforms. On average, neurons in one group ($n = 164$) fired at higher rates with narrower and more asymmetric spike waveforms than neurons in the other group ($n = 1374$; Fig. 2c). They will be referred to as narrow-spiking (NS; putative inhibitory) and wide-spiking (WS; putative pyramidal) neurons, respectively. We optically tagged 42 WS, 19 NS, and five unclassified neurons in Rxfp3-Cre mice, as well as 47 WS and 15 NS neurons in Efr3a-Cre mice (Fig. 2d, e; light responses of all optically tagged WS and NS neurons are shown in Supplementary Fig. 4). Measures of optical tagging accuracy were similar between tagged WS and NS neurons (SALT test $p$ value, two-way ANOVA, main effect of cell type, $F_{(1,119)} = 0.020$, $p = 0.887$; main effect of Cre-line, $F_{(1,119)} = 3.406$, $p = 0.067$; cell type × Cre-line interaction, $F_{(1,119)} = 0.020$, $p = 0.889$) or significantly higher in NS than WS neurons (spike waveform correlation, main effect of cell type, $F_{(1,119)} = 15.796$, $p = 1.2×10^{-4}$; main effect of Cre-line, $F_{(1,119)} = 0.494$, $p = 0.483$; cell type × Cre-line interaction, $F_{(1,119)} = 0.409$, $p = 0.524$). Also, peak response latency following optical stimulation did not differ significantly between optically tagged WS and NS neurons (main effect of cell type, $F_{(1,119)} = 3.048$, $p = 0.083$; main effect of Cre-line, $F_{(1,119)} = 1.780$, $p = 0.185$; cell type × Cre-line interaction, $F_{(1,119)} = 0.002$, $p = 0.966$; Fig. 2e). These results argue against indirect activation of NS neurons by optical stimulation, supporting the existence of NS neurons that express an IT or PT neuronal marker. To explore the possibility that the optically tagged NS neurons are parvalbumin

(PV)-expressing inhibitory interneurons[17], we examined the overlap between Cre and PV expressions using separate groups of Rxfp3-Cre ($n = 5$) and Efr3a-Cre ($n = 5$) mice that were crossed with Ai9 tdTomato reporter line (JAX 007909, Jackson Laboratory) mice (see Methods). None of the Cre-expressing neurons expressed PV (Fig. 2f, g), indicating that they are different from PV-expressing inhibitory interneurons.

**IT neurons carry stronger target signals than PT neurons during delay period.** The tetrode-implanted Rxfp3-Cre and Efr3a-Cre mice performed two consecutive blocks of the delayed match-to-sample task, one with fixed (4 s; 60–80 trials) and the other with variable- (1–7 s; 60–80 trials) delay durations (see Supplementary Fig. 2 for behavioral performance). Here, we show activity of IT and PT neurons during the fixed-delay condition (see Supplementary Fig. 5 for comparison of delay-period activity across the fixed- and variable-delay conditions). Throughout the study, of all the units recorded with tetrodes, we analyzed those units with mean firing rates during task performance ≥0.5 Hz and recorded in the sessions with ≥10 correct trials for both targets (26 WS IT, 31 WS PT, 19 NS IT, 14 NS PT, and 758 untagged WS neurons; numbers of neurons included in each analysis are summarized in Supplementary Table 1). Also, to be conservative, we included only optically tagged WS neurons in the main analyses (see below for the results obtained from optically tagged NS neurons). Their mean discharge rates during task performance were 2.9 ± 2.1 (mean ± SD; WS IT) and 2.8 ± 2.7 Hz (WS PT). We used only correct trials in the analysis of target-dependent firing of IT and PT neurons because relatively small numbers of IT and PT neurons, along with small numbers of error trials (left target,

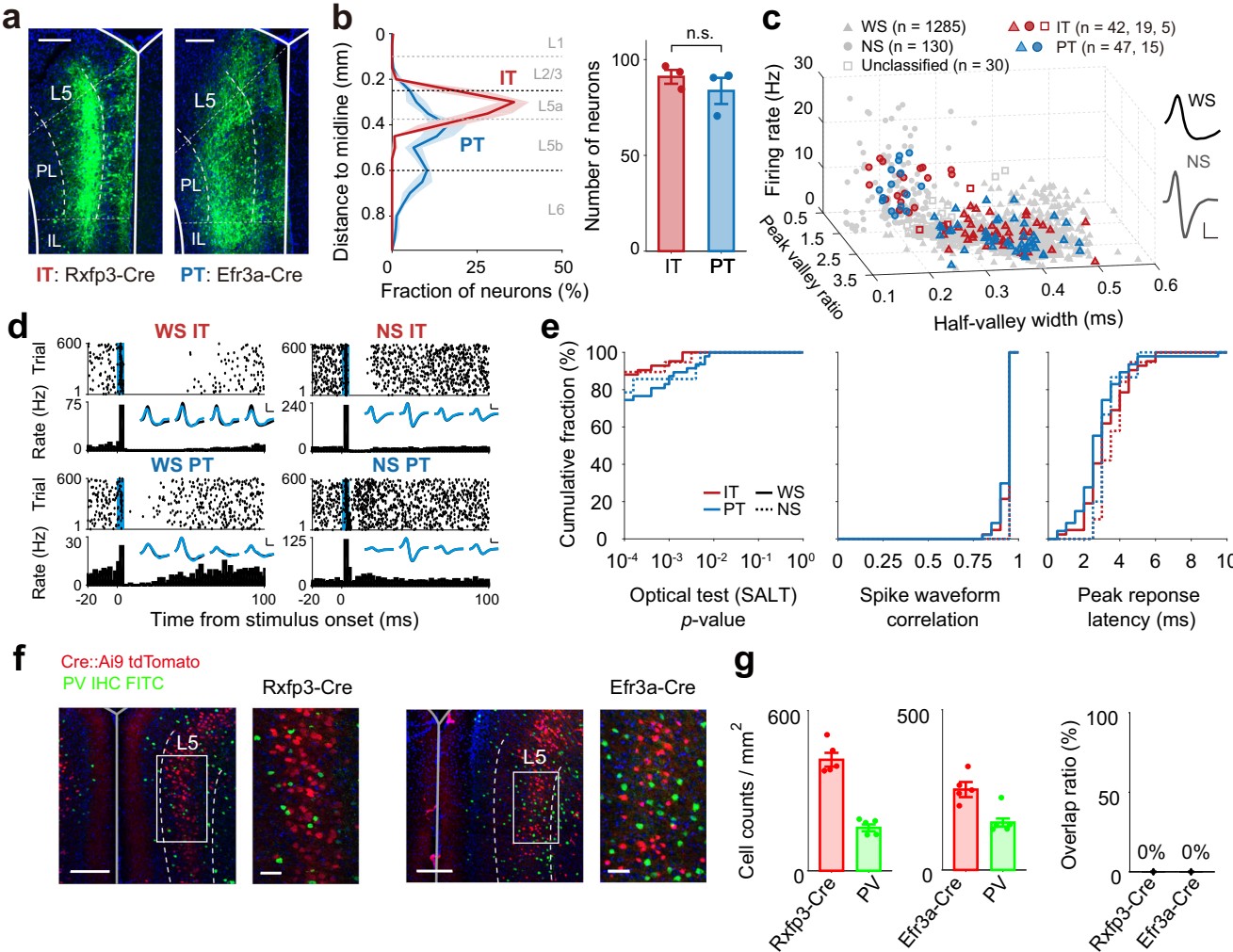

**Fig. 2 Optical tagging and unit classification. a** Example coronal brain sections (representative of 14 Rxfp3-Cre and 12 Efr3a-Cre mice) showing ChR2-eYFP expression (green). Dashed lines indicate borders of layer 5. Scale bar, 200 µm. **b** Laminar distributions of fluorescence-expressing cells (50 µm bins; left) and total cell counts from $n = 3$ independent brain sections (right; $p = 0.397$, t-test). Shading and error bars, SEM across $n = 3$ animals; circles, individual animal data. **c** Neurons were classified as wide-spiking (WS; filled triangles) or narrow-spiking (NS; filled circles) neurons (open squares, unclassified neurons) based on physiological characteristics. Red and blue, optogenetically tagged IT and PT neurons, respectively (gray, untagged neurons). Inset, averaged spike waveforms of example NS and WS neurons. Calibration, 250 µs and 50 µV. **d** Example responses of optically-tagged IT and PT neurons to light stimulation. Top, spike raster plots; each row is one trial and each dot is one spike. Bottom, peri-stimulus time histograms. Time 0 denotes the onset of 5-ms light stimulation (blue bar). The averaged spike waveforms of spontaneous (black) and optically driven (blue) spikes recorded through four tetrode channels are shown superimposed (inset). Calibration, 250 µs and 50 µV. **e** Cumulative plots for SALT-test $p$ value (left), spike waveform correlation between spontaneous and optically-driven spikes (middle), and peak response latency since laser onset (right) for WS IT, WS PT, NS IT, and NS PT neurons ($n = 42, 47, 19,$ and 15, respectively). **f** Example coronal brain sections from Rxfp3-Cre (left) and Efr3a-Cre (right) mice showing tdTomato (red, crossed with Ai9) and PV (green, immunostaining) expressions (representative of five Rxfp3-Cre and five Efr3a-Cre mice). Scale bar, left, 200 µm; right, 50 µm. **g** Quantification of Cre-tdTomato and PV expressions (mean ± SEM across five animals; circles, individual animal data). Note that there is no overlap between Cre-tdTomato and PV expressions.

$n = 4.7 \pm 5.6$; right target, $n = 6.7 \pm 6.3$ per session; mean ± SD) and large variability in neural activity across trials (c.f., Fig. 3a), prevented reliable estimation of target-dependent neural activity in error trials. Error-trial neural activity was analyzed using untagged WS neurons and also IT neurons recorded with calcium imaging (see below).

We found IT neurons tend to show delay-period activity that varies according to target (i.e., the left versus right water lick port; Fig. 3a). The number of time windows showing significant target-dependent firing (1-s time window advanced in 0.1 s steps; t-test, $p < 0.05$) was significantly larger for IT than PT neurons during the delay period (t-test, $t_{55} = 2.068$, $p = 0.043$; Fig. 3b, c). In addition, absolute target selectivity index (see Methods)

calculated with the entire delay-period (4 s) activity was significantly larger in IT than PT neurons (t-test, $t_{55} = 3.0641$, $p = 0.003$), although IT and PT neuronal populations did not show biased firing toward the ipsilateral versus contralateral target (target selectivity index, difference from 0, t-test, IT, $t_{25} = 0.806$, $p = 0.428$; PT, $t_{30} = -1.692$, $p = 0.101$; Fig. 3d). Also, decoding of target using delay-period neuronal ensemble activity improved as a function of ensemble size for the IT neurons, but remained similar to the level of chance (50%) across all tested PT neuron ensemble sizes (Fig. 3e). For instance, at ensemble size of 26 neurons (vertical dashed line in Fig. 3e), the neural decoding of target was significantly above the level of chance for IT, but not PT neurons ($n = 100$ decoding iterations, t-test, IT, $t_{99} = 11.048$,

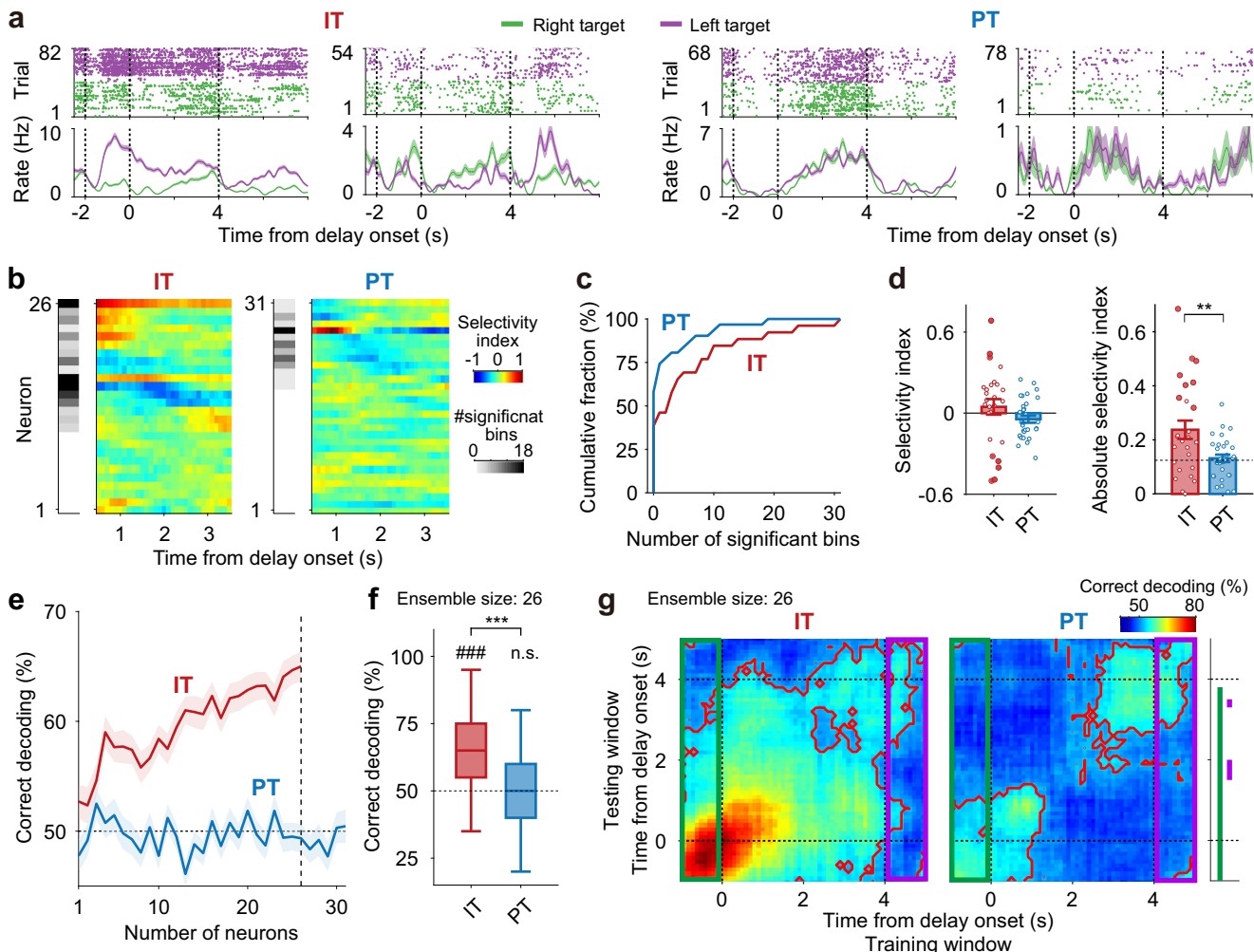

**Fig. 3 IT neurons convey stronger target signals than PT neurons.** Target-selectivity of delay-period activity was compared between IT and PT neurons using correct trials. **a** Example IT and PT neuronal responses during the fixed-delay (4 s) condition (correct trials only). Top, spike raster plots; bottom, spike density functions ($\sigma = 100$ ms). Green, right-target trials; purple, left-target trials. **b** Heat maps showing target-dependent firing (target selectivity index; 1-s window advanced in 0.1 s steps) of all analyzed IT and PT neurons. Red and blue denote higher firing in ipsilateral- and contralateral-target trials, respectively. The number of bins with significant target-dependent firing ($p < 0.05$, t-test) is indicated in gray scale for each neuron on the left. Neurons were sorted according to the time of absolute peak selectivity index. **c** Cumulative plots for the number of bins with significant target-dependent firing. **d** Distributions of target selectivity index (left; positive and negative, preferential firing toward ipsilateral and contralateral targets, respectively) and absolute target selectivity index (right). Dotted line represents mean target-dependent firing calculated with trial-shuffled data. Circles denote individual neurons (IT, $n = 26$, PT, $n = 31$ neurons) and filled circles indicate neurons with a significant selectivity index ($p < 0.05$, t-test). **$p < 0.01$ (IT versus PT, t-test). **e** Neural decoding of target as a function of ensemble size. **f** Decoding performance at ensemble size of 26 neurons (vertical dashed line in **e**). ###$p < 0.001$ (above chance level, t-test); ***$p < 0.001$ (IT versus PT, t-test). **g** Heat maps for cross-temporal population decoding ($n = 26$ neurons; 1-s window advanced in 0.1-s steps). Red contours indicate significant decoding above chance level ($p < 0.05$, t-test). Green and purple rectangles denote cross-temporal decoding using sample- and choice-phase neural activity as the training data, respectively. Green and purple bars on the right denote the analysis time windows with a significant difference between IT and PT neuronal cross-temporal decoding using sample- and choice-phase neural activity as the training data, respectively. Shading and error bars, SEM across trials (**a**), neurons (**d**), or 100 decoding iterations (**e**). Boxplots (**f**) show median, interquartile range, and maximum and minimum within 1.5 interquartile range.

$p = 5.8 \times 10^{-19}$; PT, $t_{99} = -0.501$, $p = 0.613$; IT versus PT, $t_{198} = 8.386$, $p = 9.4 \times 10^{-15}$; Fig. 3f). This cannot be attributed to the difference between Rxfp3-Cre and Efr3a-Cre mice because target decoding using untagged WS neurons was similar between the two animal groups (Supplementary Fig. 6). Thus, IT neurons carry stronger target signals than PT neurons during the delay period.

We then performed a cross-temporal decoding analysis[18,19] to examine neural dynamics underlying sensory-to-motor mapping during the delay period. We used the same sets of IT and PT neurons ($n = 26$ and 31, respectively) for this analysis and, for a direct comparison between IT and PT neurons, we matched their

ensemble sizes ($n = 26$ neurons). In this analysis, the decoder is trained with neuronal ensemble activity in one time window and tested with neuronal ensemble activity in a different time window. We found that IT neurons show stronger target-dependent activity than PT neurons during the sample phase (comparison of target decoding, last 1 s before delay onset, $n = 100$ decoding iterations, t-test, $t_{198} = 20.632$, $p = 3.2 \times 10^{-51}$; Fig. 3g). Also, as denoted by the red contours in Fig. 3g, IT, but not PT, neurons showed a broad domain of significant ($n = 100$ decoding iterations, t-test, $p < 0.05$) cross-temporal decoding during the delay period, indicating that IT neurons tend to maintain target-dependent activity persistently during the delay period. When we

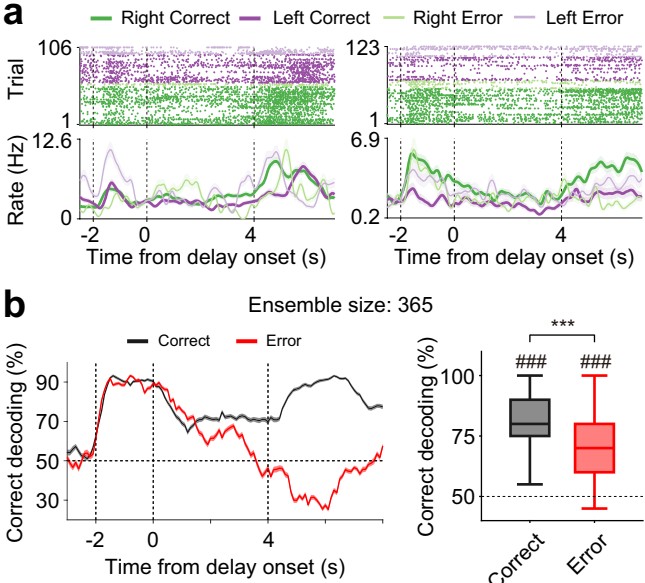

**Fig. 4 Delay-period activity of untagged WS neurons during correct and error trials. a** Example responses of untagged WS neurons ($n = 365$ recorded in the sessions with ≥2 error trials for both targets) during correct and error trials. The same format as in Fig. 3a except that correct (saturated colors) as well as error (soft colors) trials are shown. **b** Left, neuronal population decoding of target using untagged WS neurons ($n = 365$; 1-s window advanced in 0.1-s steps). Correct trials were used as training data, and both correct (black) and error (red) trials were used as test data. Right, neural decoding of target based on entire delay-period (4 s) activity. ###$p$ < 0.001 (difference from chance level, $t$-test); ***$p$ < 0.001 (correct versus error, paired $t$-test). Shading, SEM across trials (**a**) or 100 decoding iterations (**b**, left). Boxplots (**b**, right) show median, interquartile range, and maximum and minimum within 1.5 interquartile range.

used sample-phase activity as the training data (green rectangles in Fig. 3g), IT neurons yielded above chance decoding of target for a prolonged time period whereas PT neurons yielded poor target prediction except immediately after delay onset, so that decoding performance was significantly better for IT than PT neurons largely throughout the delay period ($t$-test; green bars on the right in Fig. 3g). When we used choice-phase activity as the training data (first 1 s since delay offset; purple rectangles in Fig. 3g), target prediction was poor throughout most of the delay period, except toward the end, for both IT and PT neurons, and little difference in decoding performance was found between IT and PT neurons (purple bars on the right in Fig. 3g). These results indicate that IT neurons carry robust target signals by maintaining sample-phase activity more persistently than PT neurons during the delay period.

**Delay-period activity differs between correct and error trials.** Next, we examined whether and how target-dependent delay-period activity differs between correct and error (wrong target choice) trials using those untagged WS neurons (i.e., randomly sampled putative pyramidal neurons) recorded in sessions with ≥2 error trials for both targets ($n = 365$ neurons, mean firing rate during task performance, 2.3 ± 2.4 Hz, mean ± SD; two examples are shown in Fig. 4a). We performed neural decoding of target using correct trials as training data. When we used correct trials as test data, a stable level of correct target decoding was achieved throughout the delay period. However, when we used error trials as test data, correct target decoding decreased gradually to the chance level during the delay period and below the chance level

following delay offset (Fig. 4b). Target prediction using the entire delay-period (4 s) ensemble activity was above the chance level in both correct and error trials ($t$-test, correct, $t_{99} = 32.632$, $p = 8.8 \times 10^{-55}$; error, $t_{99} = 16.565$, $p = 2.7 \times 10^{-30}$), but significantly better in correct than error trials (paired $t$-test, $t_{99} = 6.925$, $p = 4.6 \times 10^{-10}$; Fig. 4b). These results indicate that target signals were successfully registered during the sample phase, but poorly maintained during the delay period in error trials. When we analyzed miss (no choice) trials, the level of correct target decoding differed significantly from that of error trials during the late delay period and the choice phase (Supplementary Fig. 7). This suggests that the source for unsuccessful trial completion might differ between error and miss trials.

**Stronger coupling of IT than PT neurons to theta-frequency activity.** The theta-frequency component of the mPFC local field potential (LFP) has been implicated in hippocampus-prefrontal cortex (PFC) synchrony and spatial working memory[20,21]. We therefore examined LFP power and spike-LFP coupling in the theta band during the delay period (Fig. 5a) using the LFP and spike data recorded in the sessions with ≥2 error trials for both targets and <1% of total LFP signals reaching the ceiling ($n = 130$ sessions). Because high-frequency (type I) and low-frequency (type II) theta oscillations are observed primarily during voluntary motor behaviors and immobile states, respectively[22,23], and because we recorded neural activity in head-fixed mice, we focused on low-frequency theta in this study. LFP power in the low-frequency theta band (4–8 Hz) during the delay period (4 s) did not vary significantly between correct and error trials (paired $t$-test, $t_{129} = -1.095$, $p = 0.276$; Fig. 5b, c). However, spike-field coherence of WS neurons ($n = 352$ recorded in 130 sessions) in the theta band was significantly stronger in correct than error trials during the delay period ($t_{351} = 6.582$, $p = 1.7 \times 10^{-10}$; Fig. 5d, e). There was no significant difference in mean firing rate or burst firing between correct and error trials during the delay period ($p$ values >0.280; Fig. 5f, g). Thus, spike-field coherence in the theta band was stronger during the delay period in correct than error trials without a difference in overall firing rate or burst firing in the mPFC.

We then asked whether IT and PT ($n = 25$ and 31, respectively, recorded in the sessions with <1% of total LFP signals reaching the ceiling) neuronal spikes show different degrees of coupling to the theta LFP using only correct trials. Spike-field coherence in the theta band (4–8 Hz) differed significantly between IT and PT neurons during the delay period ($t$-test, $t_{54} = 2.230$, $p = 0.030$; Fig. 5h, i). However, there was no significant difference in mean firing rate or burst firing between IT and PT neurons during the delay period ($t$-test, $p$ values >0.870; Fig. 5j, k). Thus, spike-field coherence in the theta band was stronger in IT than PT neurons during the delay period without a significant difference in mean firing rate or burst firing. These results suggest IT neurons may contribute to working memory not only via target-dependent firing, but also by synchronous discharges at theta frequencies.

**PT neurons carry more temporal information than IT neurons.** Delay-period activity of PFC neurons conveys not only working memory, but also temporal information[14,24,25]. To assess the temporal information carried by delay-period activity, we performed neural decoding of temporal information using correct trials of IT and PT neurons ($n = 26$ and 31, respectively). We divided the delay period into ten equal bins, leaving out the first 0.5 s to exclude sensory response-related neural activity, leaving a total of 3.5 s. We then decoded bin identity based on IT or PT neuronal ensemble activity to construct a normalized decoding probability map[26] (actual bin number versus predicted bin number; see Methods;

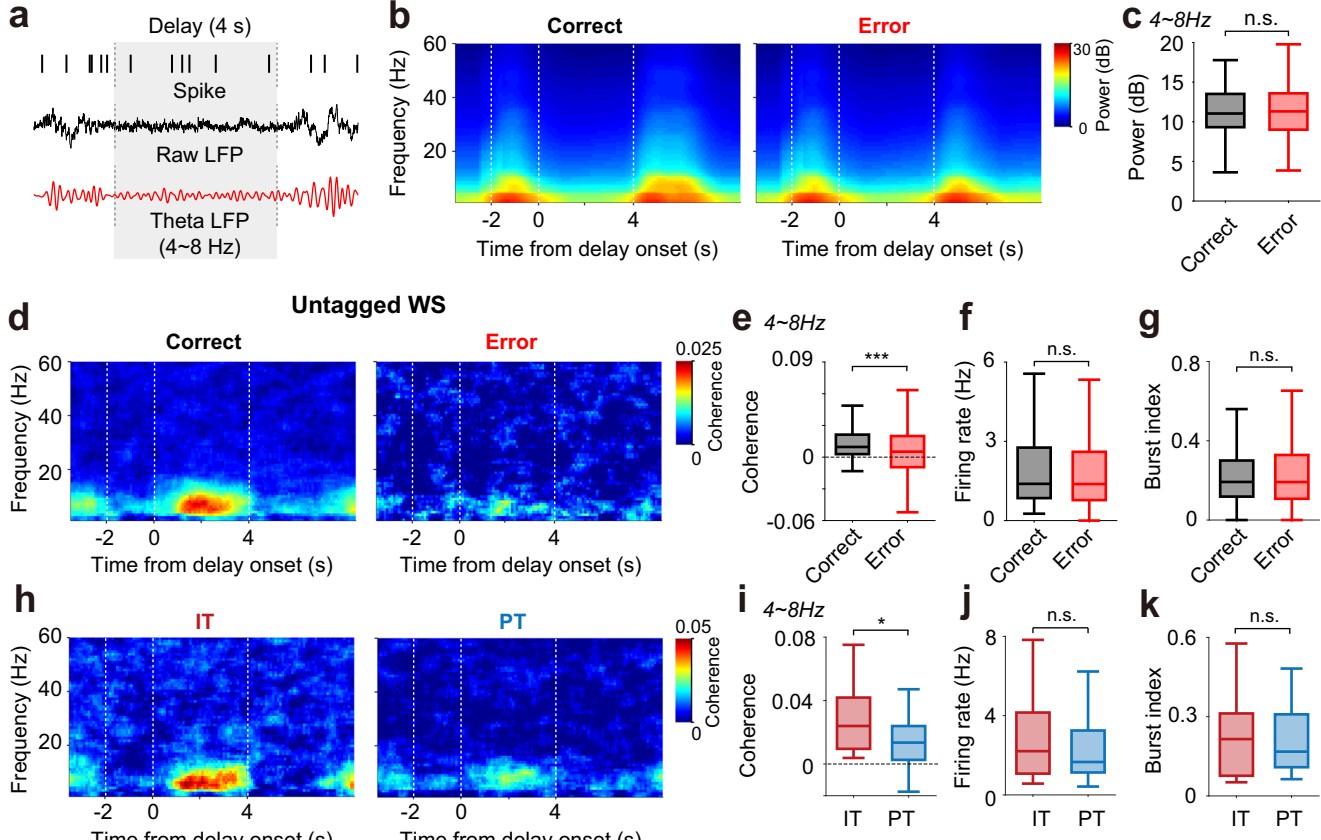

**Fig. 5 Coupling of spikes to theta-frequency activity. a** Example spike trains and LFP signals during the delay period. **b** LFP power spectrograms during correct and error trials ($n = 130$ sessions with ≥2 error trials for both targets). **c** Mean theta (4–8 Hz) LFP power during the delay period in correct and error trials. n.s., $p > 0.05$ (paired $t$-test). **d–g** Spike-field coherence of untagged WS neurons ($n = 352$ recorded in the sessions with ≥2 error trials for both targets) in correct and error trials. **d** Frequency-dependent spike-field coherence. The trial-shifted coherogram was subtracted from the raw coherogram to remove spurious covariation. **e–g** Mean spike-field coherences in the theta (4–8 Hz) band, firing rates, and burst indices during the delay period in correct and error trials. n.s., $p > 0.05$, ***$p < 0.001$ (paired $t$-test). **h–k** Spike-field coherences in the theta band, firing rates, and burst indices of IT ($n = 25$) and PT ($n = 31$) neurons in correct trials. The same format as in **d–g** except that only correct trials were analyzed. n.s., $p > 0.05$, *$p < 0.05$ ($t$-test). Boxplots (**c**, **e–g** and **i–k**) show median, interquartile range, and maximum and minimum within 1.5 interquartile range.

Fig. 6a). The correlation between actual and predicted bin numbers ($r$) was significantly greater for PT than IT neuronal ensembles ($n = 100$ decoding iterations at ensemble size = 26 neurons, $t$-test, $t_{198} = -19.720$, $p = 1.3 \times 10^{-48}$; Fig. 6b, c). In addition, decoding accuracy (see Methods) was significantly greater for PT than IT neurons ($t_{198} = -15.594$, $p = 2.7 \times 10^{-36}$; Fig. 6b, c). These results are not because of a general tendency of mPFC neurons of Efr3a-Cre mice to carry more temporal information than those of Rxfp3-Cre mice (Supplementary Fig. 8). These findings indicate that PT neurons convey more temporal information than IT neurons during the delay period.

To explore characteristics of timing-related mPFC neuronal activity, we performed a demixed principal component analysis (dPCA; analysis time window, [−3.5 8] s from delay onset) using the untagged WS neurons ($n = 365$) recorded in the sessions with ≥2 error trials for both targets. We confirmed that the target-independent demixed principal components (dPCs; those conveying no target signals) explained the majority of variance, as was previously reported[24,25]. These dPCs showed similar responses between correct and error trials, suggesting that they do not contribute to the animal's choice of target. This was in contrast to the target-dependent dPCs that showed different responses between correct and error trials (Fig. 6d; c.f., delay-period components of target-dependent PT neuronal dPCs did not carry significant target signals; Supplementary Fig. 9). In the dPCA

using correct trials of PT neurons ($n = 31$), the first principal component (dPC1) showed ramping activity during the delay period and accounted for ~60% of neural activity variance. In contrast, there was no monotonic delay-period ramping activity among the first three dPCs of IT neurons ($n = 26$; only correct trials were analyzed), although these explained ~80% of the total variance in neural activity (Fig. 6e). In addition, when dPC1 was removed from the analysis, PT, but not IT, neuronal decoding of the elapse of delay was greatly reduced (Fig. 6f, g; $n = 100$ decoding iterations, difference from zero, $t$-test, $\Delta r$, IT, $t_{99} = 0.269$, $p = 0.789$; PT, $t_{99} = 60.518$, $p = 5.2 \times 10^{-80}$; $\Delta$decoding accuracy, IT, $t_{99} = 0.944$, $p = 0.348$; PT, $t_{99} = 36.611$, $p = 2.4 \times 10^{-59}$; IT versus PT, $t$-test, $\Delta r$, $t_{198} = -48.300$, $p = 1.6 \times 10^{-111}$; $\Delta$decoding accuracy, $t_{198} = -23.601$, $p = 1.9 \times 10^{-59}$). This indicates that PT neuronal temporal information during the delay period was largely carried by the ramping activity (Fig. 3a). Collectively, these results indicate that the delay-period activity of IT neurons conveys strong target signals with limited temporal information. In contrast, the delay-period activity of PT neurons conveys a large amount of temporal information in the form of ramping activity with minimal target signal.

**IT and PT neurons are responsive to diverse task events.** Next, we compared IT ($n = 26$) and PT ($n = 31$) neuronal activity during the sample and choice phases. For this, we first compared

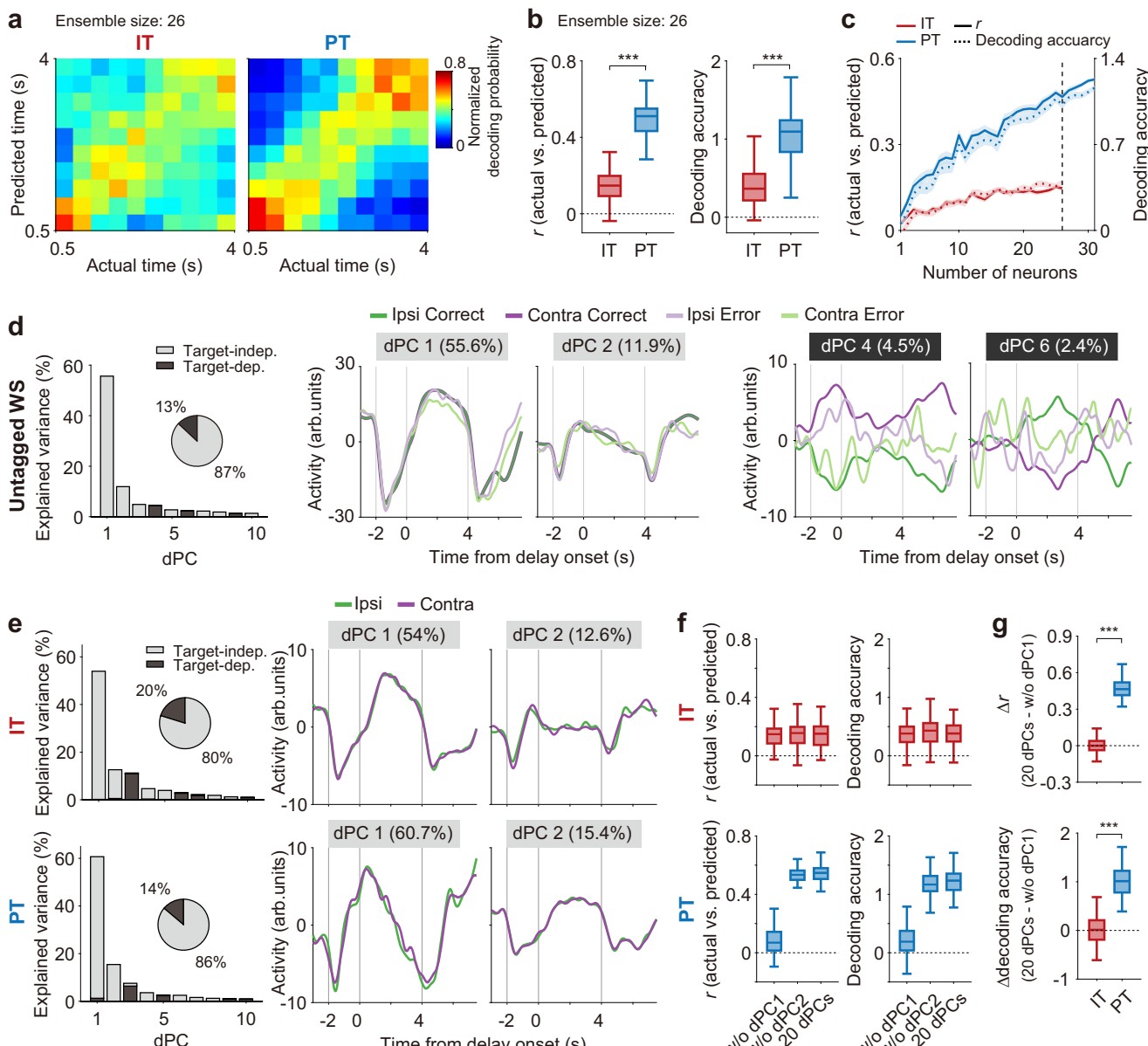

**Fig. 6 PT neurons convey more temporal information than IT neurons. a–c** Neural decoding of time using IT and PT neurons (*n* = 26 and 31, respectively). **a** Mean normalized decoding probabilities (actual versus predicted bins) using 26 neurons. **b** Decoding performances of IT and PT neuronal ensembles (*n* = 26 neurons; vertical dashed line in c). Left, correlation (*r*) between the actual and predicted bins; right, decoding accuracy (see Methods). ***p* < 0.001 (*t*-test). **c** Decoding performance as a function of ensemble size. Shading, SEM across 100 decoding iterations. **d** Left, variance in neural activity explained by individual dPCs for untagged WS neurons (*n* = 365 recorded in the sessions with ≥2 error trials for both targets). Gray and black denote target-independent and target-dependent dPCs, respectively (see Methods). The pie chart shows the total variance explained by the target-dependent and target-independent dPCs. Middle and right, time courses of the top two target-independent (middle) and target-dependent (right) dPCs. Same color code as in Fig. 4 (green and purple for the ipsilateral and contralateral targets, respectively; saturated and soft colors, correct and error trials, respectively). **e** Left, variance explained by individual dPCs for IT (*n* = 26) and PT (*n* = 31) neurons. Right, dPC1 and dPC2. **f** Temporal decoding using the first 20 dPCs compared with those lacking dPC1 (w/o dPC1) or dPC2 (w/o dPC2). **g** The amount of decoding lost by excluding dPC1 (upper, correlation; lower, decoding accuracy). ***p* < 0.001 (*t*-test). Boxplots (**b**, **f**, and **g**) show median, interquartile range, and maximum and minimum within 1.5 interquartile range. Only correct trials were analyzed for IT and PT neurons (**a–c** and **e–g**), and both correct and error trials were analyzed for untagged WS neurons (**d**).

target-dependent firing of IT and PT neurons during the 1-s time windows since the first lick after sample onset (sample phase) and delay offset (choice phase; see Methods). Two-way mixed ANOVA revealed that some neurons were significantly responsive to target (Fig. 7a, middle panel), some to phase (Fig. 7a, bottom panel), and some to target × phase interaction. The fraction of neurons significantly responsive to target × phase interaction (two-way mixed ANOVA, *p* < 0.05) was significantly larger

for IT than PT neurons (Fisher's exact test, *p* = 0.039; Fig. 7b), indicating more conjunctive encoding of target and phase by IT than PT neurons. Note that activity of these interaction-sensitive neurons cannot be explained by pure motor-related responses (left lick versus right lick). Both IT and PT neuronal populations showed unbiased bidirectional responses to target (target selectivity index, difference from 0, *t*-test, IT, $t_{25} = 0.675$, *p* = 0.5058; PT, $t_{30} = 0.528$, *p* = 0.601) as well as phase (phase selectivity

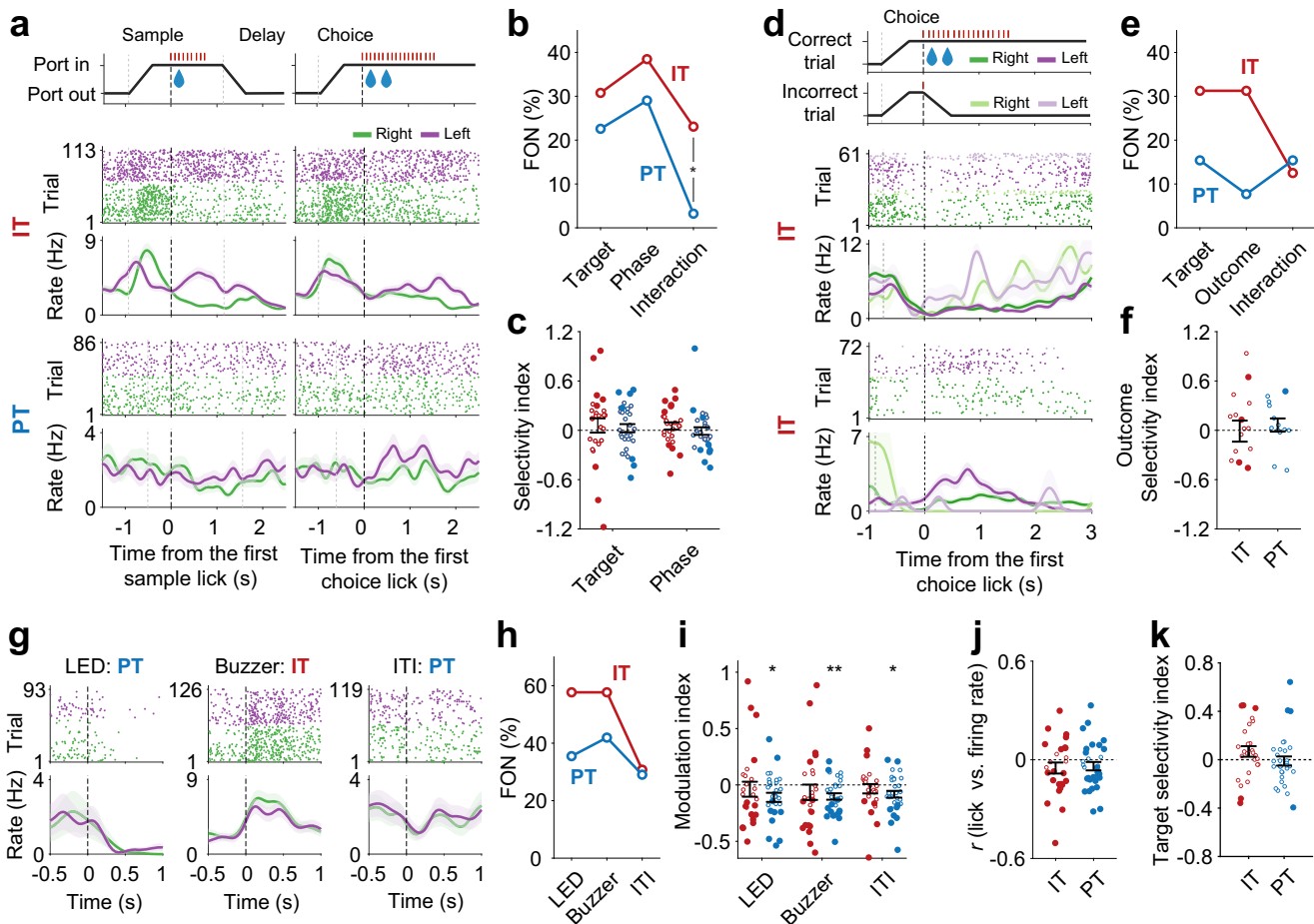

**Fig. 7 IT and PT neurons are responsive to diverse task events. a** Example neuronal responses during the sample and choice phases in correct trials. Top, schematic of task phases. Bottom, spike rasters and spike density functions ($\sigma = 100$ ms) of two example neurons responsive to target (upper) or phase (lower; two-way mixed measures ANOVA). Trials were grouped according to target (green, right lick port; purple, left lick port). **b** Fractions of neurons (FON) significantly responsive to target, phase, and their interaction. *$p < 0.05$ (Fisher's exact test). **c** Response selectivity to target (positive and negative, higher responses to the ipsilateral and contralateral targets, respectively; sample- and choice-phase data combined) or phase (positive and negative, higher activity during the sample and choice phases, respectively; ipsilateral- and contralateral-target trials combined). **d** Example neurons showing outcome-dependent firing during the choice phase. Trials were grouped according to target (green and purple) and outcome (saturated and soft colors, correct and error trials, respectively). **e** Fractions of neurons significantly responsive to target, outcome, and their interaction (two-way ANOVA). **f** Response selectivity to outcome (positive and negative, higher activity in correct and error trials, respectively). **g** Example neurons showing differential firing between before and after LED, buzzer, or ITI onset. **h** Fractions of neurons significantly (paired $t$-test, 0.5-s windows before and after event onset were compared) responsive to LED, buzzer, or ITI onset. **i** Distributions of event modulation index (positive and negative, higher and lower firing after than before event onset, respectively). *$p < 0.05$, **$p < 0.01$ (difference from 0, $t$-test). **j** Distributions of correlation coefficient between lick rate and firing rate. **k** Target selectivity index during the 1-s time period before the first choice lick. Filled and open circles denote significantly and insignificantly responsive neurons, respectively, in **c**, **f**, and **i–k**. Shading, SEM across trials (**a**, **d**, and **g**). Error bars, SEM across neurons (**c**, **f**, and **i–k**). Only correct trials were analyzed ($n = 26$ IT and 31 PT neurons) except the analyses related to outcome-dependent neural activity (**d–f**; $n = 16$ IT and 13 PT neurons recorded in the sessions with $\geq 2$ error trials for both targets).

index, difference from 0, $t$-test, IT, $t_{25} = 1.185$, $p = 0.247$; PT, $t_{30} = -0.197$, $p = 0.845$; Fig. 7c). We then compared IT and PT neuronal activity between correct and error trials during the choice phase (first 1 s since the first lick after delay offset) to examine trial outcome-dependent firing using those IT and PT neurons ($n = 16$ and 13, respectively) recorded in the sessions with $\geq 2$ error trials for both targets. Some neurons were significantly responsive to trial outcome (Fig. 7d, middle and bottom panels), some to target, and some to outcome × target interaction (two-way ANOVA, $p < 0.05$; Fig. 7e). Again, activity of the interaction-sensitive neurons cannot be explained by pure motor-related responses. The fraction of neurons responsive to one or more of these variables (two-way mixed ANOVA, $p < 0.05$) did not differ significantly between IT (9 out of 16) and PT (4 out of

13) neurons. Also, IT and PT neuronal populations did not show biased directional responses to trial outcome (outcome selectivity index, difference from 0, $t$-test, IT, $t_{15} = -0.078$, $p = 0.939$; PT, $t_{12} = 0.796$, $p = 0.442$; Fig. 7f). These results show that IT and PT neurons are responsive to diverse combinations of target, outcome, and task phase.

We then examined IT ($n = 26$) and PT ($n = 31$) neuronal responses to various sensory events by comparing neuronal activity before (0.5 s) and after (0.5 s) LED onset, buzzer onset, or inter-trial interval (ITI) onset (both lick ports began to retract at ITI onset) using paired $t$-test. Figure 7g shows example neurons that are significantly responsive to these events. Both IT and PT neuronal populations contained neurons that were significantly responsive to LED onset, buzzer onset, or ITI onset. The fraction

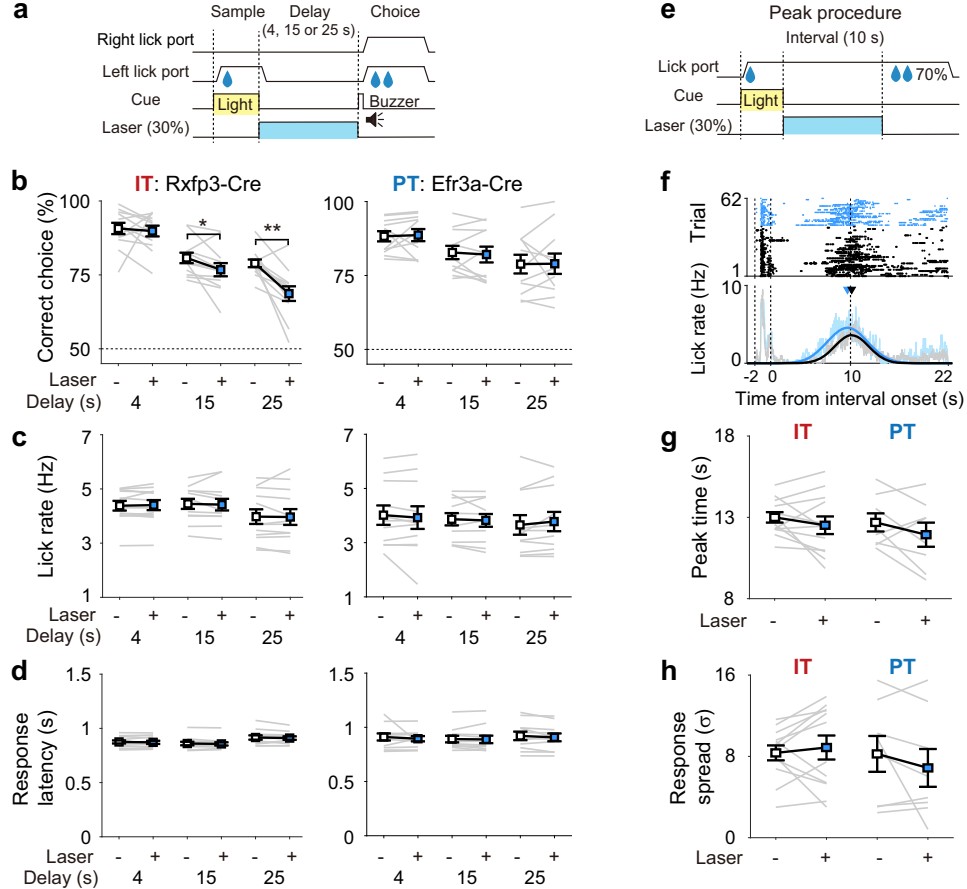

**Fig. 8 Inactivation of IT neurons impairs working memory. a** Schematic for optogenetic inactivation. Laser was given throughout the delay period (blue bar; 4, 15, or 25 s) in a randomly chosen 30% of trials. **b** Behavioral performance in the absence (−) or presence (+) of laser stimulation for different delay durations (4, 15, and 25 s). *$p < 0.05$; **$p < 0.01$ (two-way repeated measures ANOVA followed by Bonferroni's post hoc tests). **c** Lick rate. **d** Response latency (the latency to the first lick since delay offset). **e** Schematic for optogenetic inactivation in the peak procedure. **f** An example peak procedure session. Only omission trials are shown. Lick raster plots (upper) and lick histograms (bottom) with (blue) or without (gray) laser stimulation are shown. Peak time and SD of the lick response were determined by fitting a Gaussian curve to each lick histogram (thick curves). **g, h** Group data. Peak time (**g**) and SD (**h**) of Gaussian-fitted lick responses during omission trials with (+) or without (−) laser stimulation. Gray lines, individual animals; Squares and error bars, mean and SEM across animals ($n = 12$ Rxfp3-Cre and 11 Efr3a-Cre mice).

of significantly responsive neurons did not differ significantly between IT and PT neurons for all of these events (Fisher's exact test, p-values >0.115; Fig. 7h). While event-responsive IT neurons ($n = 15$, 15, and 8 for LED, buzzer, and ITI onsets, respectively) showed unbiased bidirectional responses to these events, event-responsive PT neurons ($n = 11$, 13, and 9 for LED, buzzer, and ITI onsets, respectively) tended to decrease their firing rates after compared to before an event onset (event modulation index, difference from 0, t-test, LED, IT, $t_{25} = -0.570$, $p = 0.574$, PT, $t_{30} = -2.724$, $p = 0.011$; buzzer, IT, $t_{25} = -0.956$, $p = 0.348$, PT, $t_{30} = -3.620$, $p = 0.001$; ITI, IT, $t_{25} = -0.809$, $p = 0.426$, PT, $t_{30} = -2.738$, $p = 0.010$; Fig. 7i).

We also examined how IT and PT neuronal activity is affected by licking behavior. For this, we divided the entire fixed-delay session into 1-s bins, and examined correlation between mean lick rates and mean firing rates during 1-s windows. We found that the majority of IT and PT neurons show significant correlations (IT, 21 out of 26, 80.8%; PT, 25 out of 31, 80.7%). Both positive and negative correlations were found between firing rate and lick rate for IT and PT neurons with no significant bias toward the positive or negative direction (t-test, IT, $t_{25} = -1.498$, $p = 0.147$; PT, $t_{30} = -1.509$, $p = 0.142$; Fig. 7j). Finally, given the proposed role of motor cortex PT neurons in motor planning[7,27], we examined upcoming movement-related activity of IT and PT

neurons by analyzing neural activity during the 1-s time period before the first lick since delay offset. Similar fractions of IT and PT neurons showed significantly different activity (t-test, $p < 0.05$) before the ipsilateral versus contralateral licks (IT, 5 out of 26, 19.2%; PT, 4 out of 31, 12.9%; Fisher's exact test, $p = 0.718$), and neither showed biased firing toward the ipsilateral or contralateral lick (t-test, IT, $t_{25} = 1.558$, $p = 0.132$; PT, $t_{30} = -0.293$, $p = 0.771$; Fig. 7k), which argues against a direct involvement of mPFC PT neurons in controlling the direction of licking. To summarize, both IT and PT neurons were responsive to diverse combinations of sensory/motor events during the task ("mixed selectivity")[28].

**Inactivation of IT neurons impairs working memory.** We then examined the effect of selective inactivating IT or PT neurons on behavioral performance using separate groups of Rxfp3-Cre ($n = 12$) and Efr3a-Cre ($n = 11$) mice in the fixed-delay condition. We expressed soma-targeted *Guillardia theta* anion-conducting channelrhodopsin-2 (stGtACR2)[29] and implanted optic fibers bilaterally in the mPFC. After training the animals with 4-s fixed delay to the performance criterion, we applied continuous laser stimulation during the entire delay period in a randomly selected 30% of 100–200 total trials (Fig. 8a; histological identification of optic fiber locations in Supplementary Fig. 1). This procedure was

repeated with 15- and then 25-s fixed delays. We found optical inhibition significantly impaired behavioral performance in trials with 15- and 25-s delays in Rxfp3-Cre mice (two-way repeated measures ANOVA, main effect of laser stimulation, $F(1,11) = 18.760$, $p = 0.001$; main effect of delay duration, $F(2,22) = 46.117$, $p = 1.3 \times 10^{-8}$; stimulation × duration interaction, $F(2,22) = 4.254$, $p = 0.027$; Bonferroni's post hoc test for stimulation effect, 4-s duration, $p = 0.727$; 15-s, $p = 0.037$; 25-s, $p = 0.002$; Fig. 6b, left), but not in Efr3a-Cre mice (main effect of laser stimulation, $F(1,10) = 0.003$, $p = 0.959$; main effect of delay duration, $F(2,20) = 6.558$, $p = 0.006$; stimulation × duration interaction, $F(2,20) = 0.094$, $p = 0.911$; Fig. 8b, right). Importantly, this occurred without any effect on lick rate or lick response latency (two-way repeated measures ANOVA, no significant stimulation or stimulation × duration interaction effects, $p$ values >0.164; Fig. 8c, d). Thus, behavioral performance was compromised by inactivating IT, but not PT, neurons when the duration of delay was sufficiently long (≥15 s). These results indicate that selective inactivation of IT neurons, but not PT neurons, impairs working memory.

The lack of any significant effect of IT/PT neuronal inactivation on response latency in the present task may arise from the fact that the delay offset was explicitly signaled by the sound of a buzzer and by the protrusion of the lick ports. We therefore tested the same animals in a peak procedure. In this procedure, a response after a fixed latency since trial onset is rewarded in some percentage of trials, and internal timing is evaluated based on the temporal response profile in reward-omission trials[30,31]. One lick port was advanced to the animal and delivered water. Then, after a

10-s interval, water was again delivered in a randomly chosen 70% of trials (Fig. 8e). We also delivered laser stimulation throughout the 10-s interval period in a randomly chosen 30% of trials (Fig. 8e). In this experiment, we did not observe any significant change in the time course of the animal's licking response (peak time and SD of licking response) upon inactivation of either IT or PT neurons during the reward-omission trials (paired $t$-test, $p$ values >0.244; Fig. 8f–h). This indicates that neither IT nor PT neurons are indispensable for controlling the timing of the licking response in the present peak procedure.

**Responses of optically-tagged NS neurons.** We subjected only optically-tagged WS neurons to the main analyses because they showed physiological characteristics of typical cortical pyramidal neurons[32,33] and they out-numbered optically-tagged NS neurons (Fig. 2c). However, that none of the optically-tagged neurons express PV raises a possibility that optically-tagged NS neurons might be projection neurons as well. We therefore examined target-dependent activities of NS IT and NS PT neurons ($n = 19$ and 14, respectively) as well as timing-related activities of them. On the one hand, we found that NS IT neurons carry stronger target signals than PT neurons during the delay period ($n = 100$ decoding iterations at ensemble size = 14 neurons, $t$-test, above chance level, NS IT, $t_{99} = 10.798$, $p = 2.0 \times 10^{-18}$; NS PT, $t_{99} = -0.333$, $p = 0.740$; NS IT versus NS PT, $t_{198} = 8.044$, $p = 7.8 \times 10^{-14}$; Fig. 9a, b), which is consistent with the results obtained from the analysis of optically tagged WS neurons. On the other hand, we found that both NS IT and NS PT neurons carry substantial amounts of temporal

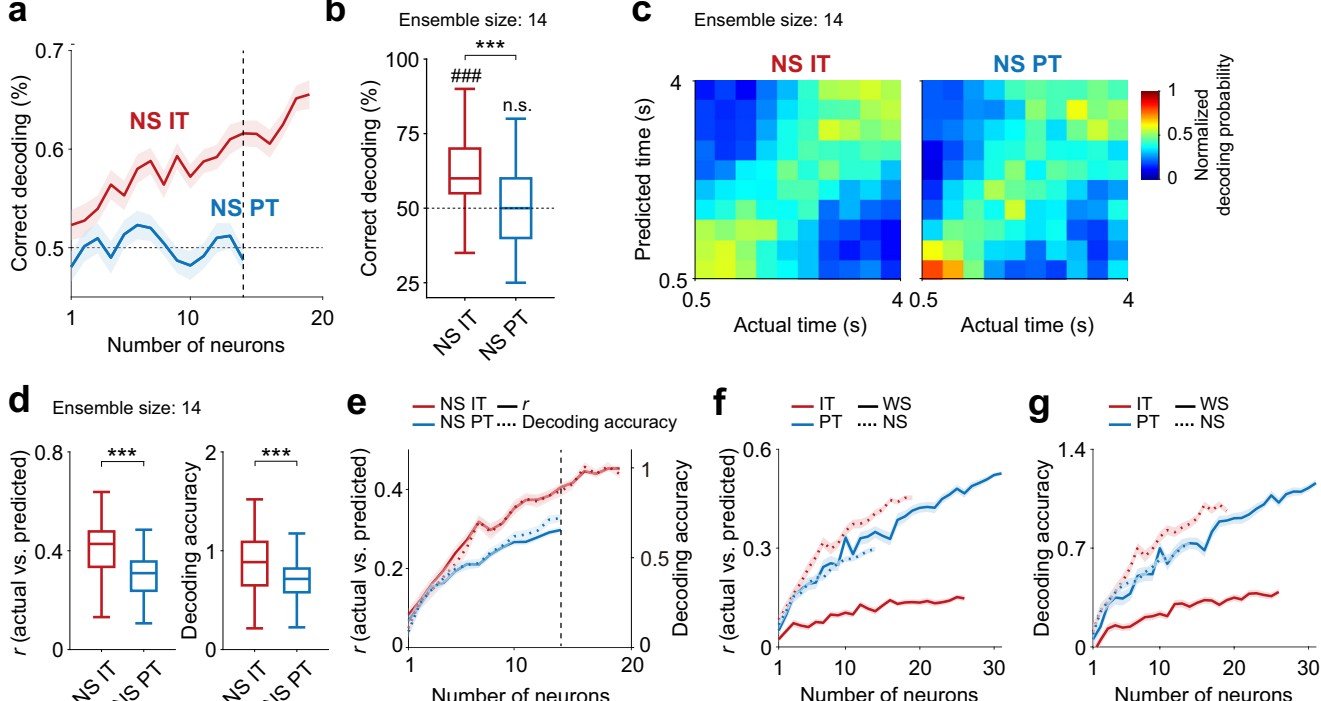

**Fig. 9 Target and temporal information carried by optically-tagged NS neurons. a**, **b** Neural decoding of target based on delay-period (4 s) neuronal ensemble activity ($n = 19$ NS IT and 14 NS PT neurons). **a** Decoding performance as a function of ensemble size. **b** Decoding performance at ensemble size of 14 neurons (vertical dashed line in **a**). ###$p < 0.001$ (above chance level, $t$-test); ***$p < 0.001$ (IT versus PT, $t$-test). The same format as in Fig. 3e, f. **c**–**e** Timing-related activity of NS IT and NS PT neurons ($n = 19$ and 14, respectively). **c** Heat maps showing mean normalized decoding probabilities (actual versus predicted bins; ensemble size = 14 neurons). **d** Decoding performance (left, correlation between the actual and predicted bins; right, decoding accuracy) of NS IT and NS PT neuronal ensembles ($n = 14$, vertical dashed line in **e**). ***$p < 0.001$ ($t$-test). **e** Decoding performance as a function of ensemble size. The same format as in Fig. 6a–c. **f**, **g** Comparison of temporal decoding across different types of neurons. **f** Correlation between the actual and predicted bins. **g** Decoding accuracy. Shading, SEM across 100 decoding iterations (**a**, **e**–**g**). Boxplots (**b** and **d**) show median, interquartile range, and maximum and minimum within 1.5 interquartile range.

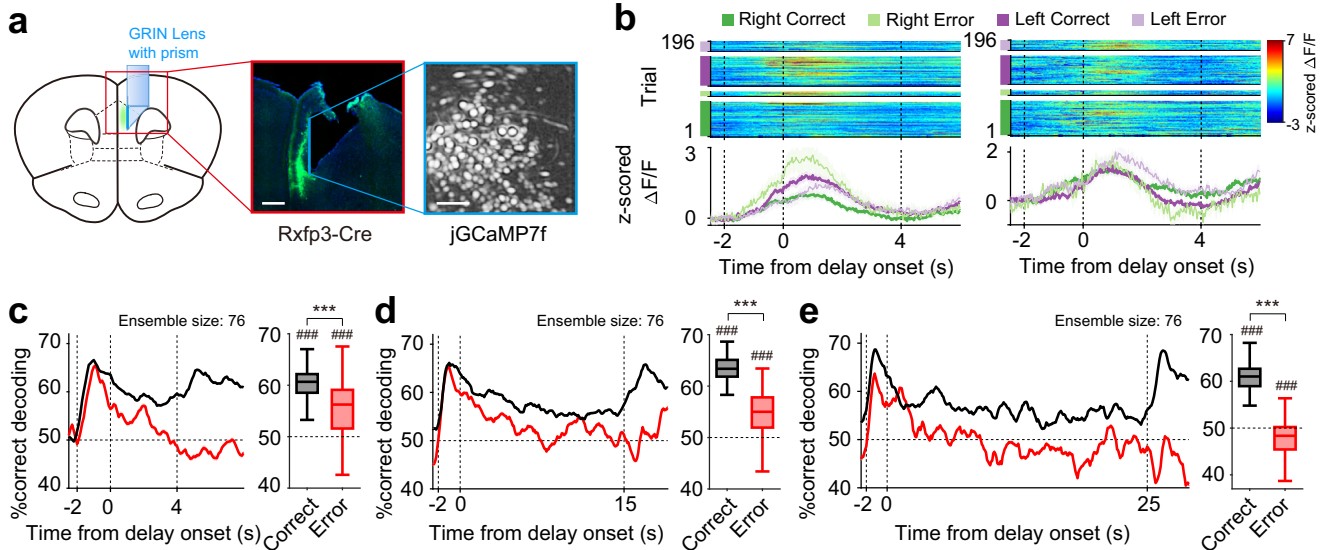

**Fig. 10 Activity of IT neurons during extended delay periods. a** Schematic (left) and a coronal brain section (representative of four mice; middle) showing the GRIN prism lens position and the spread of jGCaMP7f (green) in the mPFC. Right, field of view in an example session. Scale bar, left, 500 μm, right, 100 μm. **b** Example calcium responses of IT neurons during correct and error trials. The same format as in Fig. 4a except that ΔF/F traces are shown instead of spike rasters. **c–e** Left, results of neural decoding using calcium transient signals obtained from four Rxfp3-Cre mice (ensemble size = 76 neurons; 1-s window advanced in 0.1-s steps). Delay durations, 4, 15, and 25 s. Right, results of neural decoding using the entire delay-period activity (4, 15, or 25 s). ###$p < 0.001$ (difference from chance level, $t$-test); ***$p < 0.001$ (correct versus error, paired $t$-test). Shading, SEM across trials (**b**) and 100 decoding iterations (**c–e**, left). Boxplots (**c–e**, right) show median, interquartile range, and maximum and minimum within 1.5 interquartile range.

information and that NS IT neurons carry significantly more temporal information than NS PT neurons ($n = 100$ decoding iterations at ensemble size = 14 neurons, $t$-test, $r$, $t_{198} = 8.150$, $p = 4.1 \times 10^{-14}$; decoding accuracy, $t_{198} = 4.059$, $p = 7.1 \times 10^{-5}$; Fig. 9c–e), which is inconsistent with the results obtained from the analysis of optically tagged WS neurons. The amount of temporal information carried by optically-tagged NS neurons was comparable to that carried by WS PT neurons at equivalent ensemble sizes (Fig. 9f, g). In fact, NS IT neurons carried a significantly larger amount of temporal information than WS PT neurons, albeit the difference was small ($n = 100$ decoding iterations at ensemble size = 19 neurons, $t$-test, $r$, $t_{198} = 2.273$, $p = 0.024$; decoding accuracy, $t_{198} = 1.703$, $p = 0.090$). Thus, optically tagged NS and WS neurons were similar to one another regarding target-dependent neural activity, but different from one another regarding timing-related neural activity.

For comparison with WS IT and WS PT neurons, we also examined sensory/motor event-related activity of NS IT and NS PT neurons. Both NS IT ($n = 19$) and NS PT ($n = 14$) neurons showed significant responses to diverse sensory/motor events during the task. However, the patterns of response differed somewhat from those of WS IT and WS PT neurons (Supplementary Fig. 10). For example, whereas WS PT neurons tended to decrease firing rates after event onset, NS PT neurons tended to increase firing rates after event onset (Supplementary Fig. 10i). Finally, we compared theta band spike-field coherence of NS IT and NS PT neurons. Theta band spike-field coherence during the delay period did not differ significantly between NS IT and NS PT neurons (Supplementary Fig. 11). These results further support the possibility that optically tagged WS and NS neurons represent distinct populations.

**IT neuronal population maintains target signals for a long duration.** While our physiological recording of mPFC delay-period activity is only up to 7 s (4 s fixed delay and 1–7 s variable delay), we found significant behavioral effects of optogenetic IT neuronal inactivation only at longer delays (15 and 25 s). We therefore tested whether IT neurons can maintain working memory up to 25 s. For

this, we recorded activity of IT neurons using calcium imaging (see Methods; Fig. 10a, b) from the mPFC (histological identification of lens locations in Supplementary Fig. 1) of four Rxfp3-Cre mice performing the delayed response task with 4, 15, or 25-s fixed delay (performance criterion, >70% correct choices; performance during recording, 4 s, $82.8 \pm 6.8\%$ correct choices; 15 s, $78.8 \pm 6.0\%$; 25 s, $77.0 \pm 2.2\%$; mean $\pm$ SD). Because we recorded calcium signals from a large number of IT neurons simultaneously ($n = 247.8 \pm 106.2$ per session with the minimum calcium event rate of 0.025 Hz during task performance; mean event rate, $0.062 \pm 0.041$ Hz; mean $\pm$ SD), we analyzed delay-period neural activity during correct as well as error trials. We also performed neural decoding of target using only simultaneously recorded IT neurons (i.e., without pooling neurons across sessions), which allowed to use all error trials in the analysis. We analyzed one session for each delay duration for each animal (number of errors in each analyzed session, 4-s delay, $n = 13, 46, 20$, and 15; 15-s delay, $n = 21, 40, 21$, and 34; 25-s delay, $n = 20, 27, 29$, and 41). Note that we could not classify the recorded neurons as WS and NS neurons with calcium imaging. We found that IT neurons stably maintain target signals throughout the delay up to 25 s in correct trials (Fig. 10c–e; $n = 100$ decoding iterations, difference from chance level, $t$-test, 4 s, $t_{99} = 37.414$, $p = 3.1 \times 10^{-60}$; 15 s, $t_{99} = 51.737$, $p = 1.8 \times 10^{-73}$; 25 s, $t_{99} = 41.143$, $p = 4.6 \times 10^{-64}$). We also found that target-dependent IT neuronal activity deteriorates during the delay period in error trials (Fig. 10b) so that target prediction was significantly poorer in error than correct trials (Fig. 10c–e; $n = 100$ decoding iterations, correct versus error, paired $t$-test, 4 s, $t_{99} = 7.995$, $p = 2.5 \times 10^{-12}$; 15 s, $t_{99} = 18.311$, $p = 1.5 \times 10^{-33}$; 25 s, $t_{99} = 29.651$, $p = 4.8 \times 10^{-51}$). These results indicate that IT neurons can convey target signals at least up to 25 s and that such signals are poorly maintained in error trials.

## Discussion

Working memory and interval timing are critical components of cognition. Here, we found that IT neurons convey stronger target signals than PT neurons during delay period. In contrast, PT

neurons carry more temporal information than IT neurons during delay period. These results show that, rather than being supported by a common neural substrate[24,34,35], working memory and interval timing are mediated primarily by two distinct types of projection neurons in the mouse mPFC. This finding may reflect a more generalized functional segregation between neural networks that include IT and PT neurons. For example, the PFC may contribute to high-order cognitive functions by sharing working memory-related signals with other cortical areas (IT neuronal network) and to the temporal organization of behavior by sending temporal information to its downstream subcortical structures (PT neuronal network). Given the widespread nature of the cortical neural activity related to working memory and interval timing[36–41], future studies will be needed to determine whether parallel processing of working memory and temporal information by IT and PT neurons is a general characteristic across the many areas of the cortex.

While PT neurons make profuse connections with one another and receive strong projections from IT neurons, IT neurons have sparser interconnections and do not receive projections from PT neurons[2,42–44]. Because profuse interconnections may promote reverberatory activity, PT neurons have been proposed to play a more important role in working memory than IT neurons[44–46]. But inconsistent with this hypothesis, our results indicate a major role of IT neurons, rather than PT neurons, in supporting working memory in the present delayed response task. First, IT neurons, but not PT neurons, persistently maintained target signals during the delay period. Results from the cross-temporal decoding analysis indicate that IT neurons maintain sample-phase activity more persistently than PT neurons during the delay period. Note that our study was not designed to tease apart retrospective memory from prospective memory. Nevertheless, these results indicate a major contribution of IT neurons to maintaining sensory information (retrospective memory) that can be used to generate an appropriate behavioral response (prospective memory) in the absence of the relevant sensory information. Also, that decoding based on sample- and choice-phase activity deteriorates and elevates, respectively, before delay offset suggests that retrospective memory gives rise to prospective memory toward the end of the delay period, which is consistent with previous reports[47,48]. Second, the inactivation of IT neurons, rather than PT neurons, during the delay period impaired behavioral performance. Similar numbers of fluorescence-expressing neurons were found in Rxfp3-Cre and Efr3a-Cre mice (Fig. 2b), indicating comparable efficiencies for Cre-dependent gene expression between the two mouse lines. Thus, even though we cannot rigorously rule out the possibility for different optical inhibition efficiencies for IT versus PT neurons, our results indicate an indispensable role of mPFC IT neurons, but not PT neurons, in the current task. Recent studies indicate a more important role of agranular insular cortex than mPFC in working memory[19,49]. However, these studies used an olfactory delayed paired association task that requires the maintenance of olfactory information, but not sensory-to-motor mapping, during the delay period. The mPFC may be dispensable for static maintenance of sensory information. Third, IT neurons showed stronger coupling to theta frequency oscillations than PT neurons during the delay period. Delay-period theta oscillations and spike-LFP theta synchrony are enhanced in the mPFC in correct trials during working memory tasks[20,50,51], and this enhancement is accompanied by increased hippocampus-mPFC theta synchrony[20,52] with the level of synchrony correlated with behavioral performance[53]. In addition, monosynaptic hippocampal projections to the mPFC preferentially target IT neurons[54]. Even though these studies probed spatial working memory in freely navigating animals unlike in our study, our

results are well in line with the findings in these studies. Together, our results consistently indicate the importance of IT neurons, rather than PT neurons, in working memory in the present delayed response task. These findings need to be confirmed using different types of working memory task in the future, because roles of IT and PT neurons in working memory might vary across different behavioral settings.

Previous studies have shown that somatostatin (SST)-expressing interneurons in the mPFC play an important role in working memory. SST neurons, but not PV neurons, show significant target-dependent activity during the delay period, and the inactivation of SST neurons, but not PV neurons, impairs performance in a spatial working memory task[55]. Also, during probabilistic classical conditioning tasks, SST neurons convey strong expected outcome information until the trial outcome[56]. In addition, SST neurons facilitate hippocampus-mPFC theta synchrony and their inactivation disrupts the directionality of synchrony during a spatial working memory task[21]. These results suggest IT and SST neurons might work together to maintain target signals during the delay period and to regulate the hippocampus-mPFC theta synchrony that facilitates the transfer of spatial information from the hippocampus to the mPFC. It will be important in future studies to investigate how IT and SST neurons work together to support working memory.

The PFC has been implicated in interval timing[57]. We found mPFC PT neurons carry more temporal information than IT neurons during the delay period. This suggests PT neurons play a major role in keeping track of the passage of time in the mPFC. Temporally varying delay-period neural activity might be related to tracking the passage of delay or predicting the time of a future event such as a motor response and reward delivery. We found PT neurons carry comparable temporal information under the variable-delay condition during which the time for a motor response or reward delivery could not be predicted. This suggests the role of PT neurons in keeping track of the elapse of delay rather than predicting the time of a future event. We have shown previously that the mPFC conveys temporal information largely based on ramping activity[58]. Consistent with this, we found that most PT neuronal temporal information is supported by ramping activity. Unlike upcoming choice-related PT neuronal activity before delay offset, PT neuronal ramping activity spanning the delay period showed similar activity profiles between ipsilateral and contralateral choice trials, further supporting its role in timing rather than motor planning. PT neurons show spike doublets and weaker spike adaptation than IT neurons[43]. In addition, PT neuron pairs show higher paired-pulse ratios and stronger interconnectivity than IT neuron pairs[44]. These facilitatory properties of PT neurons may shape PT neuron ramping activity, but this remains to be tested.

Although PT neurons convey more temporal information, optogenetic inactivation of PT neurons does not significantly change animal response latency. This indicates PT neurons are dispensable for controlling the timing of the licking response in the present behavioral tasks (delayed match-to-sample and peak procedure tasks). We found that optically-tagged NS (both IT and PT) neurons convey high levels of temporal information during the delay period. It may be that temporal information conveyed by either PT (WS and/or NS) or IT (NS) neurons is sufficient for controlling the timing of the licking response, such that inactivation of one neuron type (IT or PT) is insufficient to alter timing-related behavior in our behavioral tasks. It is also possible that the mPFC plays a more important role in controlling perceptual-timing[58,59] than motor-timing behavior. These remain unclear.

We unexpectedly identified NS IT and NS PT neurons that do not express PV. The results we obtained from optically-tagged NS and WS neurons were similar to one another with regard to the

neural decoding of target. They were different from one another, however, with regard to the neural decoding of time; both NS IT and NS PT neurons convey temporal information comparable to that conveyed by WS PT neurons. In fact, NS IT neurons convey slightly, but significantly more temporal information than NS PT and WS PT neurons. This suggests optically-tagged NS and WS neurons represent functionally distinct populations. In this regard, we previously showed that the NS neuronal population conveys quantitative value signals in the mPFC, while optically-tagged PV neurons convey only valence signals, but not value signals[56]. This indicates the existence of NS neurons that are functionally distinct from PV neurons in the mPFC. These may correspond to the optically-tagged NS neurons in the present study. It is possible that these optically-tagged NS neurons in the mPFC represent long-range GABAergic neurons[60,61]. But other studies have also identified PV-positive long-range neurons in the mPFC[62,63]. Thus, the exact identity of PV-negative NS neurons must still be clarified. Our results show that NS IT neurons convey both working-memory and temporal information during the delay period. We still need to confirm whether these represent a specific subset of projection neurons that transmit combined working-memory and temporal information. If they do, we also need to identify the downstream targets of their projections.

The PFC is implicated in diverse neuropsychiatric disorders, such as schizophrenia, attention deficit hyperactivity disorder, obsessive-compulsive disorder, and depression[37,64,65], all of which are associated with impaired working memory and/or interval timing[66–69]. This raises the possibility that these neuropsychiatric disorders are associated with different patterns of abnormalities in IT versus PT neurons. Thus, a better understanding of how IT and PT neuronal abnormalities in the PFC are associated with the symptoms of various neuropsychiatric disorders should provide important insights into the etiologies of these diseases.

## Methods

**Subjects**. Rxfp3-Cre (MMRRC 036667, Mutant Mouse Resource and Research Center, CA, USA; $n = 37$) and Efr3a-Cre knock-in (MMRRC 036660; $n = 32$) mice were used to target deep-layer IT and PT neurons, respectively, in the mPFC. Some mice (14 Rxfp3-Cre and 12 Efr3a-Cre) were used for neurophysiological recordings, some (12 Rxfp3-Cre and 11 Efr3a-Cre) for optogenetic inactivation, some (four Rxfp3-Cre) for calcium imaging, some (two Rxfp3-Cre and four Efr3a-Cre) for the verification of IT/PT neurons (Supplementary Fig. 3), and the rest (five Rxfp3-Cre and five Efr3a-Cre) were crossed with Ai9 tdTomato reporter line (JAX 007909, Jackson Laboratory) to examine the overlap between Cre and PV expressions (Fig. 2f, g). Additional six WT mice (C57BL/6 J, JAX000664, Jackson Laboratory) were used in the muscimol-infusion experiment. The animals were individually housed after surgery under a 12 h light/dark cycle (temperature, 19–23 °C; humidity, 45–55%). After 2–3 days of handling, the animals were water-restricted and allowed to drink water only during the task. All animal care and experimental procedures were performed in accordance with protocols approved by the directives of the Animal Care and Use Committee of Korea Advanced Institute of Science and Technology (approval number KA2018-08).

**Working memory task**. The details of the working memory task are described in our previous study[14]. The animals were trained in a delayed match-to-sample task under head fixation (Fig. 1). Each trial comprised sample, delay, and choice phases. Two water lick ports, located on the left and right sides of the animal, were in the retracted position before trial onset. Each trial began (sample phase onset) when a green LED was turned on above a randomly chosen lick port (sample) and the lit lick port was advanced to the animal delivering a drop of water (1.5 μl). After 2 s since LED onset, the LED light was turned off (onset of the delay period) and the sample lick port was retracted. The duration of the delay period was either fixed (4, 15, or 25 s) or variable (1–7 s; uniform random distribution). A brief (100 ms) buzzer sound was delivered at the end of the delay period (onset of the choice phase) and then both water lick ports were advanced to the animal. If the animal licked the lick port that was presented during the sample phase (target lick port), two drops of water (1.5 μl each) were delivered at the chosen water lick port. Then, both water lick ports were maintained at their advanced positions for 7 s before being retracted (correct trial). Both lick ports were retracted immediately without water delivery if the animal licked the opposite port (incorrect trial). If the animal did not lick either port within 4 s since the delay offset, both lick ports were

retracted without water delivery (miss trial; $1.8 \pm 5.1\%$ in neurophysiological recording sessions and $3.0 \pm 5.1\%$ in optogenetic inhibition sessions; mean ± SD). A variable ITI (0–10 s, uniform random distribution) was then imposed before the next trial began.

The type (fixed versus variable) and duration of delay varied across four groups of mice. All mice were initially trained with a fixed delay (4 s) until they reached the performance criterion (>70% correct choices for three consecutive daily sessions; 5–10 sessions; 120–160 trials per session). The first group of mice (six WT) were then tested under the fixed-delay (4 s) condition with or without drug infusion into the mPFC (in the order of ACSF infusion, muscimol infusion, and no drug infusion for three daily sessions). Subsequently, the mice were trained to perform two consecutive blocks of fixed-delay (4 s; 60–80 trials per block) and variable-delay (1–7 s; 60–80 trials per block) trials (their order reversed across successive sessions) and, after reaching the performance criterion (70% correct choices under both delay conditions for three consecutive daily sessions), tested with drug infusion (ACSF, muscimol, and no drug infusion for three daily sessions). The second group of mice (14 Rxfp3-Cre and 12 Efr3a-Cre) were subjected to physiological recordings under the fixed-delay (4 s) condition for 2–4 sessions after reaching the performance criterion. Then, during physiological recordings, the mice were trained to perform two consecutive blocks of fixed-delay (4 s; 60–80 trials per block) and variable-delay (1–7 s; 60–80 trials per block) trials (their order reversed across successive sessions). Unit signals recorded during the variable-delay blocks before reaching the performance criterion (1–3 training sessions) were excluded from the analysis. The third group of mice (12 Rxfp3-Cre and 11 Efr3a-Cre), after reaching the performance criterion with the 4-s fixed delay, was subjected to optogenetic inactivation (two sessions, 100–200 trials per session). This procedure (i.e., training until reaching the performance criterion and investigating the effects of optogenetic inactivation for two sessions) was repeated while increasing the delay duration to 15 s and then to 25 s. The fourth group of mice (four Rxfp3-Cre), after reaching the performance criterion with the 4-s fixed delay, was subjected to calcium imaging. The delay duration was increased gradually to 15 s and then to 25 s over 9–20 days while performing calcium imaging (100–120 trials per session).

**Peak procedure**. Each trial began as a green LED light located above the right water lick port was turned on. The lit right lick port was advanced to the animal and delivered a drop of water (1.5 μl), with the left lick port remaining retracted throughout the test. The LED light was turned off 2 s after trial onset to signal interval onset. The water lick port remained in the advanced position and the animal was allowed to lick freely during the interval phase (10 s). Two drops of water (1.5 μl each) were delivered without any external cue at the interval offset. Then, the water lick port was retracted 12 s after the interval offset. The animals were trained for 2–3 days (150–220 trials per session) until a sharp increase in anticipatory licking was observed around the interval offset. The animals were then tested with optogenetic inactivation (in randomly chosen 30% of trials) while reward delivery was omitted in randomly chosen 30% of trials. Three animals that showed low levels of anticipatory licking were excluded from the inactivation experiment.

**Muscimol infusion**. Two cannulas (diameter, 1 mm) were implanted bilaterally targeting the prelimbic cortex (1.84 mm anterior and 0.4 mm lateral to the bregma; 1.2 mm ventral to brain surface) under isoflurane (1.5–2.0% [v/v] in 100% oxygen) anesthesia. After recovery from surgery (3–5 days) and training in the task (phase 1, 4-s fixed delay only, 5–10 days; phase 2, fixed and variable delays, 3–5 days), the animal's performance was tested with ACSF, muscimol, or no drug infusion. ACSF (Tocris Bioscience, Bristol, UK, 0.2 μl) or muscimol (5 mM; Sigma-Aldrich, MO, USA, 0.2 μl) was injected into the prelimbic cortex at a rate of 0.02 μl/min 45 min before behavioral testing. Fluorescein (5 mM; Sigma-Aldrich, MO, USA, 0.2 μl) was injected before sacrifice to visualize drug spread in the brain. Histological examination showed the spread of fluorescein mostly in the prelimbic cortex, with a small amount spreading to the upper infralimbic cortex (Fig. 1c).

**Virus injection**. A small burr hole (diameter, 0.5 mm) was drilled into the skull (1.84 mm anterior and 0.4–0.5 mm lateral to bregma) under isoflurane (1.5–2.0% [v/v] in 100% oxygen) anesthesia, and a bolus of virus (0.5 μl) was injected 1.25–1.75 mm below the brain surface targeting the mPFC at a rate of 0.05 μl/min. For optogenetic tagging of IT and PT neurons, a double-floxed (DIO) Cre-dependent adeno-associated virus (AAV) vector carrying the gene for ChR2 in-frame and fused to eYFP (AAV2/2-EF1a-DIO-hChR2(H134R)-eYFP, UNC Vector Core, NC, USA) was injected unilaterally in the left or right mPFC (counter-balanced across animals). For the verification of IT and PT neurons, Cre-dependent AAV carrying either eYFP or tdTomato (AAV2/Ef1a-DIO-eYFP or AAV2-CAG-FLEX-tdTomato, UNC Vector Core, NC, USA) was injected unilaterally in the right mPFC. For calcium imaging of IT neurons, Cre-dependent AAV carrying calcium indicator jGCaMP7f (pGP-AAV1-syn-FLEX-jGCaMP7f-WPRE, Addgene plasmid #104492, http://n2t.net/addgene:104492) was injected unilaterally in the right mPFC. For optogenetic inactivation of IT or PT neurons, Cre-dependent AAV carrying stGtACR2 (pAAV1-hSyn1-SIO-stGtACR2-

FusionRed, Addgene plasmid # 105677, http://n2t.net/addgene:105677) was injected bilaterally in the mPFC.

**Neurophysiology and optogenetics.** A hyperdrive, containing an optic fiber (core diameter, 200 μm) and eight tetrodes, was implanted unilaterally above the prelimbic cortex (1.84 mm anterior and 0.45 mm lateral to bregma; 1.25 mm ventral to brain surface) immediately after virus injection. The optic fiber was left in the same location, but individual tetrodes were advanced gradually each day (0.05–0.1 mm per day) once unit recording began to record different unit signals across days. For optical tagging of IT or PT neurons, 473 nm laser pulses (5-ms duration at 1 Hz; 0.2, 0.5, and 1 mW; 200 pulses at each intensity; Omicron-Laserage, Rodgau-Dudenhofen, Germany; controlled by Labview, National Instruments, TX, USA) were delivered at the end of each recording session. The majority of neurons were recorded from the prelimbic cortex (numbers of prelimbic and infralimbic cortical neurons, Rxfp3-Cre mice, 39 and three optically-tagged WS, 18 and one optically-tagged NS, 596 and 98 untagged WS, and 55 and 22 untagged NS neurons, respectively; Efr3a-Cre, 46 and one optically-tagged WS, 11 and four optically-tagged NS, 497 and 94 untagged WS, and 30 and 23 untagged NS neurons, respectively) as judged by the locations of tetrode marking lesions and tetrode advancement histories. Unit signals and LFPs were amplified (10,000× and 1000×, respectively), band-pass filtered (600–6000 Hz and 0.1–9000 Hz, respectively), and digitized (32 and 1 kHz, respectively) using the Cheetah data acquisition system (Neuralynx, MT, USA).

For optical inactivation of IT and PT neurons, two optic fibers were implanted above the prelimbic cortex. A continuous 473 nm laser stimulation (9 mW) was delivered in a randomly chosen subset of trials (30% of a total of 100–200 trials) throughout the delay period (4, 15, or 25 s) in the delayed match-to-sample task and throughout the interval phase (10 s) in the peak procedure.

**Calcium imaging.** A gradient refractive index (GRIN) lens with prism (diameter, 1 mm, length, 4.3 mm, 1 mm × 1 mm prism attached, 1050-004601, Inscopix, CA, USA) was implanted ~0.1 mm lateral to layer 5a of the prelimbic cortex (1.84 mm anterior and 0.5–0.55 mm lateral to bregma; 2 mm ventral to brain surface; targeting IT neurons located at 1.34–2.31 mm anterior to bregma and 1–2 mm ventral to brain surface). The baseplate for nVista 2.0 microscope (Inscopix, CA, USA) was attached to the skull 4–6 weeks since lens implantation. Calcium signals were acquired through the implanted GRIN lens and nVista microscope at 30 Hz frame rate using LED power of 0.42–0.54 mW/mm². Spatial downsampling (× 1/4) and motion correction of calcium imaging data were performed using Inscopix Data Processing Software (version 1.3.1). The processed video was exported in TIFF format and analyzed with the CNMF-E algorithm[70] to extract single unit signals. Details of lens implantation and image processing are described in a previous report[71].

**Histology.** Coronal sections (40 μm) of the brains were prepared according to a standard histological procedure and the locations of tetrode marking lesions (produced by 20-mA, 15-s electrical current), electrode tracks, cannula tracks, optic fiber tracks, and/or GRIN lens tracks were determined by examining images obtained (10x) with a Zeiss Axio Scan.Z1 slide scanner (Zeiss, Jena, Germany). To examine the overlap between PV neurons and optically tagged NS neurons, Rxfp3- and Efr3a-Cre mice were crossed with the Ai9 tdTomato reporter line (JAX 007909, Jackson Laboratory) and PV neurons were immunostained according to the following procedure. Brain slices were incubated with a blocking solution (5% goat serum in PBS with 0.2% Triton-X 100) for 1 h at room temperature and then with primary antibodies (anti-PV primary antibody, ab11427, Abcam, Cambridge, UK; 1:1000) diluted in the blocking solution overnight at 4 °C. The slices were rinsed with the blocking solution and incubated with secondary antibodies (FITC-conjugated anti-rabbit IgG, AP132F, Millipore, Darmstadt, Germany; 1:200) for 2 h at room temperature. The slices were then washed three times with PBS and mounted with 4′,6-diamidino-2-phenylindole (DAPI)-containing Vectashield (Vector Laboratories, CA, USA). Cre- and PV-positive cells were counted manually from the brain sections containing the mPFC. Cell counting was done from three regions of interest (500 μm × 500 μm) showing high levels of Cre expression, each from a different brain section, for each animal. To examine laminar distributions of Rxfp3-Cre and Efr3a-Cre cells, fluorescence-expressing cells were counted from the prelimbic cortex of the mice used for electrophysiological recordings (expressing hChR2-eYFP; one Rxfp3-Cre and two Efr3a-Cre mice) and those used for the verification of IT/PT neurons (expressing eYFP or tdTomato; two Rxfp3-Cre and one Efr3a-Cre mice). Cell counting was done in three separate brain sections from each animal. Laminar locations of fluorescence-expressing cells were determined based on the distance from the midline.

**Isolation and classification of units.** Putative single units were isolated by manually clustering various spike-waveform parameters using the MClust Software (version 3.5B.03, http://redishlab.neuroscience.umn.edu/MClust/MClust.html). Only units with an L-ratio <0.1 (0.03 ± 0.03; mean ± SD; $n = 1573$), an isolation distance >19 (47.1 ± 32.3; mean ± SD), and an inter-spike interval <2 ms were included in the analysis[72].

The isolated units were classified into two groups based on three physiological parameters (mean firing rate during task performance and peak-valley ratio and half-valley width of the averaged spike waveform) using a Gaussian mixture model[55]. The units were grouped into two clusters in the three-dimensional parameter space (Fig. 2c) assuming they were generated from a mixture of two Gaussian distributions. The parameters for Gaussian distributions were determined using the "fitgmdist" function of MATLAB (Mathworks Inc., Natick, MA, USA), which finds maximum likelihood estimates of parameters using the expectation-maximization algorithm. The mean (±SD) firing rate, peak-valley ratio, and half-valley width were 1.6 ± 2.3 Hz, 2.3 ± 0.5, and 372.7 ± 57.4 μs, respectively, for one unit cluster ($n = 1374$), and 7.3 ± 7.4 Hz, 1.1 ± 0.2, and 216.9 ± 39.0 μs, respectively, for the other cluster ($n = 164$). They are referred to as WS and NS neurons, respectively, in the present study. Only those neurons with mean firing rates ≥0.5 Hz during task performance and recorded in the sessions with ≥10 correct trials for both targets were included in the analysis (WS IT, 26 out of 42; WS PT, 31 out of 47; NS IT, 19 out of 19; NS PT, 14 out of 15; untagged WS, 758 out of 1285; See Supplementary Table 1).

**Identification of IT and PT neurons.** Those units that met the following two criteria were considered to be optically tagged neurons. First, the latency of the first spike during the 5-ms window after laser stimulation onset should be significantly ($p < 0.01$) shorter than that in the absence of laser stimulation (a 5-ms window chosen randomly from the 900-ms time period before laser stimulation onset) as determined by both the log-rank test and the stimulus-associated spike latency test (SALT)[73]. Second, the correlation between laser-driven and spontaneous waveforms should be ≥0.8. The peak response latency was the first peak of a peri-stimulus time histogram since optical stimulation.

**Neural decoding of target.** We combined tetrode spike data across different sessions and predicted the target (the lick port presented during the sample phase) based on delay-period neuronal ensemble activity using the support vector machine (SVM; "fitcsvm" function of MATLAB using the linear kernel, Mathworks Inc., Natick, MA, USA). We randomly selected a given number of neurons for a given ensemble size and, for each selected neuron, randomly selected ten correct left-sample and ten correct right-sample trials from the fixed-delay (4 s) block (out of 10–84 correct left-sample and 11–88 correct right-sample trials). Then a single trial was removed, and sample lick port in that trial (target) was predicted based on neuronal ensemble activity in that trial (test trial) and neuronal ensemble activity in the remaining trials (training trials) separated by target (leave-one-out cross validation). This procedure was repeated 100 times. For neural decoding of target in the variable-delay block, we used only those trials with delay durations ≥4 s and randomly selected ten correct trials for both targets. For cross-temporal decoding[18,19], we trained the classifier with ensemble activity in a specific time bin (1-s window) and tested with neuronal ensemble activity in all time bins spanning 1 s before and after the delay period (1-s window advanced in 0.1-s steps). For neural decoding of target in error trials, we used untagged WS neurons recorded in the sessions with ≥2 error trials for both left and right targets ($n = 365$ untagged WS neurons from 117 sessions). We trained the classifier with ensemble activity in each time bin (1-s bin advanced in 0.1-s steps) in correct trials and tested with neuronal ensemble activity in correct as well as error trials. For the comparison of target decoding in error versus miss trials, those sessions with ≥2 error and ≥2 miss trials for both targets were used ($n = 82$ untagged WS neurons from 22 sessions). For neural decoding of context (fixed- versus variable-delay blocks), we randomly selected 20 correct trials regardless of sample target from each context and predicted the context in the same manner (leave-one-out cross validation).

For neural decoding of target using calcium imaging data, ΔF/F was z-scored relative to trial-by-trial ΔF/F values during the baseline period (1-s period before sample onset). Since a large number of cells were imaged in each session (95–451, 247.8 ± 106.2, mean ± SD), instead of creating a pseudo-ensemble by pooling cells across sessions, we performed neural decoding separately for each individual session using only simultaneously recorded neurons. This allowed to use all trials from each session for neural decoding. We used the session with the lowest choice bias between the left and right targets for each delay length (4, 15, and 25 s) for each animal. Again, we trained the SVM classifier only with correct trials, and tested with correct as well as error trials using the same classifier. Per iteration, we randomly selected 76 (80% of the minimum ensemble size, $n = 95$) neurons and performed neural decoding of target using all correct trials (leave-one-out cross validation) as well as all error trials. Decoding performance was averaged across animals and this procedure was repeated 100 times for each delay duration.

**Neural decoding of time.** Timing-related delay-period neural activity was assessed by decoding elapsed time based on neuronal ensemble activity using the SVM. We excluded the first 0.5 s of the delay period from this analysis to exclude sensory response-related neural activity. We combined spike data recorded from different sessions, randomly selected a set of neurons for a given ensemble size, and randomly selected 20 correct trials regardless of sample target for each neuron from the fixed-delay (4 s) block. We divided the delay period (0.5–4 s; total 3.5 s) into ten equal duration bins, trained the SVM to predict the bin number (rather than target), decoded the bin number using a leave-one-out cross validation procedure,

and peak-normalized the outcome (decoded bin counts) for each training bin to generate a normalized decoding probability map[26]. The decoding performance was estimated in two different ways. First, we calculated Pearson's correlation between the actual and predicted bin numbers. Second, we calculated the decoding accuracy —the difference in decoding error (the absolute difference between actual and predicted bin numbers) between temporal bin-shuffled data (i.e., the order of ten temporal bins were randomly shuffled) and the original data. Decoding accuracy measures the degree to which temporal prediction is improved (reduced decoding error) by using the original instead of the temporally shuffled neural data. This procedure was repeated 100 times (using a randomly selected set of neurons and a randomly selected set of 20 correct trials regardless of sample target for each neuron per iteration). For temporal decoding in the variable-delay block, we used only those trials with delay durations ≥4 s.

**LFP power spectrogram**. The sessions with ≥1% of LFP signals reaching the ceiling were excluded from the analysis. A notch filter (60 Hz) was applied to the LFP signals, and the resulting signals were down-sampled to 1 kHz and then band-pass filtered (0.5–100 Hz). Those trials in which SD of LFP signals exceeded 3 SD above the mean calculated across trials were excluded from the analysis. The power spectrogram of LFP signals was calculated using the "mtspecgramc" function in the Chronux toolbox of MATLAB (1-s window, 0.1-s steps, 3-Hz bandwidth with five tapers was used for multi-taper spectrograms).

*Spike-field coherence*. The spike-field coherogram was generated using the "coh-gramc" function in the Chronux toolbox (version 2.12 v02) of MATLAB (1-s window, 0.1-s steps, 3-Hz bandwidth with five tapers) as the following:

$$C_{\text{spike,LFP}} = \frac{\left| S_{\text{spike,LFP}} \right|}{\sqrt{S_{\text{spike}} S_{\text{LFP}}}} \quad (1)$$

where $S_{\text{spike, LFP}}$ is the cross-spectrum of spikes and LFP, representing the correlation between the two signals in the frequency domain. $S_{\text{spike}}$ and $S_{\text{LFP}}$ are power spectrums of spikes and LFPs, respectively. For each neuron, the trial-shifted coherogram was subtracted from the raw coherogram to remove any spurious covariation such as stimulus-induced coherence[74]. This procedure was repeated for the LFP signals recorded from all tetrodes other than the one detecting a given unit signal, and the resulting coherograms were averaged across tetrodes.

**Demixed principal component analysis (dPCA)**. We performed a dPCA[25] to dissociate neural activities related to working memory and temporal information. A spike density function (σ = 200 ms) spanning the time window 3.5 before and 8 s after delay onset was generated for each trial type (ipsilateral- and contralateral-sample trials) for each neuron using correct trials. Then, a three-dimensional matrix of neuronal population activity (neuron × time × trial type) was constructed. The matrix was used to demix target-dependent and target-independent variances and to obtain PCs as previously described[25]. The first five dPCs explained 86.1, 88.4, and 78.2% of the total variance for IT, PT, and untagged WS population activities, respectively. Spike density functions of ipsilateral- and contralateral-target error trials were projected onto the dPC decoder axis when comparing correct versus error trial neural activities (Fig. 6d). To assess the contributions from individual dPCs to the representation of temporal information during the delay period, we compared neural decoding of the elapse of delay (see Neural decoding of time) by projecting neuronal ensemble activities to the first 20 dPCs with and without a specific dPC. The contributions from individual target-dependent dPCs to working memory were similarly assessed (see Neural decoding of target).

**Neuronal response indices**. Target-dependent firing of single neurons was assessed by calculating the target selectivity index as the following:

$$\text{Target selectivity index} = (\text{mean(Ipsi)} - \text{mean(Contra)})/\sqrt{\text{sd(Ipsi)}^2 + \text{sd(Contra)}^2} \quad (2)$$

where Ipsi and Contra denote spike counts in the ipsilateral- and contralateral-target trials, respectively, during a given analysis time window, and mean and sd indicate the mean and SD across trials, respectively. The absolute value of the target selectivity index was used as a measure for target information carried by delay-period activity of individual neurons.

Neural activity differentiating between the sample and choice phases was assessed by calculating the phase selectivity index as the following:

$$\text{Phase selectivity index} = \text{mean(Sample)} - \text{mean(Choice)}/\sqrt{\text{sd(Sample)}^2 + \text{sd(Choice)}^2} \quad (3)$$

where Sample and Choice denote spike counts during the sample and choice periods, respectively, regardless of the target of a given trial. Here, the sample and choice periods were 1-s time windows since the first sample and choice licks, respectively. We aligned the analysis time windows to the first sample/choice lick instead of delay onset/offset in this analysis because some animals showed slightly different latencies in the first lick between left-target and right-target trials.

Trial outcome-dependent neural activity was assessed by calculating the outcome selectivity index as the following:

$$\text{Outcome selectivity index} = (\text{mean(Correct)} - \text{mean(Error)})/\sqrt{\text{sd(Correct)}^2 + \text{sd(Error)}^2} \quad (4)$$

where Correct and Error denote spike counts during the 1-s time window after the first choice lick in correct and error trials, respectively.

Neuronal responses to LED, buzzer, and ITI onsets were assessed by calculating an event modulation index for each of these events as the following:

$$\text{Modulation index} = (\text{mean(After)} - \text{mean(Before)})/\sqrt{\text{sd(After)}^2 + \text{sd(Before)}^2} \quad (5)$$

where After and Before denote spike counts during 0.5-s time windows before and after LED, buzzer, or ITI onset.

**Licking versus firing rate correlation**. We calculated Pearson's correlation between firing rate and licking rate for each unit. The entire task period in the fixed-delay condition was divided into 1-s bins, and then firing and licking rates for each bin were calculated. Lick-responsive units were defined as those with a significant correlation ($p < 0.05$) between firing rate and licking rate.

**Statistical tests**. Results are presented as mean ± SEM unless noted otherwise. All statistical analyses were performed with MATLAB (version 2017a). Student's $t$-test, one-way repeated measures ANOVA, two-way ANOVA, two-way repeated measures ANOVA, two-way mixed ANOVA, Bonferroni's post hoc test, Friedman test, and Fisher's exact test were used for group comparison. All tests were two-sided and a $p$ value <0.05 was used as the criterion for a significant statistical difference unless otherwise specified.

**Reporting Summary**. Further information on research design is available in the Nature Research Reporting Summary linked to this article.

## Data availability
The neural and behavioral datasets used in this study are available at: https://www.dropbox.com/sh/y520iwmg8kly1w7/AAAq6umIPZIIJgJmuzK_kZ1Fa?dl=0. Source data for figures are provided with this paper. Source data are provided with this paper.

## Code availability
Code is available in the public Github repository: https://github.com/Jungwon17/ncomms_BAE_2021.git

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

## Acknowledgements

This work was supported by the Research Center Program of the Institute for Basic Science (IBS-R002-A1; M.W.J.) and the National Research Foundation of Korea (NRF) grant funded by the Korean government (MSIT) (NRF-2019R1A2C4069863 to S.P.).

## Author contributions

J.W.B. and M.W.J. designed the study. J.W.B. and H.J. collected and analyzed experimental data. Y.J.Y. collected calcium imaging data. C.M.B. performed histology. H.L. and S.-B.P. analyzed spike-field coherence. J.W.B., H.J., and M.W.J. wrote the manuscript with inputs from all authors. M.W.J. supervised all aspects of the study.

## Competing interests

The authors declare no competing interests.
