## [Peer Review File · Nature Communications]

REVIEWER COMMENTS

Reviewer #1 (Remarks to the Author):

In the current study, the authors examined the discharge characteristics and inactivation effects of intratelencephalic (IT) and pyramidal tract (PT) neurons in the medial prefrontal cortex of mice performing a delayed response task. They demonstrated that IT neurons, but not PT neurons, convey significant working memory-related signals. Inactivation of IT neurons, but not PT neurons, impairs behavioral performance. In contrast, PT neurons convey more temporal information than IT neurons during the delay period. The current study nicely revealed a division of labor between IT and PT projection neurons in the prefrontal cortex for the maintenance of working memory and for tracking the passage of time, respectively. However, there are several concerns need to be resolved and more experiments and analysis need to be performed before the acceptance for publication.

Major concerns:

1. IT and PT neurons exhibited different spatial patterns: IT neurons are distributed across all layers of the cortex and PT neurons are in layer 5. Is the difference observed in coding due to the difference in laminar distribution? The optogenetic tagging results depend on the specificity in laminar distribution of Rxfp3-Cre and Efr3a-Cre positive neurons. However, the size and resolution of Figure 2a and Suppl. Fig. 2a are not good enough for clear verification of layer-5 specific distribution. The authors should quantitatively examine the laminar distribution of Rxfp3-Cre and Efr3a-Cre labeled neurons and show better images clearly demonstrating the laminar specificity of the corresponding mice line.
2. The authors only analyzed the correct trials. Is there any difference in coding between correct and error trials? Such results will be important to show the behavioral relevance respective coding of IT and PT neurons in working memory. Similarly, is there any difference in spike-field coherence of IT and PT neurons in correct vs. error trials? How about FS-IT and FS-PT neurons?
3. How to distinguish between timing encoding versus prediction of reward? The authors should design experiments to distinguish between these two functions, for example by using timing behavior without reward expectation or peak-interval procedure.
4. Optogenetic inactivation does not impair performance for 4-sec delay condition and only impair performance for 15-25 sec delay (Fig. 6b). What is the coding ability of IT neurons for 15-25 sec delay condition? The authors should perform experiments to demonstrate it.
5. Different working memory design may require different neural elements. PT neurons might well encode working memory information in other behavioral designs. Therefore, the authors should be cautious about the generality of their conclusion and restrict their conclusion in their specific design.

Minor concerns:

6. Page 9, line 301, "optical stimulation" is better replaced with "optical inactivation"

Reviewer #2 (Remarks to the Author):

This paper examines the role of intertelencephalic and pyramidal tract neurons in the mPFC during a WM task. The results reveal that IT but not PT neurons encode the sample stimulus during the delay period, thus indicating differential computational contributions of different neuron types in the same cortical layer during the same task. The authors go on to show that IT, but not PT, inhibition significantly affects task performance. While IT neurons coded for information related to WM, PT neurons (but not IT) encoded elapsed time during the delay period. Overall this is a very nice study with clear and robust results, and provides fascinating evidence of a sharp computational specialization between different types of pyramidal neurons in the same task. The results also provide important insights into both WM and timing, and remind us of the potential pitfalls of interpreting electrophysiological results without knowing the neuronal subtypes being

recorded from.

All my concerns are minor.

Do the PT and/or IT responses change between the 4 s and 1-7 s delay blocks (as I understood it some of these blocks were on the same day)? In other words, if the IT neurons encode the sample one might expect these units not to change. In contrast, if the PT units are encoding time, one might expect their firing patterns to change during the variable delay block as it is no longer possible to anticipate the delay. The authors should consider adding an additional supplemental figure contrasting the PTSH of some neurons during both blocks, as well as a direct contrast of the decoding of the first 4 s. In other words, when only using the same set and number of neurons is there a statistically significant difference in the decoding of 4 s between Figs 5C and S4E?

A statement should be made about whether the authors believe it is possible that some of the differences between IT and PT inhibition could be attributed to the number of cells stimulated (it seems there are more labeled IT neurons).

State in the Methods whether the SVM kernel was linear or nonlinear (if so, what was it).

Li 418. Saying the mPFC is "critical" for interval timing is a bit of an overstatement (indeed the current paper suggests it may not be, as the PT units did not affect the peak interval performance). Perhaps it would be better to say it has been implicated in interval timing.

Reviewer #3 (Remarks to the Author):

I read the manuscript by Bae et al with great interest, and agree with the authors that the functional differentiation of IT vs PT neurons in the PFC is poorly investigated. In their study, Bae et al., take advantage of available knock-in mouse lines¹ to target IT and PT neurons of the mouse medial prefrontal cortex (mPFC). The authors make use of a delayed response task for head-fixed mice they have used previously², to investigate the roles prefrontal cortical IT and PT neurons play in working memory (WM). Tetrode recordings and optogenetics are applied, in combination for optotagging of IT and PT neurons, and optogenetics alone for broad silencing of prefrontal IT or PT neurons. Based on the experiments, the authors conclude that the delay-period activity of IT neurons conveys working memory-related signals with limited temporal information, while delay-period activity of PT neurons conveys precise information about the elapsed delay period in the form of ramping activity with minimal working memory-related signal.

Major concerns:

My major concern regards how WM is approached and conceived; in short, I don't find probing of WM convincingly demonstrated. This is partly due to the fact that too little behavioral data is presented (including the neuronal correlates). The concept of WM is vague in the current ms, and the author's definition of WM and what is specifically investigated unclear: "processing of WM", "maintenance of WM", and "WM related signals" are mentioned eg. At one place the authors write "sample identity (i.e., working memory)". The present ms gives the impression that WM in essence is spiking that happens during a task delay (any lengths) and that correlates to a stimulus location, which is a too simplistic view. It is generally agreed that WM is a multicomponent system that manipulates information storage for greater and more complex cognitive utility. WM is also considered to have a capacity limit. It would be relevant to know how eg how the authors separate WM from other types of memory, particularly short-term memory (experimentally and/or conceptually). The methods section gives away that several different fixed delay lengths were used, as well as random delays (the authors use, and present data from, different delay protocols for different experiments, unclear why). The beh data also includes different types of error trials, but the data for this, including the neuronal activity, is at large not presented. I wonder, how does eg the delay length affect behavior and neuronal activity, and how does the behavioral dataset give support for probing of WM specifically? Related, the authors recorded an impressive number of single units, but much of the recording data is not presented, particularly not the variability in

the activities, and it is not clear why and how different sets and numbers of neurons are used. In essence, the authors put in a lot of work in the behavior and recordings but refrain from presenting some key data, data that could possibly allow investigation of prefrontal processing involved in WM. The unclear selection of data and analysis together with some conceptual imprecisions leave the impression of a patchy story, and unfortunately also give the sense of cherry-picking (although I find that the eventual cherry-picking more than anything devalues any potential findings).

Detailed comments:

Neurons in frontal cortical areas have been shown to be tuned to multiple task variables (eg sensory and motor variables)^{4,5} and mixed selectivity is also prominent in prefrontal regions. Despite the sophistication of the task design used, eg Figure 3a only shows 4 example units(?), in correct trials specifically. It would be appropriate for the author to present/represent the full dataset in a main figure, and also population analysis. The study would be greatly strengthened by presentation of the richness of the dataset and of how identified prefrontal subpopulations align to the different task variables (eg light, sound, licking). As mentioned, accounting for activities in error trials and in relation to delay lengths is also of importance. Presenting the whole data set would also aid the readers' interpretation of how representative the optotagged activities are. Additionally, videos of representative behavior in the DMS task would be a great addition to the manuscript, and allow the reader to qualitatively assess the task used and the robustness and details of the behavior.

In Figure 5, the authors find with a demixed principal component analysis (dPCA) that PT neurons encode the passage of time, and display ramping activity during the delay. As the authors are aware, PT and IT subpopulations have been studied during motor planning in a similar head-fixed paradigm (the sample epoch is different in this paradigm - the animals have to assess the position of a pole presented in front of their whiskers), in a different prefrontal subregion (ALM)³. In this study, PT neuron activity was shown to ramp at the end of the delay preceding contralateral (to the recorded hemisphere) licking movements. Motor planning has been defined as "internal processes by which the brain plans and executes volitional movements"⁶ and the ramping activity during the delay epoch before the onset of movement has been shown to represent motor-related information^{3,7}. In light of these findings, it is important that the authors assess possible PT ramping activity before contralateral licking movements, as this has bearing on their interpretations. Further, to support a claim that "PT neurons precisely encode the passage of time", experiments showing how the ramping activity reflects temporal variations (eg different delay durations) are warranted. These are only suggestive starting points for the interpretation that PT neurons convey time-related information.

In their photoinhibition experiments, the authors find no effect on beh performance with optogenetic inhibition of PT and IT neurons with use of 4 s delay – inhibition of IT neurons deteriorated behavior with use of longer delays (15 s and 25 s). All the prior results in the study are based on a 4 s delay and eg the electrophysiological data for 15 s and 25 s delay appears to be missing from the manuscript. It is thus unclear if the findings obtained by use of a 4 s delay hold true for longer delays, and overall, unified findings based on ALL the generated data are warranted. In my view, refraining from showing (and/or analyzing) the whole dataset severely hampers the quality and interpretation of the study. The authors also need to reconcile their findings with findings in two recently published studies on WM in frontal regions in mice^{8,9}.

The absence of parvalbumin expression in Rxfp3-Cre and Efr3a-Cre positive neurons and the physiological identification of putative fast-spiking (FS) opto-tagged units in both Rxfp3-Cre and Efr3a-Cre is a contradictory finding requiring clarification. The legend of Figure 2d indicates that immunostainings were performed in two animals (1 per genotype). These histological experiments should be repeated in more animals with statistically sound methods to ensure their robustness. In the case of a lack between Cre and PV positive neuronal populations, the authors need to clarify if the presence of FS opto-tagged units could be related to indirect recruitment of FS neurons upon light stimulation. Details of the light-response latencies for the 4 neuronal categories (PP-IT, PP-PT, FS-IT and FS-PT) as well as cross-correlograms between all opto-tagged units would help to elucidate if FS units have higher latencies than PP units and if PP and FS can have monosynaptic functional connections explaining indirect recruitment of FS units (PP->FS pairs with a positive

peak lag). I strongly recommend that the authors carry out these additional experiments/analysis.

In the peak procedure behavioral task, the authors conclude that "This indicates that neither IT nor PT neurons are indispensable for controlling the timing of the licking response in the present peak procedure" (p11). This statement appears to contradict the previously claimed result: "PT neurons precisely encode the passage of time". Do the authors mean that PT neurons encode the passage of time in one behavioral paradigm but not in another, or that prefrontal PT neurons encode the passage of time but this information is not used for timing in behavior (eg licking behavior)?

I, unfortunately, find the spike-LFP part weak, and question how this part supports the findings in the study. Important findings on spike-LFP theta synchrony in HPC-PFC interactions in spatial WM have been published, and while the authors refer to this work, their own probing of LFP activities in WM is shallow. The mPFC theta component during WM in the task used is not really investigated, but rather the analysis is solely a comparison of spike-theta coherence between IT and PT neurons specifically during correct trials with 4 s delays. I do not find that the quite limited analysis supports the suggestion that "IT neurons may contribute to the maintenance of working memory". Further, I have problems envisioning the mechanism the authors imply in the last part of the sentence: "...IT neurons may contribute to the maintenance of working memory, not only via sample-dependent firing, but also by enhancing synchronous firing at theta frequencies". Whose synchronous firing are the IT neurons enhancing here? I apologize if I am misunderstanding what is meant here. Fig 2b also implies that IT neurons are more strongly coupled to a broad LFP spectra than PT neurons (albeit not significantly), which is an interesting finding not pursued. Of note, most of the previous work investigated spatial WM; the current study did not; despite this the authors interpret their results based on findings in spatial WM task and refrain from discussing the validity of doing so, which I find troublesome.

Specific points:

The study needs clarification throughout the text and figures regarding what neurons are included in different analyses/panels (and why), not just a statement of the n of neurons. It is hard to grasp what data regards the same neurons, and not, across the study.

I strongly recommend stating in the (all) figure panels what data is presented, eg what colors, line weight etc represents. This would greatly improve the accessibility of the data.

I find the use of the term 'sample identity' being questionable for the task design used. The spatial location of the sample lick spout appears to be what is to be remembered(?)

I find the use of refs 40-42 as support for the statement "...PT neurons have been proposed to play a more important role in the maintenance of working memory than IT neurons40-42" questionable.

In line with the 2 last comments, I recommend the authors to revise the ms to optimally reflect published findings and concepts.

While the manuscript is well written, some statements are hard to grasp eg (p2) "In the primary motor cortex, IT neurons represent the deviation between current and subsequent behaviors, while PT neurons specifically encode the direction and speed of a subsequent behavioral response6".

Most of the methodological section provide sufficient detail. However, details regarding the training of the mice (number of days, evolution of the performance, representative video recordings of the task etc) are needed to support any concepts and conclusions.

Summary

The authors hold relevant and significant datasets, with potential novelty. However, both the behavioral and electrophysiological data need a more complete analysis and presentation. I recommend the authors to base their interpretations and claims on the whole dataset, and to

potentially reevaluate their concept of WM and the role of mPFC IT and PT neurons in the light of their extended findings. I am convinced the datasets hold important information on prefrontal processing in goal-oriented behavior, and particularly the involvement of PT and IT neurons, respectively, but I find that the authors need to let the (whole) data guide what conclusions can be drawn.

References

1. Gerfen, C. R., Paletzki, R. & Heintz, N. GENSAT BAC Cre-Recombinase Driver Lines to Study the Functional Organization of Cerebral Cortical and Basal Ganglia Circuits. *Neuron* 80, 1368–1383 (2013).
2. Park, J. C., Bae, J. W., Kim, J. & Jung, M. W. Dynamically changing neuronal activity supporting working memory for predictable and unpredictable durations. *Sci. Rep.* 9, 15512 (2019).
3. Li, N., Chen, T.-W., Guo, Z. V., Gerfen, C. R. & Svoboda, K. A motor cortex circuit for motor planning and movement. *Nature* 519, 51–56 (2015).
4. Chen, T.-W., Li, N., Daie, K. & Svoboda, K. A Map of Anticipatory Activity in Mouse Motor Cortex. *Neuron* 94, 866-879.e4 (2017).
5. Pinto, L. & Dan, Y. Cell-Type-Specific Activity in Prefrontal Cortex during Goal-Directed Behavior. *Neuron* 87, 437–450 (2015).
6. Svoboda, K. & Li, N. Neural mechanisms of movement planning: motor cortex and beyond. *Curr. Opin. Neurobiol.* 49, 33–41 (2018).
7. Inagaki, H. K., Fontolan, L., Romani, S. & Svoboda, K. Discrete attractor dynamics underlies persistent activity in the frontal cortex. *Nature* 566, 212 (2019).
8. Liu, D. et al. Medial prefrontal activity during delay period contributes to learning of a working memory task. *Science* 346, 458–463 (2014).
9. Zhu, J. et al. Transient Delay-Period Activity of Agranular Insular Cortex Controls Working Memory Maintenance in Learning Novel Tasks. *Neuron* 105, 934-946.e5 (2020).

RE: Revision of MS# NCOMMS-20-38699-T

Parallel processing of working memory and temporal information by distinct types of cortical projection neurons

Reviewer #1

In the current study, the authors examined the discharge characteristics and inactivation effects of intratelencephalic (IT) and pyramidal tract (PT) neurons in the medial prefrontal cortex of mice performing a delayed response task. They demonstrated that IT neurons, but not PT neurons, convey significant working memory-related signals. Inactivation of IT neurons, but not PT neurons, impairs behavioral performance. In contrast, PT neurons convey more temporal information than IT neurons during the delay period. The current study nicely revealed a division of labor between IT and PT projection neurons in the prefrontal cortex for the maintenance of working memory and for tracking the passage of time, respectively. However, there are several concerns need to be resolved and more experiments and analysis need to be performed before the acceptance for publication.

Major concerns:

*1. IT and PT neurons exhibited different spatial patterns: IT neurons are distributed across all layers of the cortex and PT neurons are in layer 5. Is the difference observed in coding due to the difference in laminar distribution? The optogenetic tagging results depend on the specificity in laminar distribution of *Rxfp3-Cre* and *Efr3a-Cre* positive neurons. However, the size and resolution of Figure 2a and Suppl. Fig. 2a are not good enough for clear verification of layer-5 specific distribution. The authors should quantitatively examine the laminar distribution of *Rxfp3-Cre* and *Efr3a-Cre* labeled neurons and show better images clearly demonstrating the laminar specificity of the corresponding mice line.*

Response: We used *Rxfp3-Cre* (deep layer IT-specific) and *Efr3a-Cre* (PT-specific) mice to express ChR2 in deep, but not superficial, layers of the mPFC. As expected, fluorescence expressions were largely limited to deep layers of the mPFC. As suggested by the reviewer, we now show quantitative laminar distribution profiles of *Rxfp3-Cre* and *Efr3a-Cre* labeled neurons in Fig. 2b of the revised manuscript. We also replaced the images in Fig. 2a and Supplementary Fig. 3 with new ones to show laminar specificity of *Rxfp3-Cre* and *Efr3a-Cre* labeled neurons more clearly. In addition, the brain histology obtained from *Rxfp3-Cre* or *Efr3a-Cre* mice crossed with a reporter line clearly shows localization of Cre-expressing neurons in deep layers (Fig. 2f; red, Cre-tdTomato).

2. The authors only analyzed the correct trials. Is there any difference in coding between correct and error trials? Such results will be important to show the behavioral relevance respective coding of IT and PT neurons in working memory. Similarly, is there any difference in spike-field

coherence of IT and PT neurons in correct vs. error trials? How about FS-IT and FS-PT neurons?

Response: We focused on delay-period activity during correct trials in the original manuscript because the number of error trials was rather small (8.5 ± 7.6 error trials per session; mean \pm SD). This, in combination with relatively small numbers of IT and PT neurons (16 PP IT, 13 PP PT, 11 FS IT, and 2 FS PT neurons satisfying ≥ 0.5 Hz and ≥ 10 correct and ≥ 2 error trials for each target) and large variability in neural activity across trials, prevented reliable estimation of working memory-related neural activity in error trials. We agree, however, that it is important to compare correct- and error-trial neural activity to assess behavioral relevance of working memory-related neural activity. We therefore performed the following analyses. First, we analyzed error trials of optically-untagged PP neurons ($n = 365$). Second, we performed a new experiment collecting neural data from a large number of IT neurons with calcium imaging. This provides an opportunity to reliably analyze IT neuronal activity during error trials because the number of recorded neurons is large ($n = 95 \sim 451$ per session). The results from these analyses clearly show that working memory-related signals are poorly maintained during the delay period in error trials (Fig. 4 and 10). We also show that neuronal activity differs between different types of incorrect trials, namely error (wrong choice of target) and miss (no choice) trials (Supplementary Fig. 7). Regarding theta coherence, for the same reasons as above, we analyzed untagged PP neurons instead of IT and PT neurons. We found that spike-field coherence in the theta band is lower in error compared to correct trials during the delay period (Fig. 5d-g).

3. How to distinguish between timing encoding versus prediction of reward? The authors should design experiments to distinguish between these two functions, for example by using timing behavior without reward expectation or peak-interval procedure.

Response: As the reviewer indicated, timing-related neural activity during the fixed (4 s) delay may encode the elapse of delay or the time until reward delivery. Please note, however, that we found a largely similar pattern of time encoding by PT neurons in the variable (1-7 s) delay condition (Supplementary Fig. 5g and i). This result favors encoding the elapse of delay over encoding the time until reward because the exact time of reward delivery could not be predicted in the variable delay condition. We now discuss this issue in the revised text (L 495-498).

4. Optogenetic inactivation does not impair performance for 4-sec delay condition and only impair performance for 15-25 sec delay (Fig. 6b). What is the coding ability of IT neurons for 15-25 sec delay condition? The authors should perform experiments to demonstrate it.

Response: We performed calcium imaging experiments to examine working memory-related signals of IT neurons during a long (up to 25 s) delay. We chose to use calcium imaging over tetrode recording because optogenetic identification of mPFC neurons with tetrode recording is time consuming (we expect > 1 year to collect a sufficient number of optogenetically identified IT neurons), and because we can collect a large number IT neurons which enables reliable estimation of working memory-related delay-period activity in error trials (see our response to

the comment above). Results from this new experiment show that IT neuronal population carries significant working memory-related signals throughout the entire delay period (up to 25 s). This is now shown in Fig. 10 of the revised manuscript.

5. Different working memory design may require different neural elements. PT neurons might well encode working memory information in other behavioral designs. Therefore, the authors should be cautious about the generality of their conclusion and restrict their conclusion in their specific design.

Response: We agree. We now mention this caveat in the discussion (L 476-478).

Minor concerns:

6. Page 9, line 301, “optical stimulation” is better replaced with “optical inactivation”

Response: Thanks for pointing this out. It’s been corrected.

Reviewer #2

This paper examines the role of intertelencephalic and pyramidal tract neurons in the mPFC during a WM task. The results reveal that IT but not PT neurons encode the sample stimulus during the delay period, thus indicating differential computational contributions of different neuron types in the same cortical layer during the same task. The authors go on to show that IT, but not PT, inhibition significantly affects task performance. While IT neurons coded for information related to WM, PT neurons (but not IT) encoded elapsed time during the delay period. Overall this is a very nice study with clear and robust results, and provides fascinating evidence of a sharp computational specialization between different types of pyramidal neurons in the same task. The results also provide important insights into both WM and timing, and remind us of the potential pitfalls of interpreting electrophysiological results without knowing the neuronal subtypes being recorded from.

All my concerns are minor.

Do the PT and/or IT responses change between the 4 s and 1-7 s delay blocks (as I understood it some of these blocks were on the same day)? In other words, if the IT neurons encode the sample one might expect these units not to change. In contrast, if the PT units are encoding time, one might expect their firing patterns to change during the variable delay block as it is no longer possible to anticipate the delay. The authors should consider adding an additional supplemental figure contrasting the PTSH of some neurons during both blocks, as well as a direct contrast of the decoding of the first 4 s. In other words, when only using the same set and

number of neurons is there a statistically significant difference in the decoding of 4 s between Figs 5C and S4E?

Response: Thanks for the suggestion. As suggested, we compare IT and PT neuronal activity across the fixed and variable delay conditions and show the results in Supplementary Fig. 5 of the revised manuscript. Please note that we have previously compared delay-period activity across the fixed and variable delay conditions using a large number of untagged neurons. As reported previously for untagged mPFC pyramidal neurons (Park, 2019, Sci Reports), we found that both IT and PT neurons tend to change their activity profiles across the fixed and variable delay conditions, suggesting strong representation of behavioral context by both IT and PT neurons in the mPFC. Nevertheless, we found that working memory-related signals tend to be maintained across the fixed and random delay conditions for IT, but not PT, neurons (Supplementary Fig. 5f). Conversely, temporal information tended to be maintained across the fixed and random delay conditions for PT, but not IT, neurons (Supplementary Fig. 5). Even though the reviewer predicted disappearance of timing-related activity in the variable delay condition, this finding may be interpreted as evidence for PT neuronal encoding of the elapse of delay rather than the time until reward delivery (see our response to comment #3 of the reviewer #1). Our results suggest that the animal might keep track of the elapse of delay regardless of the predictability of delay duration. This may be because of the animal's innate tendency to learn and exploit regularities in the environment (e.g., a large body of studies have shown that subjects keep track of cue-outcome contingencies even when it is totally random) or to evaluate values of actions precisely considering temporal discounting (reward delivered long after an action will be discounted more).

A statement should be made about whether the authors believe it is possible that some of the differences between IT and PT inhibition could be attributed to the number of cells stimulated (it seems there are more labeled IT neurons).

Response: In fact, a close histological examination revealed that the total number of fluorescence-expressing neurons is similar between *Rxrfp3-Cre* and *Efr3a-Cre* mice (Fig. 2b). This argues against the possibility that some of the differences between IT and PT inhibition are because of differential labelling between IT and PT neurons. Nevertheless, we discussed this issue in the discussion (L 458-463) as suggested.

State in the Methods whether the SVM kernel was linear or nonlinear (if so, what was it).

Response: The SVM kernel was linear. This is mentioned in the revised text (L 727).

Li 418. Saying the mPFC is "critical" for interval timing is a bit of an overstatement (indeed the current paper suggests it may not be, as the PT units did not affect the peak interval performance). Perhaps it would be better to say it has been implicated in interval timing.

Response: We agree. We revised the text as suggested (L 491).

Reviewer #3

I read the manuscript by Bae et al with great interest, and agree with the authors that the functional differentiation of IT vs PT neurons in the PFC is poorly investigated. In their study, Bae et al., take advantage of available knock-in mouse lines¹ to target IT and PT neurons of the mouse medial prefrontal cortex (mPFC). The authors make use of a delayed response task for head-fixed mice they have used previously², to investigate the roles prefrontal cortical IT and PT neurons play in working memory (WM). Tetrode recordings and optogenetics are applied, in combination for optotagging of IT and PT neurons, and optogenetics alone for broad silencing of prefrontal IT or PT neurons. Based on the experiments, the authors conclude that the delay-period activity of IT neurons conveys working memory-related signals with limited temporal information, while delay-period activity of PT neurons conveys precise information about the elapsed delay period in the form of ramping activity with minimal working memory-related signal.

Major concerns:

My major concern regards how WM is approached and conceived; in short, I don't find probing of WM convincingly demonstrated. This is partly due to the fact that too little behavioral data is presented (including the neuronal correlates). The concept of WM is vague in the current ms, and the author's definition of WM and what is specifically investigated unclear: "processing of WM", "maintenance of WM", and "WM related signals" are mentioned eg. At one place the authors write "sample identity (i.e., working memory)". The present ms gives the impression that WM in essence is spiking that happens during a task delay (any lengths) and that correlates to a stimulus location, which is a too simplistic view. It is generally agreed that WM is a multicomponent system that manipulates information storage for greater and more complex cognitive utility. WM is also considered to have a capacity limit. It would be relevant to know how eg how the authors separate WM from other types of memory, particularly short-term memory (experimentally and/or conceptually). The methods section gives away that several different fixed delay lengths were used, as well as random delays (the authors use, and present data from, different delay protocols for different experiments, unclear why). The beh data also includes different types of error trials, but the data for this, including the neuronal activity, is at large not presented. I wonder, how does eg the delay length affect behavior and neuronal activity, and how does the behavioral dataset give support for probing of WM specifically? Related, the authors recorded an impressive number of single units, but much of the recording data is not presented, particularly not the variability in the activities, and it is not clear why and how different sets and numbers of neurons are used. In essence, the authors put in a lot of work in the behavior and recordings but refrain from presenting some key data, data that could possibly allow investigation of prefrontal processing involved in WM. The unclear selection of data and analysis together with some conceptual imprecisions leave the impression of a patchy story, and unfortunately also give the sense of cherry-picking (although I find that the eventual

cherry-picking more than anything devalues any potential findings).

Response: We appreciate the reviewer's detailed and constructive comments. In the initial manuscript, given that very little is known about IT and PT neuronal roles in working memory, we tried to focus on IT and PT neuronal contributions to a specific and well-studied neural process (target-related delay-period activity) using a popular behavioral paradigm (delayed response task). In other words, we tried to keep the agenda simple. However, as the reviewer indicated, a more comprehensive presentation of the behavioral and neural data would be helpful to better understand how IT and PT neurons contribute to working memory required for the current delayed response task. To address the reviewer's concerns, we've made substantial changes to the manuscript. Specifically, in the revised manuscript, we used similar sets of neurons across different analyses by applying general criteria for cell inclusion and explain clearly which neurons were included in each analysis (L 146-151 and throughout the text; also summarized in Supplementary Table 1); indicate which aspect of working memory we're interested in ('maintenance of task-relevant information in the absence of external sensory cues', Introduction, L 64-66), replaced 'working memory-related signals' with 'target-related signals' throughout the text where appropriate, and discuss not only information maintenance, but also information manipulation aspects of working memory ('retrospective versus prospective memory'; Discussion, L 448-457); show additional analysis results for behavioral performance (Fig. 1d, e; Supplementary Fig. 2) as well as neural activity during the task (Fig. 3b-d; Fig. 7); explain more clearly the experimental procedures including delay protocols (L 580-603); show the results from the analysis of error-trial (and also miss-trial) neural activity (Fig. 4; Fig. 5b-g; Fig.10; Supplementary Fig. 7; also see our response to the second comment of reviewer #1); and compare behavioral performance and neural activity across the fixed and variable delay conditions (Supplementary Fig. 5). We believe our revised manuscript provides much richer information about the way IT and PT neurons contribute to working memory.

Detailed comments:

Neurons in frontal cortical areas have been shown to be tuned to multiple task variables (eg sensory and motor variables)^{4,5} and mixed selectivity is also prominent in prefrontal regions. Despite the sophistication of the task design used, eg Figure 3a only shows 4 example units(?), in correct trials specifically. It would be appropriate for the author to present/represent the full dataset in a main figure, and also population analysis. The study would be greatly strengthened by presentation of the richness of the dataset and of how identified prefrontal subpopulations align to the different task variables (eg light, sound, licking). As mentioned, accounting for activities in error trials and in relation to delay lengths is also of importance. Presenting the whole data set would also aid the readers' interpretation of how representative the optotagged activities are. Additionally, videos of representative behavior in the DMS task would be a great addition to the manuscript, and allow the reader to qualitatively assess the task used and the robustness and details of the behavior.

Response: We agree that it would be helpful to present behavioral and neural data in a more comprehensive manner. As suggested, we now show the diversity of IT and PT neuronal

responses as a separate figure (Fig. 7). The new figure shows IT and PT neuronal responses to diverse events of the task. We also show summary heat maps for target-dependent activity of all IT and PT neurons that were included in the analysis (Fig. 3b). We also show the results of error-trial analysis (Fig. 4; Fig. 5b-g; Fig.10; Supplementary Fig. 7), compare neural activity across fixed and variable delay conditions (Supplementary Fig. 5), and show delay-period activity during long (15 and 25 s) delays (Fig. 10). Finally, we uploaded a video of representative behavior (Supplementary video 1).

In Figure 5, the authors find with a demixed principal component analysis (dPCA) that PT neurons encode the passage of time, and display ramping activity during the delay. As the authors are aware, PT and IT subpopulations have been studied during motor planning in a similar head-fixed paradigm (the sample epoch is different in this paradigm - the animals have to assess the position of a pole presented in front of their whiskers), in a different prefrontal subregion (ALM)³. In this study, PT neuron activity was shown to ramp at the end of the delay preceding contralateral (to the recorded hemisphere) licking movements. Motor planning has been defined as “internal processes by which the brain plans and executes volitional movements”⁶ and the ramping activity during the delay epoch before the onset of movement has been shown to represent motor-related information^{3,7}. In light of these findings, it is important that the authors assess possible PT ramping activity before contralateral licking movements, as this has bearing on their interpretations. Further, to support a claim that “PT neurons precisely encode the passage of time”, experiments showing how the ramping activity reflects temporal variations (eg different delay durations) are warranted. These are only suggestive starting points for the interpretation that PT neurons convey time-related information.

Response: As the reviewer pointed out, timing-related PT neuronal activity might be related to encoding the passage of time or, alternatively, motor planning. However, as we mentioned in our response to comment #3 of reviewer #1, we found a largely similar pattern of time encoding by PT neurons under the variable (1-7 s) delay condition (Supplementary Fig. 5g and i). This cannot be explained by motor planning because the time of motor response could not be predicted in the variable delay condition. Note that we did find potential motor planning-related PT neuronal activity during the last ~1 s during the delay period (Fig. 3g). However, PT neuronal population did not show biased firing toward contralateral licking responses (Fig. 7k), indicating that it is distinct from motor planning-related activity found in the motor cortex. More importantly, PT neuronal ramping activity spanning the delay period showed similar ramping activity profiles between ipsilateral and contralateral choice trials (“target-independent dPC”, Fig. 6e; activities of sample PT neurons, Fig 3a and Supplementary Fig. 5a) unlike upcoming choice-related PT neuronal activity at the end of the delay period. These results consistently indicate that PT neuronal ramping activity spanning the delay period is related to the passage of time rather than motor planning. We discussed this matter in the revised text (L 326-333).

In their photoinhibition experiments, the authors find no effect on beh performance with optogenetic inhibition of PT and IT neurons with use of 4 s delay – inhibition of IT neurons deteriorated behavior with use of longer delays (15 s and 25 s). All the prior results in the study

are based on a 4 s delay and eg the electrophysiological data for 15 s and 25 s delay appears to be missing from the manuscript. It is thus unclear if the findings obtained by use of a 4 s delay hold true for longer delays, and overall, unified findings based on ALL the generated data are warranted. In my view, refraining from showing (and/or analyzing) the whole dataset severely hampers the quality and interpretation of the study. The authors also need to reconcile their findings with findings in two recently published studies on WM in frontal regions in mice^{8,9}.

Response: Reviewer #1 also raise the same concern. We initially examined neural activity using only 4 s (fixed condition) and 1-7 s (variable condition) delays. We performed a new experiment (recording IT neuronal activity with a long delay) to address this concern and show the results in Fig. 10. Please see our response to comment #4 of the reviewer #1. We also discussed our results in relation of the findings of Liu 2014 Science and Zhu 2020 Neuron (L 463-467).

The absence of parvalbumin expression in Rxfp3-Cre and Efr3a-Cre positive neurons and the physiological identification of putative fast-spiking (FS) opto-tagged units in both Rxfp3-Cre and Efr3a-Cre is a contradictory finding requiring clarification. The legend of Figure 2d indicates that immunostainings were performed in two animals (1 per genotype). These histological experiments should be repeated in more animals with statistically sound methods to ensure their robustness. In the case of a lack between Cre and PV positive neuronal populations, the authors need to clarify if the presence of FS opto-tagged units could be related to indirect recruitment of FS neurons upon light stimulation. Details of the light-response latencies for the 4 neuronal categories (PP-IT, PP-PT, FS-IT and FS-PT) as well as cross-correlograms between all opto-tagged units would help to elucidate if FS units have higher latencies than PP units and if PP and FS can have monosynaptic functional connections explaining indirect recruitment of FS units (PP->FS pairs with a positive peak lag). I strongly recommend that the authors carry out these additional experiments/analysis.

Response: We agree that the number of animals for immunostaining shown in Fig. 2 is rather small. We therefore collected data from additional animals (now total n = 5 animals for each genotype; Fig. 2f, g). We also performed further analysis to probe potential indirect activation of FS IT and FS PT neurons by light stimulation. Results from the analysis of spike waveform correlation and spike latency do not support indirect activation of FS IT and FS PT neurons (Fig. 2e). Results from the cross-correlation analysis do not support indirect activation, either, albeit the number of simultaneously tagged neurons is too small to draw a definitive conclusion (n = 2 pairs in Rxfp3-Cre and n = 3 pairs in Efr3a-Cre mice; see the figure below).

Cross-correlation between simultaneously recorded optically-tagged PP and optically-tagged FS neurons in Rxfp3-Cre (a) and Efr3a-Cre (b) mice. Red dotted line, 95% percentile of cross-correlation calculated using jittered spikes.

In the peak procedure behavioral task, the authors conclude that “This indicates that neither IT nor PT neurons are indispensable for controlling the timing of the licking response in the present peak procedure” (p11). This statement appears to contradict the previously claimed result: “PT neurons precisely encode the passage of time”. Do the authors mean that PT neurons encode the passage of time in one behavioral paradigm but not in another, or that prefrontal PT neurons encode the passage of time but this information is not used for timing in behavior (eg licking behavior)?

Response: We found that FS IT neurons convey not only target-related signals, but also strong temporal information during the delay period. Hence, it is not surprising that inactivation of PT neurons does not impair timing-related behavior. We revised the text to explain this more clearly (L 511-515). Perhaps inactivating both PT (PP PT and FS PT) and FS IT neurons would be necessary to impair timing-related behavior, but, unfortunately, it is currently not feasible to selectively inactivate FS IT neurons without inactivating PP IT neurons.

I, unfortunately, find the spike-LFP part weak, and question how this part supports the findings in the study. Important findings on spike-LFP theta synchrony in HPC-PFC interactions in spatial WM have been published, and while the authors refer to this work, their own probing of LFP activities in WM is shallow. The mPFC theta component during WM in the task used is not really investigated, but rather the analysis is solely a comparison of spike-theta coherence between IT and PT neurons specifically during correct trials with 4 s delays. I do not find that the quite limited analysis supports the suggestion that “IT neurons may contribute to the maintenance of working memory”. Further, I have problems envisioning the mechanism the authors imply in the last part of the sentence: “...IT neurons may contribute to the maintenance of working memory, not only via sample-dependent firing, but also by enhancing synchronous firing at theta frequencies”. Whose synchronous firing are the IT neurons enhancing here? I apologize if I am misunderstanding what is meant here. Fig 2b also implies that IT neurons are more strongly coupled to a broad LFP spectra than PT neurons (albeit not significantly), which is an interesting finding not pursued. Of note, most of the previous work investigated spatial WM; the current study did not; despite this the authors interpret their results based on findings in spatial WM task and refrain from discussing the validity of doing so, which I find troublesome.

Response: Thank you for pointing this out. We revised this part considering the reviewer's constructive comments. We now show LFP power spectrograms and spike-field coherence of untagged PP neurons in correct and error trials. The results show that theta band spike-field coherence during the delay period is stronger in correct than error trials even though theta LFP power during the delay period is similar between correct and error trials (Fig. 5b-e). In addition, we replaced "... but also by enhancing synchronous firing at theta frequencies" with "... but also by synchronous discharges at theta frequencies" (L 239-241). We meant that synchronous discharges among different IT neurons at theta frequencies may contribute to working memory. We'd rather refrain from pursuing the difference in spike-field coherence between IT and PT neurons at other frequencies than theta because the difference is small and we have limited statistical power to explore such small differences due to small numbers of IT and PT neurons (25 and 31, respectively, with mean firing rates ≥ 0.5 Hz with $<1\%$ of LFP signals reaching the ceiling). Finally, we revised the discussion to acknowledge that previous studies investigated spatial working memory unlike our study (L 472-474).

Specific points:

The study needs clarification throughout the text and figures regarding what neurons are included in different analyses/panels (and why), not just a statement of the n of neurons. It is hard to grasp what data regards the same neurons, and not, across the study.

Response: We apologize for not clearly explaining these in the initial manuscript. In the revised manuscript, we applied the following criteria to select neurons. In all analyses, we used those units with the minimum firing rate of 0.5 Hz and recorded in the sessions with at least 10 correct trials for both targets. Additionally, for the analysis of error (or miss) trials, we used those units recorded in the sessions with at least 2 error (or miss) trials for both targets. For the analysis of spike-field coupling, those neurons recorded in the sessions with more than 1% of LFP signals reaching the ceiling were excluded. For the analysis of variable delay trials, neurons recorded in the sessions with at least 10 correct trials with the length of delay ≥ 4 s were used. These are now clearly explained in the revised text (L 146-151 and throughout the text) as well as summarized in Supplementary Table 1.

I strongly recommend stating in the (all) figure panels what data is presented, eg what colors, line weight etc represents. This would greatly improve the accessibility of the data.

Response: We revised the figures and legends to improve clarity.

I find the use of the term 'sample identity' being questionable for the task design used. The spatial location of the sample lick spout appears to be what is to be remembered(?)

Response: 'Sample identity' meant to denote a set of trial-unique sensory/motor signals the animal could rely on to perform the task. To avoid confusion, we replaced 'sample identity' with 'target' throughout the text.

I find the use of refs 40-42 as support for the statement "...PT neurons have been proposed to play a more important role in the maintenance of working memory than IT neurons40-42" questionable.

Response: These papers explicitly proposed an important role of PT neurons in supporting working memory (an erroneous citation in Ref 44 has been corrected; Before correction, Dembrow, 2014, Front Neural Circuits; after correction, Dembrow, 2010, J Neurosci).

Ref 43: "CPn subgroups could amplify their spiking activities using the temporal properties of facilitation, higher reciprocal connectivity, and their intrinsic firing properties. Persistent firing during working memory in the frontal cortex is thought to underlie working memory (Wang, 1999). The synaptic properties and interconnectedness of CPn pairs described here suggest these neurons may provide a suitable substrate for generating persistent depolarization by recurrent excitation." (p10 L390; left column; CPn stands for 'corticopontine' corresponding to PT type neurons)

Ref 44: "The ability of cholinergic modulation to make neurons fire persistently was also dependent on their projection target. CPn neurons were more likely to fire persistently than COM neurons... cholinergic modulation makes them able to fire persistently beyond their stimulus input, making them well poised to contribute to mnemonic persistent activity that occurs during the delay period of working memory-like tasks." (p16 L935; left column; CPn and COM stand for 'corticopontine' and 'commissural' corresponding to PT and IT type neurons, respectively)

Ref 45: "Pronounced synaptic augmentation and PTP (post-tetanic potentiation) are also common phenomena found between cPCs in the mPFC.... Facilitation lasting hundreds of milliseconds, synaptic augmentation lasting up to 10 s, and PTP lasting up to minutes may be important to sustain the activity during short-term memories like those involved in a working memory task" (p5 L40; right column; cPCs stands for 'complex pyramidal cells' indicating PT type neurons)

In line with the 2 last comments, I recommend the authors to revise the ms to optimally reflect published findings and concepts.

Response: Please see our responses to the above two comments.

While the manuscript is well written, some statements are hard to grasp eg (p2) "In the primary motor cortex, IT neurons represent the deviation between current and subsequent behaviors, while PT neurons specifically encode the direction and speed of a subsequent behavioral response6".

Response: We revised the sentence so that it is clearer as the following (L 42-45):

“In the primary motor cortex, PT neurons specifically encode the direction and speed of a subsequent behavior during sensory-guided navigation. In contrast, layer 2/3 neurons are activated by unexpected direction changes induced by visual perturbations independent of behavioral response.”

Most of the methodological section provide sufficient detail. However, details regarding the training of the mice (number of days, evolution of the performance, representative video recordings of the task etc) are needed to support any concepts and conclusions.

Response: Done as suggested (L 580 - 603).

Summary

The authors hold relevant and significant datasets, with potential novelty. However, both the behavioral and electrophysiological data need a more complete analysis and presentation. I recommend the authors to base their interpretations and claims on the whole dataset, and to potentially reevaluate their concept of WM and the role of mPFC IT and PT neurons in the light of their extended findings. I am convinced the datasets hold important information on prefrontal processing in goal-oriented behavior, and particularly the involvement of PT and IT neurons, respectively, but I find that the authors need to let the (whole) data guide what conclusions can be drawn.

Response: Again, thanks for the detailed and constructive comments. As suggested, we improved our manuscript by adding additional experimental results, presenting more details of behavioral and physiological data, and revising the text to avoid confusion with respect to the concept of working memory.

References

1. Gerfen, C. R., Paletzki, R. & Heintz, N. GENSAT BAC Cre-Recombinase Driver Lines to Study the Functional Organization of Cerebral Cortical and Basal Ganglia Circuits. *Neuron* 80, 1368–1383 (2013).
2. Park, J. C., Bae, J. W., Kim, J. & Jung, M. W. Dynamically changing neuronal activity supporting working memory for predictable and unpredictable durations. *Sci. Rep.* 9, 15512 (2019).
3. Li, N., Chen, T.-W., Guo, Z. V., Gerfen, C. R. & Svoboda, K. A motor cortex circuit for motor planning and movement. *Nature* 519, 51–56 (2015).
4. Chen, T.-W., Li, N., Daie, K. & Svoboda, K. A Map of Anticipatory Activity in Mouse Motor Cortex. *Neuron* 94, 866-879.e4 (2017).
5. Pinto, L. & Dan, Y. Cell-Type-Specific Activity in Prefrontal Cortex during Goal-Directed Behavior. *Neuron* 87, 437–450 (2015).
6. Svoboda, K. & Li, N. Neural mechanisms of movement planning: motor cortex and beyond. *Curr. Opin. Neurobiol.* 49, 33–41 (2018).
7. Inagaki, H. K., Fontolan, L., Romani, S. & Svoboda, K. Discrete attractor

dynamics underlies persistent activity in the frontal cortex. Nature 566, 212 (2019).

8. *Liu, D. et al. Medial prefrontal activity during delay period contributes to learning of a working memory task. Science 346, 458–463 (2014).*

9. *Zhu, J. et al. Transient Delay-Period Activity of Agranular Insular Cortex Controls Working Memory Maintenance in Learning Novel Tasks. Neuron 105, 934-946.e5 (2020).*

Response: Thank you much for kindly providing these references.

REVIEWER COMMENTS

Reviewer #1 (Remarks to the Author):

The authors have sufficiently responded to all my concerns and I have no further questions.

Reviewer #2 (Remarks to the Author):

The authors have done an excellent job addressing my concerns, and I believe of the other reviewers as well. The study undoubtedly advances the field and provides the basis for a novel dichotomy between the role of IT neurons in working memory and PT neurons in timing. The paper presents a large amount of data, and now has 10 figures (three more than the first submission) the results are very compelling (but of course leave a number of fascinating questions open for future studies) and I don't think it would be reasonable to ask the authors to provide any more data/analyses.

Reviewer #3

We congratulate the authors to the well-executed revision of their study, the added experiments, analyses, information, and raw data significantly improve the scientific content and aid the interpretation of the data. We also much appreciate the increased stringency and precision in the wording and concepts. We below comment on the revised study, and the authors will see that the only remaining major question mark regards the classification of the neuron types. Apart from this our comments are suggestions and highlighting of need for clarifications, given to hopefully aid the authors to further improve the presentation on the data and the readers' understanding of the work.

I read the manuscript by Bae et al with great interest, and agree with the authors that the functional differentiation of IT vs PT neurons in the PFC is poorly investigated. In their study, Bae et al., take advantage of available knock-in mouse lines¹ to target IT and PT neurons of the mouse medial prefrontal cortex (mPFC). The authors make use of a delayed response task for head-fixed mice they have used previously², to investigate the roles prefrontal cortical IT and PT neurons play in working memory (WM). Tetrode recordings and optogenetics are applied, in combination for optotagging of IT and PT neurons, and optogenetics alone for broad silencing of prefrontal IT or PT neurons. Based on the experiments, the authors conclude that the delay-period activity of IT neurons conveys working memory-related signals with limited temporal information, while delay-period activity of PT neurons conveys precise information about the elapsed delay period in the form of ramping activity with minimal working memory-related signal.

Major concerns:

My major concern regards how WM is approached and conceived; in short, I don't find probing of WM convincingly demonstrated. This is partly due to the fact that too little behavioral data is presented (including the neuronal correlates). The concept of WM is vague in the current ms, and the author's definition of WM and what is specifically investigated unclear: "processing of WM", "maintenance of WM", and "WM related signals" are mentioned eg. At one place the authors write "sample identity (i.e., working memory)". The present ms gives the impression that WM in essence is spiking that happens during a task delay (any lengths) and that correlates to a stimulus location, which is a too simplistic view. It is generally agreed that WM is a multicomponent system that manipulates information storage for greater and more complex cognitive utility. WM is also considered to have a capacity limit. It would be relevant to know how eg how the authors separate WM from other types of memory, particularly short-term memory (experimentally and/or conceptually). The methods section gives away that several different fixed delay lengths were used, as well as random delays (the authors use, and present data from, different delay protocols for different experiments, unclear why). The beh data also includes different types of error trials, but the data for this, including the neuronal activity, is at large not presented. I wonder, how does eg the delay length affect behavior and neuronal activity, and how does the behavioral dataset give support for probing of WM specifically? Related, the authors recorded an impressive number

of single units, but much of the recording data is not presented, particularly not the variability in the activities, and it is not clear why and how different sets and numbers of neurons are used. In essence, the authors put in a lot of work in the behavior and recordings but refrain from presenting some key data, data that could possibly allow investigation of prefrontal processing involved in WM. The unclear selection of data and analysis together with some conceptual imprecisions leave the impression of a patchy story, and unfortunately also give the sense of cherry-picking (although I find that the eventual cherry-picking more than anything devalues any potential findings).

Response: We appreciate the reviewer's detailed and constructive comments. In the initial manuscript, given that very little is known about IT and PT neuronal roles in working memory, we tried to focus on IT and PT neuronal contributions to a specific and well-studied neural process (target-related delay-period activity) using a popular behavioral paradigm (delayed response task). In other words, we tried to keep the agenda simple. However, as the reviewer indicated, a more comprehensive presentation of the behavioral and neural data would be helpful to better understand how IT and PT neurons contribute to working memory required for the current delayed response task. To address the reviewer's concerns, we've made substantial changes to the manuscript. Specifically, in the revised manuscript, we used similar sets of neurons across different analyses by applying general criteria for cell inclusion and explain clearly which neurons were included in each analysis (L 146-151 and throughout the text; also summarized in Supplementary Table 1); indicate which aspect of working memory we're interested in ('maintenance of task-relevant information in the absence of external sensory cues', Introduction, L 64-66), replaced 'working memory-related signals' with 'target-related signals' throughout the text where appropriate, and discuss not only information maintenance, but also information manipulation aspects of working memory ('retrospective versus prospective memory'; Discussion, L 448-457); show additional analysis results for behavioral performance (Fig. 1d, e; Supplementary Fig. 2) as well as neural activity during the task (Fig. 3b-d; Fig. 7); explain more clearly the experimental procedures including delay protocols (L 580-603); show the results from the analysis of error-trial (and also miss-trial) neural activity (Fig. 4; Fig. 5b-g; Fig.10; Supplementary Fig. 7; also see our response to the second comment of reviewer #1); and compare behavioral performance and neural activity across the fixed and variable delay conditions (Supplementary Fig. 5). We believe our revised manuscript provides much richer information about the way IT and PT neurons contribute to working memory.

Clarify and streamline the description of analyses, and the terminology:

We salute the improved stringency in the language and terminology relating to WM. 'Maintenance of task-relevant information in the absence of external sensory cues' is stated as the main interest. Target-related signals are probed. It is however unclear how target-related signals relate to activity reflecting the target in the present manuscript (definition of target (row 1035): 'lick port presented during the sample phase (target)'. The authors analyze activity in relationship to the target (right vs left) but are they performing analysis of anything target-related that is not actually the target (location)? The term target-dependent is also frequently used. This is confusing – what is the difference between target / target-dependent / target-related signals (activity)? How do the analyses of the 3 things differ? If these 3 things are actually the same thing, we recommend that

a single term is consistently used to clarify this. If they are not the same thing, the difference between the terms and their definitions should be clarified. Eg. In Fig S7 the heading is 'Neural decoding of target differs between error and miss trials' but in Fig 3 'IT, but not PT, neurons convey robust target-related signals'. Isn't actually the same thing decoded in the two figures? Also relevant for e.g., Fig 4.

There is some mixing of sample and target that is a bit confusing, e.g., 'to examine dynamics of sample versus upcoming choice-dependent neural activity during the delay period' (rows 177-78). We recommend to use the term sample to specifically refer to the sample PHASE and consistently state target when the lick port presented during the sample phase is what is actually meant. In line with this, this statement is confusing: 'These results indicate that IT neurons convey robust target-related signals by maintaining sample-related, rather than choice-related, neural activity more persistently than PT neurons during the delay period' (rows 196-98). What is the difference/relationship between target (related) and sample (related) here?

Row 159: "We found IT neurons tend to show delay-period activity that varies according to target'. Row 140: 'IT, but not PT, neurons convey robust target-related signals'. Tend and robust are quite different things, and overall we question the robustness. Fig 3f shows decoding of target from opto-tagged neurons, Fig S6b from non-tagged PP neurons. Decoding of target from IT neurons (3f) is not much better than from PP neurons (S6b). The PP population holds IT neurons but also many other types of pyramidal neurons with differential projection patterns, still the decoding is not much worse than from IT neurons. Together the data indicate that encoding of target is not a general trait of (all) IT neurons, if it was the decoding should be considerable better than for the PP population (?) I.e., robust appears to be an overstatement.

In line with this Fig 3 shows that not all IT neurons display delay-period activity that varies according to target and that there is very high temporal variability in the activity within the IT population. However, the text statements read like target-dependent activity is a general property of the IT population, which is a stretch. Further, Fig 3b-c shows significant target-signals in PT neurons, but the text reads that there is a complete absence of target selectivity in PT neurons. We advise the authors to adjust the statements to reflect the actual data as to not mislead the field - the actual data is interesting and important in itself, and overstatements hurt rather than strengthen the study.

An observation:

If the whole 4s delay is included in Fig 3f, this data can be directly compared to the data in Fig 4b, right. Together these panels indicate that the target can be decoded equally well from PP neurons during *error* trials as from IT neurons during correct trials? The target is encoded by PP neurons (that presumably include IT neurons) but the animals still perform an error? The decoding of target in correct trials appear to be better for the mixed PP population (4b, right) than the pure IT population (3f)? What does this say about encoding of target by the IT population, and the possible functional role of this encoding, including during WM necessary for successful behavior? We don't know the answers to these questions but perhaps they can be fruit for thought for the authors in their interpretations and statements. Also, would be lovely to have the data for IT neurons in Fig 3 plotted as Fig 4, left, for direct comparisons.

Detailed comments:

Neurons in frontal cortical areas have been shown to be tuned to multiple task variables (eg sensory and motor variables)^{4,5} and mixed selectivity is also prominent in prefrontal regions. Despite the sophistication of the task design used, eg Figure 3a only shows 4 example units(?),

in correct trials specifically. It would be appropriate for the author to present/represent the full dataset in a main figure, and also population analysis. The study would be greatly strengthened by presentation of the richness of the dataset and of how identified prefrontal subpopulations align to the different task variables (eg light, sound, licking). As mentioned, accounting for activities in error trials and in relation to delay lengths is also of importance. Presenting the whole data set would also aid the readers' interpretation of how representative the optotagged activities are. Additionally, videos of representative behavior in the DMS task would be a great addition to the manuscript, and allow the reader to qualitatively assess the task used and the robustness and details of the behavior.

Response: We agree that it would be helpful to present behavioral and neural data in a more comprehensive manner. As suggested, we now show the diversity of IT and PT neuronal responses as a separate figure (Fig. 7). The new figure shows IT and PT neuronal responses to diverse events of the task. We also show summary heat maps for target-dependent activity of all IT and PT neurons that were included in the analysis (Fig. 3b). We also show the results of error-trial analysis (Fig. 4; Fig. 5b-g; Fig. 10; Supplementary Fig. 7), compare neural activity across fixed and variable delay conditions (Supplementary Fig. 5), and show delay-period activity during long (15 and 25 s) delays (Fig. 10). Finally, we uploaded a video of representative behavior (Supplementary video 1).

The new analysis presented in figure 7 is an important addition to the study. We encourage the authors to discuss the obtained results in reference to the concept of mixed selectivity (see references 4 and 5 in first round to reviews). The addition of the calcium imaging experiments to study the neuronal activity during long delays fill a gap present in the first version of the manuscript. In relation to this new data, we advise the authors to check their terminology – in most analyses in the manuscript activity of IT and PT neurons are mapped to specific task variables, task phases or behavioral variables but the terminology varies. The term encoding appears to be cherry picked for some variables/analyses, signals for others, and in this new section the term responsive is used, implying a very passive role of the neurons. This paints a skewed picture of the data and we strongly encourage the authors to adjust and streamline their terminology across the study to fairly reflect all data. Alternatively, state why some analysis reveals encoding and others eg responsiveness. What is the difference in the current study?

In Figure 5, the authors find with a demixed principal component analysis (dPCA) that PT neurons encode the passage of time, and display ramping activity during the delay. As the authors are aware, PT and IT subpopulations have been studied during motor planning in a similar head-fixed paradigm (the sample epoch is different in this paradigm - the animals have to assess the position of a pole presented in front of their whiskers), in a different prefrontal subregion (ALM)³. In this study, PT neuron activity was shown to ramp at the end of the delay preceding contralateral (to the recorded hemisphere) licking movements. Motor planning has been defined as “internal processes by which the brain plans and executes volitional movements”⁶ and the ramping activity during the delay epoch before the onset of movement has been shown to represent motor-related information^{3,7}. In light of these findings, it is important that the authors assess possible PT ramping activity before contralateral licking movements, as this has bearing on their interpretations. Further, to support a claim that “PT neurons precisely encode the passage of

time”, experiments showing how the ramping activity reflects temporal variations (eg different delay durations) are warranted. These are only suggestive starting points for the interpretation that PT neurons convey time-related information.

Response: As the reviewer pointed out, timing-related PT neuronal activity might be related to encoding the passage of time or, alternatively, motor planning. However, as we mentioned in our response to comment #3 of reviewer #1, we found a largely similar pattern of time encoding by PT neurons under the variable (1-7 s) delay condition (Supplementary Fig. 5g and i). This cannot be explained by motor planning because the time of motor response could not be predicted in the variable delay condition. Note that we did find potential motor planning-related PT neuronal activity during the last ~1 s during the delay period (Fig. 3g). However, PT neuronal population did not show biased firing toward contralateral licking responses (Fig. 7k), indicating that it is distinct from motor planning-related activity found in the motor cortex. More importantly, PT neuronal ramping activity spanning the delay period showed similar ramping activity profiles between ipsilateral and contralateral choice trials (‘target-independent dPC’, Fig. 6e; activities of sample PT neurons, Fig 3a and Supplementary Fig. 5a) unlike upcoming choice-related PT neuronal activity at the end of the delay period. These results consistently indicate that PT neuronal ramping activity spanning the delay period is related to the passage of time rather than motor planning. We discussed this matter in the revised text (L 326-333).

While we agree that PT neurons appear to convey more temporal information than IT neurons, we don't think the analyses performed support the statement “the delay-period activity of PT neurons conveys **precise** information about the elapsed delay period’ (rows 279-80). To state precise encoding of elapsed time, additional analyses would be needed, including eg analysis of subjective vs actual time, etc. Again the overstatement hurt rather than strengthen the study here. There are many variables/processes that can appear to reflect time/timing but that is actually a reflection of something else and that would need to be excluded to support the authors statement. We encourage the authors to modify their statements regarding precise encoding of time and to contemplate if their data actually support anything beyond PT neurons' delay activity *being related to the passage of time/convey more temporal information than IT neurons* (also statements given by the authors and that appears more appropriately reflect the data). The discussion eg states interval timing, which is not directly investigated in the current study. A role for PT neurons in timing is stated (eg discussion) but inactivation of these neurons does not change the timing of the behavior – this finding alone begs for caution in the interpretation of the data regarding PT neurons and encoding of time? Furthermore, that a temporal component can be decoded is not the same thing as the neurons convey precise information about the elapsed time.

In their photoinhibition experiments, the authors find no effect on beh performance with optogenetic inhibition of PT and IT neurons with use of 4 s delay – inhibition of IT neurons deteriorated behavior with use of longer delays (15 s and 25 s). All the prior results in the study are based on a 4 s delay and eg the electrophysiological data for 15 s and 25 s delay appears to be missing from the manuscript. It is thus unclear if the findings obtained by use of a 4 s delay hold true for longer delays, and overall, unified findings based on ALL the generated data are warranted. In my view, refraining from showing (and/or analyzing) the whole dataset severely hampers the quality and interpretation of the study. The authors also need to reconcile their

findings with findings in two recently published studies on WM in frontal regions in mice^{8,9}.

Response: Reviewer #1 also raise the same concern. We initially examined neural activity using only 4 s (fixed condition) and 1-7 s (variable condition) delays. We performed a new experiment (recording IT neuronal activity with a long delay) to address this concern and show the results in Fig. 10. Please see our response to comment #4 of the reviewer #1. We also discussed our results in relation of the findings of Liu 2014 Science and Zhu 2020 Neuron (L 463-467).

Again, this new experiment strengthens the study.

The absence of parvalbumin expression in Rxfp3-Cre and Efr3a-Cre positive neurons and the physiological identification of putative fast-spiking (FS) opto-tagged units in both Rxfp3-Cre and Efr3a-Cre is a contradictory finding requiring clarification. The legend of Figure 2d indicates that immunostainings were performed in two animals (1 per genotype). These histological experiments should be repeated in more animals with statistically sound methods to ensure their robustness. In the case of a lack between Cre and PV positive neuronal populations, the authors need to clarify if the presence of FS opto-tagged units could be related to indirect recruitment of FS neurons upon light stimulation. Details of the light-response latencies for the 4 neuronal categories (PP-IT, PP-PT, FS-IT and FS-PT) as well as cross-correlograms between all opto-tagged units would help to elucidate if FS units have higher latencies than PP units and if PP and FS can have monosynaptic functional connections explaining indirect recruitment of FS units (PP->FS pairs with a positive peak lag). I strongly recommend that the authors carry out these additional experiments/analysis.

Response: We agree that the number of animals for immunostaining shown in Fig. 2 is rather small. We therefore collected data from additional animals (now total n = 5 animals for each genotype; Fig. 2f, g). We also performed further analysis to probe potential indirect activation of FS IT and FS PT neurons by light stimulation. Results from the analysis of spike waveform correlation and spike latency do not support indirect activation of FS IT and FS PT neurons (Fig. 2e). Results from the cross-correlation analysis do not support indirect activation, either, albeit the number of simultaneously tagged neurons is too small to draw a definitive conclusion (n = 2 pairs in Rxfp3-Cre and n = 3 pairs in Efr3a-Cre mice; see the figure below).

Cross-correlation between simultaneously recorded optically-tagged PP and optically-tagged FS neurons in Rxfp3-Cre (a) and Efr3a-Cre (b) mice. Red dotted line, 95% percentile of cross-correlation calculated using jittered spikes.

We appreciate the further analysis, which strengthens the conclusion that the neurons denoted FS are not FS PV-expressing inhibitory interneurons. When considering this added data, we started to question the fast-spiking properties of the population denoted FS. When looking at Fig 2c we therefore wonder about the authors' classification of the neurons (we are aware of what is stated in the Methods section). Many FS neurons in Fig 2c appear to have down to 0 Hz in mean FR, or similar FR as many PP neurons (it is the mean FR that is plotted in Fig 2c?). Furthermore, there are putative PP neurons with a very narrow half-valley width in the plot. These two observations questions the classification.

We advise the authors to clarify how their classification supports identification and denoting of fast-spiking neurons as Fig 2c questions the classification. If FR is not a criterion in the classification (it says that it is in the Methods, but from 2c we don't understand this), the neurons can only be classified as eg narrow-spiking and wide-spiking (ie based on waveform). The authors need to make sure that their classification is of best scientific practice, and show the data for all parameters used for the classification (including the peak to valley ratio, particularly for classification of narrow-spiking neurons; using the peak to trough or the peak to baseline rather than the half-valley width will help to clarify the classification), preferentially in integration in a figure panel (3 axes plot?). More info in the methods would also be helpful. As the ms contains a section on the FS neurons, and conclusions are drawn, it is imperative that the classification is as accurate as possible. (Perhaps of interest: subpopulations of eg somatostatin interneurons in the mPFC are narrow-spiking).

In the peak procedure behavioral task, the authors conclude that "This indicates that neither IT nor PT neurons are indispensable for controlling the timing of the licking response in the present peak procedure" (p11). This statement appears to contradict the previously claimed result: "PT neurons precisely encode the passage of time". Do the authors mean that PT neurons encode the passage of time in one behavioral paradigm but not in another, or that prefrontal PT neurons encode the passage of time but this information is not used for timing in behavior (eg licking behavior)?

Response: We found that FS IT neurons convey not only target-related signals, but also strong temporal information during the delay period. Hence, it is not surprising that inactivation of PT neurons does not impair timing-related behavior. We revised the text to explain this more clearly (L 511-515). Perhaps inactivating both PT (PP PT and FS PT) and FS IT neurons would be necessary to impair timing-related behavior, but, unfortunately, it is currently not feasible to selectively inactivate FS IT neurons without inactivating PP IT neurons.

I, unfortunately, find the spike-LFP part weak, and question how this part supports the findings in the study. Important findings on spike-LFP theta synchrony in HPC-PFC interactions in spatial WM have been published, and while the authors refer to this work, their own probing of LFP activities in WM is shallow. The mPFC theta component during WM in the task used is not really investigated, but rather the analysis is solely a comparison of spike-theta coherence between IT and PT neurons specifically during correct trials with 4 s delays. I do not find that the quite limited analysis supports the suggestion that "IT neurons may contribute to the maintenance of working memory". Further, I have problems envisioning the mechanism the authors imply in the last part of the sentence: "...IT neurons may contribute to the maintenance of working memory, not only via sample-dependent firing, but also by enhancing synchronous firing at theta frequencies".

Whose synchronous firing are the IT neurons enhancing here? I apologize if I am misunderstanding what is meant here. Fig 2b also implies that IT neurons are more strongly coupled to a broad LFP spectra than PT neurons (albeit not significantly), which is an interesting finding not pursued. Of note, most of the previous work investigated spatial WM; the current study did not; despite this the authors interpret their results based on findings in spatial WM task and refrain from discussing the validity of doing so, which I find troublesome.

Response: Thank you for pointing this out. We revised this part considering the reviewer's constructive comments. We now show LFP power spectrograms and spike-field coherograms of untagged PP neurons in correct and error trials. The results show that theta band spike-field coherence during the delay period is stronger in correct than error trials even though theta LFP power during the delay period is similar between correct and error trials (Fig. 5b-e). In addition, we replaced "... but also by enhancing synchronous firing at theta frequencies" with "... but also by synchronous discharges at theta frequencies" (L 239-241). We meant that synchronous discharges among different IT neurons at theta frequencies may contribute to working memory. We'd rather refrain from pursuing the difference in spike-field coherence between IT and PT neurons at other frequencies than theta because the difference is small and we have limited statistical power to explore such small differences due to small numbers of IT and PT neurons (25 and 31, respectively, with mean firing rates ≥ 0.5 Hz with $<1\%$ of LFP signals reaching the ceiling). Finally, we revised the discussion to acknowledge that previous studies investigated spatial working memory unlike our study (L 472-474).

Specific points:

The study needs clarification throughout the text and figures regarding what neurons are included in different analyses/panels (and why), not just a statement of the n of neurons. It is hard to grasp what data regards the same neurons, and not, across the study.

Response: We apologize for not clearly explaining these in the initial manuscript. In the revised manuscript, we applied the following criteria to select neurons. In all analyses, we used those units with the minimum firing rate of 0.5 Hz and recorded in the sessions with at least 10 correct trials for both targets. Additionally, for the analysis of error (or miss) trials, we used those units recorded in the sessions with at least 2 error (or miss) trials for both targets. For the analysis of spike-field coupling, those neurons recorded in the sessions with more than 1% of LFP signals reaching the ceiling were excluded. For the analysis of variable delay trials, neurons recorded in the sessions with at least 10 correct trials with the length of delay ≥ 4 s were used. These are now clearly explained in the revised text (L 146-151 and throughout the text) as well as summarized in Supplementary Table 1.

Supplementary Table 1 is an important and appreciated addition to the manuscript.

I strongly recommend stating in the (all) figure panels what data is presented, eg what colors, line weight etc represents. This would greatly improve the accessibility of the data.

Response: We revised the figures and legends to improve clarity.

We appreciate the improvements but find need for further information in the figures. Eg Fig 1c has no scale bar, and 1d one needs to dig in the legend to find what purple and green represents, respectively. Having the neuron numbers in the figure panels plotting population data would be very helpful, including eg Fig 2c.

I find the use of the term 'sample identity' being questionable for the task design used. The spatial location of the sample lick spout appears to be what is to be remembered(?)

Response: 'Sample identity' meant to denote a set of trial-unique sensory/motor signals the animal could rely on to perform the task. To avoid confusion, we replaced 'sample identity' with 'target' throughout the text.

Please see comment about target and sample above.

I find the use of refs 40-42 as support for the statement "...PT neurons have been proposed to play a more important role in the maintenance of working memory than IT neurons40-42" questionable.

Response: These papers explicitly proposed an important role of PT neurons in supporting working memory (an erroneous citation in Ref 44 has been corrected; Before correction, Dembrow, 2014, Front Neural Circuits; after correction, Dembrow, 2010, J Neurosci).

This we agree with, and that was why we gave this comment – the papers do not state that "*PT neurons (have been proposed to) play a more important role in the maintenance of working memory than IT neurons*". This statement reflects an interpretation and stretch made by the authors, used to support their findings.

Ref 43: "CPn subgroups could amplify their spiking activities using the temporal properties of facilitation, higher reciprocal connectivity, and their intrinsic firing properties. Persistent firing during working memory in the frontal cortex is thought to underlie working memory (Wang, 1999). The synaptic properties and interconnectedness of CPn pairs described here suggest these neurons may provide a suitable substrate for generating persistent depolarization by recurrent excitation." (p10 L390; left column; CPn stands for 'corticopontine' corresponding to PT type neurons)

Ref 44: "The ability of cholinergic modulation to make neurons fire persistently was also dependent on their projection target. CPn neurons were more likely to fire persistently than COM neurons... cholinergic modulation makes them able to fire persistently beyond their stimulus input, making them well poised to contribute to mnemonic persistent activity that occurs during the delay period of working memory-like tasks." (p16 L935; left column; CPn and COM stand for 'corticopontine' and 'commissural' corresponding to PT and IT type neurons, respectively)

Ref 45: "Pronounced synaptic augmentation and PTP (post-tetanic potentiation) are also common phenomena found between cPCs in the mPFC.... Facilitation lasting hundreds of milliseconds, synaptic augmentation lasting up to 10 s, and PTP lasting up to minutes may be important to sustain the activity during short-term memories like those involved in a working memory task" (p5 L40; right column; cPCs stands for 'complex pyramidal cells' indicating PT type neurons)

In line with the 2 last comments, I recommend the authors to revise the ms to optimally reflect published findings and concepts.

Response: Please see our responses to the above two comments.

While the manuscript is well written, some statements are hard to grasp eg (p2) "In the primary motor cortex, IT neurons represent the deviation between current and subsequent behaviors, while PT neurons specifically encode the direction and speed of a subsequent behavioral response6".

Response: We revised the sentence so that it is clearer as the following (L 42-45):

"In the primary motor cortex, PT neurons specifically encode the direction and speed of a subsequent behavior during sensory-guided navigation. In contrast, layer 2/3 neurons are activated by unexpected direction changes induced by visual perturbations independent of behavioral response."

Most of the methodological section provide sufficient detail. However, details regarding the training of the mice (number of days, evolution of the performance, representative video recordings of the task etc) are needed to support any concepts and conclusions.

Response: Done as suggested (L 580 - 603).

Summary

The authors hold relevant and significant datasets, with potential novelty. However, both the behavioral and electrophysiological data need a more complete analysis and presentation. I recommend the authors to base their interpretations and claims on the whole dataset, and to potentially reevaluate their concept of WM and the role of mPFC IT and PT neurons in the light of their extended findings. I am convinced the datasets hold important information on prefrontal processing in goal-oriented behavior, and particularly the involvement of PT and IT neurons, respectively, but I find that the authors need to let the (whole) data guide what conclusions can be drawn.

Response: Again, thanks for the detailed and constructive comments. As suggested, we improved our manuscript by adding additional experimental results, presenting more details of behavioral and physiological data, and revising the text to avoid confusion with respect to the concept of working memory.

We are very happy to see that the authors found our comments constructive – the revised manuscript gives support for our initial impression that the study holds important and relevant data on prefrontal processing and functioning.

References

1. Gerfen, C. R., Paletzki, R. & Heintz, N. *GENSAT BAC Cre-Recombinase Driver Lines to Study the Functional Organization of Cerebral Cortical and Basal Ganglia Circuits. Neuron* 80, 1368–1383 (2013).
2. Park, J. C., Bae, J. W., Kim, J. & Jung, M. W. *Dynamically changing neuronal activity supporting working memory for predictable and unpredictable durations. Sci. Rep.* 9, 15512 (2019).
3. Li, N., Chen, T.-W., Guo, Z. V., Gerfen, C. R. & Svoboda, K. *A motor cortex circuit for motor planning and movement. Nature* 519, 51–56 (2015).
4. Chen, T.-W., Li, N., Daie, K. & Svoboda, K. *A Map of Anticipatory Activity in Mouse Motor Cortex. Neuron* 94, 866-879.e4 (2017).
5. Pinto, L. & Dan, Y. *Cell-Type-Specific Activity in Prefrontal Cortex during Goal-Directed Behavior. Neuron* 87, 437–450 (2015).
6. Svoboda, K. & Li, N. *Neural mechanisms of movement planning: motor cortex and beyond. Curr. Opin. Neurobiol.* 49, 33–41 (2018).
7. Inagaki, H. K., Fontolan, L., Romani, S. & Svoboda, K. *Discrete attractor dynamics underlies persistent activity in the frontal cortex. Nature* 566, 212 (2019).
8. Liu, D. et al. *Medial prefrontal activity during delay period contributes to learning of a working memory task. Science* 346, 458–463 (2014).
9. Zhu, J. et al. *Transient Delay-Period Activity of Agranular Insular Cortex Controls Working Memory Maintenance in Learning Novel Tasks. Neuron* 105, 934-946.e5 (2020).

Response: Thank you much for kindly providing these references.

Various comments:

Row 173-74: ‘This cannot be attributed to the mPFC neurons of Rxfp3-Cre mice carrying stronger target-related signals than those of Efr3a-Cre mice (Supplementary Fig. 6)’. This statement is quite confusing given what is stated before in this section. Do the authors mean that recorded PP neurons in Rxfp3-Cre mice do not only include IT neurons, and as a *population* PP neurons in Rxfp3-Cre mice do not carry ‘stronger’ target-related signals than the PP *population* in Efr3a-Cre mice? (And what is meant with stronger here?). This text would benefit from modification.

It would be helpful for the readers if the authors in the section ‘Delay-period activity differs between correct and error trials’ (and anywhere else appropriate) point out that the PP neurons population represents random sampling of putative pyramidal neurons in the two mouse lines (i.e., is different from IT or PT populations). This to eliminate confusion and misunderstanding.

While we very much appreciate Fig S4, the present format prohibits inspection of the data – the plotting is just too small.

Please consistently state what parameters (neurons, sessions, trials etc) are indicated with given numbers (n), see eg Fig S7 figure legend. Alternatively, give the n directly after the relevant parameter

instead of eg in the end of the sentence to avoid confusion, see e.g., row 260-61: ‘...using the untagged PP neurons recorded in the sessions with ≥ 2 error trials for both targets (n = 365). Much time is spent figuring out the n of parameters, and what parameters stated ns refer to.

Row 167: “Also, decoding of **correct** target using delay-period neuronal ensemble activity improved as a function of ensemble size for the IT neurons, but remained similar to the level of chance (50%) across all tested PT neuron ensemble sizes (Fig. 3e).” Is the target, or correct reporting of target, decoded here (were not only correct trials included?)

Row 295; would be helpful if ‘trial outcome-dependent firing’ was defined; what is analyzed and during what phase.

Would be helpful if the peak procedure was shortly explained, what is specifically investigated and how.

Abstract (row 26) and Discussion (row 430): “PT neurons convey far more temporal information” “Far” a quite unscientific and imprecise.

Row 163: the “absolute target selectivity index” appears to not be included in the method section. We apologize if we are mistaken.

We would suggest to move the new section “IT and PT neurons are responsive to diverse task-events” at an earlier position in the manuscript (before “IT, but not PT, neurons convey robust target-related signals” maybe) as this order would be more natural to the reader.

Authors: Bae, Jeong, Yoon, Bae, Lee, Paik & Jung

Manuscript: NCOMMS-20-38699A, "Parallel processing of working memory and temporal information by distinct types of cortical projection neurons"

We are grateful to the editor and reviewers for their positive comments and additional constructive suggestions. The original comments of reviewer #3 and our responses to them are indicated in black, the additional comments of reviewer #3 in green, and our responses to the new comments in blue.

Reviewer #3

We congratulate the authors to the well-executed revision of their study, the added experiments, analyses, information, and raw data significantly improve the scientific content and aid the interpretation of the data. We also much appreciate the increased stringency and precision in the wording and concepts. We below comment on the revised study, and the authors will see that the only remaining major question mark regards the classification of the neuron types. Apart from this our comments are suggestions and highlighting of need for clarifications, given to hopefully aid the authors to further improve the presentation on the data and the readers' understanding of the work.

Response: We appreciate the reviewer's positive evaluation of our revision. We believe that our manuscript has been improved further by incorporating the additional helpful comments of the reviewer.

I read the manuscript by Bae et al with great interest, and agree with the authors that the functional differentiation of IT vs PT neurons in the PFC is poorly investigated. In their study, Bae et al., take advantage of available knock-in mouse lines¹ to target IT and PT neurons of the mouse medial prefrontal cortex (mPFC). The authors make use of a delayed response task for head-fixed mice they have used previously², to investigate the roles prefrontal cortical IT and PT neurons play in working memory (WM). Tetrode recordings and optogenetics are applied, in combination for optotagging of IT and PT neurons, and optogenetics alone for broad silencing of prefrontal IT or PT neurons. Based on the experiments, the authors conclude that the delay-period activity of IT neurons conveys working memory-related signals with limited temporal information, while delayperiod activity of PT neurons conveys precise information about the elapsed delay period in the form of ramping activity with minimal working memory-related signal.

Major concerns:

My major concern regards how WM is approached and conceived; in short, I don't find probing of WM convincingly demonstrated. This is partly due to the fact that too little behavioral data is presented (including the neuronal correlates). The concept of WM is vague in the current ms, and the author's definition of WM and what is specifically investigated unclear: "processing of WM", "maintenance of WM", and "WM related signals" are mentioned eg. At one place the authors write "sample identity (i.e., working memory)". The present ms gives the impression that WM in essence is spiking that happens during a task delay (any lengths) and that correlates to a stimulus location, which is a too simplistic view. It is generally agreed that WM is a multicomponent system that manipulates information storage for greater and more complex

cognitive utility. WM is also considered to have a capacity limit. It would be relevant to know how eg how the authors separate WM from other types of memory, particularly short-term memory (experimentally and/or conceptually). The methods section gives away that several different fixed delay lengths were used, as well as random delays (the authors use, and present data from, different delay protocols for different experiments, unclear why). The beh data also includes different types of error trials, but the data for this, including the neuronal activity, is at large not presented. I wonder, how does eg the delay length affect behavior and neuronal activity, and how does the behavioral dataset give support for probing of WM specifically? Related, the authors recorded an impressive number of single units, but much of the recording data is not presented, particularly not the variability in the activities, and it is not clear why and how different sets and numbers of neurons are used. In essence, the authors put in a lot of work in the behavior and recordings but refrain from presenting some key data, data that could possibly allow investigation of prefrontal processing involved in WM. The unclear selection of data and analysis together with some conceptual imprecisions leave the impression of a patchy story, and unfortunately also give the sense of cherry-picking (although I find that the eventual cherry-picking more than anything devalues any potential findings).

Response: We appreciate the reviewer's detailed and constructive comments. In the initial manuscript, given that very little is known about IT and PT neuronal roles in working memory, we tried to focus on IT and PT neuronal contributions to a specific and well-studied neural process (target-related delay-period activity) using a popular behavioral paradigm (delayed response task). In other words, we tried to keep the agenda simple. However, as the reviewer indicated, a more comprehensive presentation of the behavioral and neural data would be helpful to better understand how IT and PT neurons contribute to working memory required for the current delayed response task. To address the reviewer's concerns, we've made substantial changes to the manuscript. Specifically, in the revised manuscript, we used similar sets of neurons across different analyses by applying general criteria for cell inclusion and explain clearly which neurons were included in each analysis (L 146-151 and throughout the text; also summarized in Supplementary Table 1); indicate which aspect of working memory we're interested in ('maintenance of task-relevant information in the absence of external sensory cues', Introduction, L 64-66), replaced 'working memory-related signals' with 'target-related signals' throughout the text where appropriate, and discuss not only information maintenance, but also information manipulation aspects of working memory ('retrospective versus prospective memory'; Discussion, L 448-457); show additional analysis results for behavioral performance (Fig. 1d, e; Supplementary Fig. 2) as well as neural activity during the task (Fig. 3b-d; Fig. 7); explain more clearly the experimental procedures including delay protocols (L 580-603); show the results from the analysis of error-trial (and also miss-trial) neural activity (Fig. 4; Fig. 5b-g; Fig.10; Supplementary Fig. 7; also see our response to the second comment of reviewer #1); and compare behavioral performance and neural activity across the fixed and variable delay conditions (Supplementary Fig. 5). We believe our revised manuscript provides much richer information about the way IT and PT neurons contribute to working memory.

Clarify and streamline the description of analyses, and the terminology:

We salute the improved stringency in the language and terminology relating to WM. 'Maintenance of task-relevant information in the absence of external sensory cues' is stated as the main interest. Target-related signals are probed. It is however unclear how target-related signals relate to activity reflecting the target in the present manuscript (definition of target (row 1035): 'lick port presented during the sample phase (target)'). The authors analyze activity in relationship to the target (right vs left) but are they performing analysis of anything target-related that is not actually the target (location)? The term target-dependent is also frequently used. This

is confusing – what is the difference between target / target-dependent / target-related signals (activity)? How do the analyses of the 3 things differ? If these 3 things are actually the same thing, we recommend that a single term is consistently used to clarify this. If they are not the same thing, the difference between the terms and their definitions should be clarified. Eg. In Fig S7 the heading is ‘Neural decoding of target differs between error and miss trials’ but in Fig 3 ‘IT, but not PT, neurons convey robust target-related signals’. Isn’t actually the same thing decoded in the two figures? Also relevant for e.g., Fig 4.

Response: Target-dependent activity, by definition, refers to the activity that varies according to any subset of sensory information that can be used to discriminate between the two targets. As the reviewer pointed out, it may be related to the spatial location of the lick port, a slightly different visual feature between the two lick ports, the direction of tongue movement, etc. With respect to the terminology describing target-related activity, we agree to the reviewer’s comment. We now use two expressions consistently throughout the manuscript: ‘target signals’ to denote the information about target (e.g., IT neurons convey stronger target signals than PT neurons) and ‘target-dependent’ activity to denote differential firing according to target (e.g., many IT neurons showed target-dependent activity). This is much like saying, for example, ‘auditory cortical neurons convey sound frequency signals’ and ‘auditory cortical neurons show frequency-dependent activity’.

There is some mixing of sample and target that is a bit confusing, e.g., ‘to examine dynamics of sample versus upcoming choice-dependent neural activity during the delay period’ (rows 177-78). We recommend to use the term sample to specifically refer to the sample PHASE and consistently state target when the lick port presented during the sample phase is what is actually meant. In line with this, this statement is confusing: ‘These results indicate that IT neurons convey robust target-related signals by maintaining sample-related, rather than choice-related, neural activity more persistently than PT neurons during the delay period’ (rows 196-98). What is the difference/relationship between target (related) and sample (related) here?

Response: We agree that ‘sample-related activity’ and ‘sample phase’ would be confusing. As suggested, we revised the text so that the term ‘sample’ specifically refers to the sample phase (L 191, 199, 455, and 460). For example, we changed ‘These results indicate that IT neurons convey robust target-related signals by maintaining sample-related, rather than choice-related, neural activity more persistently than PT neurons during the delay period’ to ‘These results indicate that IT neurons carry robust target signals by maintaining sample-phase activity more persistently than PT neurons during the delay period’.

Row 159: “We found IT neurons tend to show delay-period activity that varies according to target’. Row 140: ‘IT, but not PT, neurons convey robust target-related signals’. Tend and robust are quite different things, and overall we question the robustness. Fig 3f shows decoding of target from opto-tagged neurons, Fig S6b from non-tagged PP neurons. Decoding of target from IT neurons (3f) is not much better than from PP neurons (S6b). The PP population holds IT neurons but also many other types of pyramidal neurons with differential projection patterns, still the decoding is not much worse than from IT neurons. Together the data indicate that encoding of target is not a general trait of (all) IT neurons, if it was the decoding should be considerable better than for the PP population (?) i.e., robust appears to be an overstatement.

Response: As suggested, we revised the text to avoid overstating IT and PT neuronal differences in encoding target signals. For example, we changed the third section title of Results, ‘IT, but not PT, neurons convey robust target-related signals’, to ‘IT neurons carry stronger

target signals than PT neurons during delay period’.

In line with this Fig 3 shows that not all IT neurons display delay-period activity that varies according to target and that there is very high temporal variability in the activity within the IT population. However, the text statements read like target-dependent activity is a general property of the IT population, which is a stretch. Further, Fig 3b-c shows significant target-signals in PT neurons, but the text reads that there is a complete absence of target selectivity in PT neurons. We advise the authors to adjust the statements to reflect the actual data as to not mislead the field - the actual data is interesting and important in itself, and overstatements hurt rather than strengthen the study.

Response: Again, we made revisions throughout the text not to overstate IT and PT neuronal differences in encoding target signals.

An observation:

If the whole 4s delay is included in Fig 3f, this data can be directly compared to the data in Fig 4b, right. Together these panels indicate that the target can be decoded equally well from PP neurons during *error* trials as from IT neurons during correct trials? The target is encoded by PP neurons (that presumably include IT neurons) but the animals still perform an error? The decoding of target in correct trials appear to be better for the mixed PP population (4b, right) than the pure IT population (3f)? What does this say about encoding of target by the IT population, and the possible functional role of this encoding, including during WM necessary for successful behavior? We don’t know the answers to these questions but perhaps they can be fruit for thought for the authors in their interpretations and statements. Also, would be lovely to have the data for IT neurons in Fig 3 plotted as Fig 4, left, for direct comparisons.

Response: The strength of target signals (% correct decoding) was similar between correct and error trials during sample phase and early delay period, indicating intact registering of sensory information during sample presentation in error trials. However, neural decoding of target decreased to chance level toward the end of the delay period in error trials (Fig. 4b), indicating failure to maintain the registered sensory information during the delay period in error trials. These results suggest that the major source of errors in our study was the failure in maintaining target signals during the delay period. We briefly mentioned this in the revised text as the following (L 213-215): “These results indicate that target signals were successfully registered during the sample phase, but poorly maintained during the delay period in error trials.” Regarding better target decoding by untagged WS neurons (Fig. 4b) than IT neurons (Fig. 3f), this is not surprising because decoding improves as a function of ensemble size (Fig. 3e, supplementary Fig. 5d, and supplementary Fig. 7a); the results are based on different ensemble sizes (untagged WS, $n = 365$; IT, $n = 26$). To avoid confusion, we adjusted the location of ‘Ensemble size: 365’ in Fig. 4b so that it is located above both the left and right panels. Finally, we’d like to refrain from showing the sliding window analysis figure for IT and PT neurons because the small sample size ($n = 26$) doesn’t allow reliable estimation of target signals in 1-s windows.

Detailed comments:

Neurons in frontal cortical areas have been shown to be tuned to multiple task variables (eg sensory and motor variables)^{4,5} and mixed selectivity is also prominent in prefrontal regions. Despite the sophistication of the task design used, eg Figure 3a only shows 4 example units(?), in correct trials specifically. It would be appropriate for the author to present/represent the full dataset in a main figure, and also population analysis. The study would be greatly strengthened

by presentation of the richness of the dataset and of how identified prefrontal subpopulations align to the different task variables (eg light, sound, licking). As mentioned, accounting for activities in error trials and in relation to delay lengths is also of importance. Presenting the whole data set would also aid the readers' interpretation of how representative the optotagged activities are. Additionally, videos of representative behavior in the DMS task would be a great addition to the manuscript, and allow the reader to qualitatively assess the task used and the robustness and details of the behavior.

Response: We agree that it would be helpful to present behavioral and neural data in a more comprehensive manner. As suggested, we now show the diversity of IT and PT neuronal responses as a separate figure (Fig. 7). The new figure shows IT and PT neuronal responses to diverse events of the task. We also show summary heat maps for target-dependent activity of all IT and PT neurons that were included in the analysis (Fig. 3b). We also show the results of errortrial analysis (Fig. 4; Fig. 5b-g; Fig.10; Supplementary Fig. 7), compare neural activity across fixed and variable delay conditions (Supplementary Fig. 5), and show delay-period activity during long (15 and 25 s) delays (Fig. 10). Finally, we uploaded a video of representative behavior (Supplementary video 1).

The new analysis presented in figure 7 is an important addition to the study. We encourage the authors to discuss the obtained results in reference to the concept of mixed selectivity (see references 4 and 5 in first round to reviews). The addition of the calcium imaging experiments to study the neuronal activity during long delays fill a gap present in the first version of the manuscript. In relation to this new data, we advise the authors to check their terminology – in most analyses in the manuscript activity of IT and PT neurons are mapped to specific task variables, task phases or behavioral variables but the terminology varies. The term encoding appears to be cherry picked for some variables/analyses, signals for others, and in this new section the term responsive is used, implying a very passive role of the neurons. This paints a skewed picture of the data and we strongly encourage the authors to adjust and streamline their terminology across the study to fairly reflect all data. Alternatively, state why some analysis reveals encoding and others eg responsiveness. What is the difference in the current study?

Response: Thanks for these positive comments. As suggested, we briefly discuss 'mixed selectivity' in the revised manuscript (L 336-338). We use 'responsiveness' to describe the results presented in Fig. 7 because most of the analyses here are based on two-way ANOVA and some neurons were significantly responsive to interaction terms. Saying 'neurons are responsive to X' would be equivalent to saying 'neurons show X-dependent firing' (see our response to the second comment above), but it would be somewhat awkward to say, for example, 'IT and PT neurons show target × phase interaction-dependent firing'. Note that we also use 'responsiveness' to describe the ANOVA results of delay-period activity (comparison between fixed- and variable-delay conditions; Supplementary Fig. 5). That some neurons are significantly responsive to an interaction term means that different variables (e.g., task phase and target) are conjunctively encoded. We indicate this in the revised text as the following (L 290-293): 'The fraction of neurons significantly responsive to target × phase interaction (two-way mixed ANOVA, $p < 0.05$) was significantly larger for IT than PT neurons (Fisher's exact test, $p = 0.039$; Fig. 7b), indicating more conjunctive encoding of target and phase by IT than PT neurons.'

In Figure 5, the authors find with a demixed principal component analysis (dPCA) that PT neurons encode the passage of time, and display ramping activity during the delay. As the authors are aware, PT and IT subpopulations have been studied during motor planning in a

similar head-fixed paradigm (the sample epoch is different in this paradigm - the animals have to assess the position of a pole presented in front of their whiskers), in a different prefrontal subregion (ALM)³. In this study, PT neuron activity was shown to ramp at the end of the delay preceding contralateral (to the recorded hemisphere) licking movements. Motor planning has been defined as “internal processes by which the brain plans and executes volitional movements”⁶ and the ramping activity during the delay epoch before the onset of movement has been shown to represent motor-related information^{3,7}. In light of these findings, it is important that the authors assess possible PT ramping activity before contralateral licking movements, as this has bearing on their interpretations. Further, to support a claim that “PT neurons precisely encode the passage of time”, experiments showing how the ramping activity reflects temporal variations (eg different delay durations) are warranted. These are only suggestive starting points for the interpretation that PT neurons convey time-related information.

Response: As the reviewer pointed out, timing-related PT neuronal activity might be related to encoding the passage of time or, alternatively, motor planning. However, as we mentioned in our response to comment #3 of reviewer #1, we found a largely similar pattern of time encoding by PT neurons under the variable (1-7 s) delay condition (Supplementary Fig. 5g and i). This cannot be explained by motor planning because the time of motor response could not be predicted in the variable delay condition. Note that we did find potential motor planning-related PT neuronal activity during the last ~1 s during the delay period (Fig. 3g). However, PT neuronal population did not show biased firing toward contralateral licking responses (Fig. 7k), indicating that it is distinct from motor planning-related activity found in the motor cortex. More importantly, PT neuronal ramping activity spanning the delay period showed similar ramping activity profiles between ipsilateral and contralateral choice trials (‘target-independent dPC’, Fig. 6e; activities of sample PT neurons, Fig 3a and Supplementary Fig. 5a) unlike upcoming choice-related PT neuronal activity at the end of the delay period. These results consistently indicate that PT neuronal ramping activity spanning the delay period is related to the passage of time rather than motor planning. We discussed this matter in the revised text (L 326-333).

While we agree that PT neurons appear to convey more temporal information than IT neurons, we don’t think the analyses performed support the statement “the delay-period activity of PT neurons conveys **precise** information about the elapsed delay period’ (rows 279-80). To state precise encoding of elapsed time, additional analyses would be needed, including eg analysis of subjective vs actual time, etc. Again the overstatement hurt rather than strengthen the study here. There are many variables/processes that can appear to reflect time/timing but that is actually a reflection of something else and that would need to be excluded to support the authors statement. We encourage the authors to modify their statements regarding precise encoding of time and to contemplate if their data actually support anything beyond PT neurons’ delay activity *being related to the passage of time/convey more temporal information than IT neurons* (also statements given by the authors and that appears more appropriately reflect the data). The discussion eg states interval timing, which is not directly investigated in the current study. A role for PT neurons in timing is stated (eg discussion) but inactivation of these neurons does not change the timing of the behavior – this finding alone begs for caution in the interpretation of the data regarding PT neurons and encoding of time? Furthermore, that a temporal component can be decoded is not the same thing as the neurons convey precise information about the elapsed time.

Response: We agree. We made changes throughout the text according to the reviewer’s suggestion (L 25-26, 246, 258-259, 436, and 496-497). For example, we replaced the relevant section title from ‘PT neurons precisely encode the passage of time’ to ‘PT neurons carry more

temporal information than IT neurons’.

In their photoinhibition experiments, the authors find no effect on beh performance with optogenetic inhibition or PT and IT neurons with use of 4 s delay – inhibition of IT neurons deteriorated behavior with use of longer delays (15 s and 25 s). All the prior results in the study are based on a 4 s delay and eg the electrophysiological data for 15 s and 25 s delay appears to be missing from the manuscript. It is thus unclear if the findings obtained by use of a 4 s delay hold true for longer delays, and overall, unified findings based on ALL the generated data are warranted. In my view, refraining from showing (and/or analyzing) the whole dataset severely hampers the quality and interpretation of the study. The authors also need to reconcile their findings with findings in two recently published studies on WM in frontal regions in mice^{8,9}.

Response: Reviewer #1 also raise the same concern. We initially examined neural activity using only 4 s (fixed condition) and 1-7 s (variable condition) delays. We performed a new experiment (recording IT neuronal activity with a long delay) to address this concern and show the results in Fig. 10. Please see our response to comment #4 of the reviewer #1. We also discussed our results in relation of the findings of Liu 2014 Science and Zhu 2020 Neuron (L 463-467).

Again, this new experiment strengthens the study.

Response: Thanks for this positive comment.

The absence of parvalbumin expression in Rxfp3-Cre and Efr3a-Cre positive neurons and the physiological identification of putative fast-spiking (FS) opto-tagged units in both Rxfp3-Cre and Efr3a-Cre is a contradictory finding requiring clarification. The legend of Figure 2d indicates that immunostainings were performed in two animals (1 per genotype). These histological experiments should be repeated in more animals with statistically sound methods to ensure their robustness. In the case of a lack between Cre and PV positive neuronal populations, the authors need to clarify if the presence of FS opto-tagged units could be related to indirect recruitment of FS neurons upon light stimulation. Details of the light-response latencies for the 4 neuronal categories (PP-IT, PP-PT, FS-IT and FS-PT) as well as cross-correlograms between all optotagged units would help to elucidate if FS units have higher latencies than PP units and if PP and FS can have monosynaptic functional connections explaining indirect recruitment of FS units (PP- >FS pairs with a positive peak lag). I strongly recommend that the authors carry out these additional experiments/analysis.

Response: We agree that the number of animals for immunostaining shown in Fig. 2 is rather small. We therefore collected data from additional animals (now total n = 5 animals for each genotype; Fig. 2f, g). We also performed further analysis to probe potential indirect activation of FS IT and FS PT neurons by light stimulation. Results from the analysis of spike waveform correlation and spike latency do not support indirect activation of FS IT and FS PT neurons (Fig. 2e). Results from the cross-correlation analysis do not support indirect activation, either, albeit the number of simultaneously tagged neurons is too small to draw a definitive conclusion (n = 2 pairs in Rxfp3-Cre and n = 3 pairs in Efr3a-Cre mice; see the figure below).

Cross-correlation between simultaneously recorded optically-tagged PP and optically-tagged FS neurons in Rxfp3-Cre (a) and Efr3a-Cre (b) mice. Red dotted line, 95% percentile of cross-correlation calculated using jittered spikes.

We appreciate the further analysis, which strengthens the conclusion that the neurons denoted FS are not FS PV-expressing inhibitory interneurons. When considering this added data, we started to question the fast-spiking properties of the population denoted FS. When looking at Fig 2c we therefore wonder about the authors' classification of the neurons (we are aware of what is stated in the Methods section). Many FS neurons in Fig 2c appear to have down to 0 Hz in mean FR, or similar FR as many PP neurons (it is the mean FR that is plotted in Fig 2c?). Furthermore, there are putative PP neurons with a very narrow half-valley width in the plot. These two observations questions the classification.

We advise the authors to clarify how their classification supports identification and denoting of fast-spiking neurons as Fig 2c questions the classification. If FR is not a criterion in the classification (it says that it is in the Methods, but from 2c we don't understand this), the neurons can only be classified as eg narrow-spiking and wide-spiking (ie based on waveform). The authors need to make sure that their classification is of best scientific practice, and show the data for all parameters used for the classification (including the peak to valley ratio, particularly for classification of narrow-spiking neurons; using the peak to trough or the peak to baseline rather than the half-valley width will help to clarify the classification), preferentially in integration in a figure panel (3 axes plot?). More info in the methods would also be helpful. As the ms contains a section on the FS neurons, and conclusions are drawn, it is imperative that the classification is as accurate as possible. (Perhaps of interest: subpopulations of eg somatostatin interneurons in the mPFC are narrow-spiking).

Response: As explained in Methods, our classification of units was based on a Gaussian mixture model using three parameters (mean firing rate, peak-valley ratio, and half-valley width). We have tested various combinations of physiological parameters, and found that the current combination yields the best separation of mPFC units into two clusters. As an illustration, the figure below shows that using peak-to-trough duration (middle) or peak-to-baseline duration (right) instead of half-valley width (left) yields poorer separation of clusters. This is why the current classification scheme has been used in our previous studies (Kim 2016 Neuron; Jeong 2020 Prog Neurobiol) and presumably in another group's study as well (e.g., Kim 2016 Cell). On average, the cluster corresponding to putative inhibitory interneurons contains neurons with higher firing rates and narrower spike waveforms than the units in the other cluster. Previous studies called the units in this cluster as 'fast-spiking' (Jeong 2020 Prog Neurobiol) as well as 'narrow-spiking' neurons (Kim 2016 Neuron; Kim 2016 Cell). Using consistent terminology across studies would be helpful for readers, and we agree with the reviewer that 'fast-spiking' might be misleading because many neurons in this cluster fire at quite low rates. For these reasons, we replaced 'fast-spiking (FS)' with 'narrow-spiking (NS)' and 'putative pyramidal (PP)' with 'wide-spiking (WS)' in the revised manuscript while keeping the original classification

scheme. We also revised the corresponding text in Results to explain briefly why we call the two clusters as narrow-spiking and wide-spiking (L 120-124), and added more information about unit classification using a Gaussian mixture model in Methods (L 712-718).

In the peak procedure behavioral task, the authors conclude that “This indicates that neither IT nor PT neurons are indispensable for controlling the timing of the licking response in the present peak procedure” (p11). This statement appears to contradict the previously claimed result: “PT neurons precisely encode the passage of time”. Do the authors mean that PT neurons encode the passage of time in one behavioral paradigm but not in another, or that prefrontal PT neurons encode the passage of time but this information is not used for timing in behavior (eg licking behavior)?

Response: We found that FS IT neurons convey not only target-related signals, but also strong temporal information during the delay period. Hence, it is not surprising that inactivation of PT neurons does not impair timing-related behavior. We revised the text to explain this more clearly (L 511-515). Perhaps inactivating both PT (PP PT and FS PT) and FS IT neurons would be necessary to impair timing-related behavior, but, unfortunately, it is currently not feasible to selectively inactivate FS IT neurons without inactivating PP IT neurons.

I, unfortunately, find the spike-LFP part weak, and question how this part supports the findings in the study. Important findings on spike-LFP theta synchrony in HPC-PFC interactions in spatial WM have been published, and while the authors refer to this work, their own probing of LFP activities in WM is shallow. The mPFC theta component during WM in the task used is not really investigated, but rather the analysis is solely a comparison of spike-theta coherence between IT and PT neurons specifically during correct trials with 4 s delays. I do not find that the quite limited analysis supports the suggestion that “IT neurons may contribute to the maintenance of working memory”. Further, I have problems envisioning the mechanism the authors imply in the last part of the sentence: “...IT neurons may contribute to the maintenance of working memory, not only via sample-dependent firing, but also by enhancing synchronous firing at theta frequencies”. Whose synchronous firing are the IT neurons enhancing here? I apologize if I am misunderstanding what is meant here. Fig 2b also implies that IT neurons are more strongly coupled to a broad LFP spectra than PT neurons (albeit not significantly), which is an interesting finding not pursued. Of note, most of the previous work investigated spatial WM; the current study did not; despite this the authors interpret their results based on findings in spatial WM task and refrain from discussing the validity of doing so, which I find troublesome.

Response: Thank you for pointing this out. We revised this part considering the reviewer's constructive comments. We now show LFP power spectrograms and spike-field coherograms of untagged PP neurons in correct and error trials. The results show that theta band spike-field

coherence during the delay period is stronger in correct than error trials even though theta LFP power during the delay period is similar between correct and error trials (Fig. 5b-e). In addition, we replaced "... but also by enhancing synchronous firing at theta frequencies" with "... but also by synchronous discharges at theta frequencies" (L 239-241). We meant that synchronous discharges among different IT neurons at theta frequencies may contribute to working memory. We'd rather refrain from pursuing the difference in spike-field coherence between IT and PT neurons at other frequencies than theta because the difference is small and we have limited statistical power to explore such small differences due to small numbers of IT and PT neurons (25 and 31, respectively, with mean firing rates ≥ 0.5 Hz with $<1\%$ of LFP signals reaching the ceiling). Finally, we revised the discussion to acknowledge that previous studies investigated spatial working memory unlike our study (L 472-474).

Specific points: The study needs clarification throughout the text and figures regarding what neurons are included in different analyses/panels (and why), not just a statement of the n of neurons. It is hard to grasp what data regards the same neurons, and not, across the study.

Response: We apologize for not clearly explaining these in the initial manuscript. In the revised manuscript, we applied the following criteria to select neurons. In all analyses, we used those units with the minimum firing rate of 0.5 Hz and recorded in the sessions with at least 10 correct trials for both targets. Additionally, for the analysis of error (or miss) trials, we used those units recorded in the sessions with at least 2 error (or miss) trials for both targets. For the analysis of spike-field coupling, those neurons recorded in the sessions with more than 1% of LFP signals reaching the ceiling were excluded. For the analysis of variable delay trials, neurons recorded in the sessions with at least 10 correct trials with the length of delay ≥ 4 s were used. These are now clearly explained in the revised text (L 146-151 and throughout the text) as well as summarized in Supplementary Table 1.

Supplementary Table 1 is an important and appreciated addition to the manuscript.

Response: Thanks for this positive comment,

I strongly recommend stating in the (all) figure panels what data is presented, eg what colors, line weight etc represents. This would greatly improve the accessibility of the data.

Response: We revised the figures and legends to improve clarity.

We appreciate the improvements but find need for further information in the figures. Eg Fig 1c has no scale bar, and 1d one needs to dig in the legend to find what purple and green represents, respectively. Having the neuron numbers in the figure panels plotting population data would be very helpful, including eg Fig 2c.

Response: Thanks for pointing these out. We added a scale bar in Fig. 1c, 2a and 10a, made the legend for purple and green trials more visible in Fig. 1d, and added neuron numbers in Fig. 2c and supplementary Fig. 4.

I find the use of the term 'sample identity' being questionable for the task design used. The spatial location of the sample lick spout appears to be what is to be remembered (?)

Response: 'Sample identity' meant to denote a set of trial-unique sensory/motor signals the animal could rely on to perform the task. To avoid confusion, we replaced 'sample identity' with

'target' throughout the text.

Please see comment about target and sample above.

Response: Please see our response above.

I find the use of refs 40-42 as support for the statement "...PT neurons have been proposed to play a more important role in the maintenance of working memory than IT neurons40-42" questionable.

Response: These papers explicitly proposed an important role of PT neurons in supporting working memory (an erroneous citation in Ref 44 has been corrected; Before correction, Dembrow, 2014, Front Neural Circuits; after correction, Dembrow, 2010, J Neurosci).

This we agree with, and that was why we gave this comment – the papers do not state that "*PT neurons (have been proposed to) play a more important role in the maintenance of working memory than IT neurons*". This statement reflects an interpretation and stretch made by the authors, used to support their findings.

Response: We now understand why the reviewer made this comment. That said, we'd like to keep the sentence as is, because it wouldn't be a strong overstatement and explaining what are said precisely in the literature would require explaining nonessential details.

Ref 43: "CPn subgroups could amplify their spiking activities using the temporal properties of facilitation, higher reciprocal connectivity, and their intrinsic firing properties. Persistent firing during working memory in the frontal cortex is thought to underlie working memory (Wang, 1999). The synaptic properties and interconnectedness of CPn pairs described here suggest these neurons may provide a suitable substrate for generating persistent depolarization by recurrent excitation." (p10 L390; left column; CPn stands for 'cortico-pontine' corresponding to PT type neurons)

Ref 44: "The ability of cholinergic modulation to make neurons fire persistently was also dependent on their projection target. CPn neurons were more likely to fire persistently than COM neurons... cholinergic modulation makes them able to fire persistently beyond their stimulus input, making them well poised to contribute to mnemonic persistent activity that occurs during the delay period of working memory-like tasks." (p16 L935; left column; CPn and COM stand for 'cortico-pontine' and 'commissural' corresponding to PT and IT type neurons, respectively)

Ref 45: "Pronounced synaptic augmentation and PTP (post-tetanic potentiation) are also common phenomena found between cPCs in the mPFC.... Facilitation lasting hundreds of milliseconds, synaptic augmentation lasting up to 10 s, and PTP lasting up to minutes may be important to sustain the activity during short-term memories like those involved in a working memory task" (p5 L40; right column; cPCs stands for 'complex pyramidal cells' indicating PT type neurons)

In line with the 2 last comments, I recommend the authors to revise the ms to optimally reflect published findings and concepts.

Response: Please see our responses to the above two comments.

While the manuscript is well written, some statements are hard to grasp eg (p2) “In the primary motor cortex, IT neurons represent the deviation between current and subsequent behaviors, while PT neurons specifically encode the direction and speed of a subsequent behavioral response”.

Response: We revised the sentence so that it is clearer as the following (L 42-45):

“In the primary motor cortex, PT neurons specifically encode the direction and speed of a subsequent behavior during sensory-guided navigation. In contrast, layer 2/3 neurons are activated by unexpected direction changes induced by visual perturbations independent of behavioral response.”

Most of the methodological section provide sufficient detail. However, details regarding the training of the mice (number of days, evolution of the performance, representative video recordings of the task etc) are needed to support any concepts and conclusions.

Response: Done as suggested (L 580 - 603).

Summary

The authors hold relevant and significant datasets, with potential novelty. However, both the behavioral and electrophysiological data need a more complete analysis and presentation. I recommend the authors to base their interpretations and claims on the whole dataset, and to potentially reevaluate their concept of WM and the role of mPFC IT and PT neurons in the light of their extended findings. I am convinced the datasets hold important information on prefrontal processing in goal-oriented behavior, and particularly the involvement of PT and IT neurons, respectively, but I find that the authors need to let the (whole) data guide what conclusions can be drawn.

Response: Again, thanks for the detailed and constructive comments. As suggested, we improved our manuscript by adding additional experimental results, presenting more details of behavioral and physiological data, and revising the text to avoid confusion with respect to the concept of working memory.

We are very happy to see that the authors found our comments constructive – the revised manuscript gives support for our initial impression that the study holds important and relevant data on prefrontal processing and functioning.

Response: Again, we very much appreciate the constructive comments.

References

1. Gerfen, C. R., Paletzki, R. & Heintz, N. GENSAT BAC Cre-Recombinase Driver Lines to Study the Functional Organization of Cerebral Cortical and Basal Ganglia Circuits. *Neuron* 80, 1368– 1383 (2013).
2. Park, J. C., Bae, J. W., Kim, J. & Jung, M. W. Dynamically changing neuronal activity supporting working memory for predictable and unpredictable durations. *Sci. Rep.* 9, 15512 (2019).
3. Li, N., Chen, T.-W., Guo, Z. V., Gerfen, C. R. & Svoboda, K. A motor cortex circuit for motor planning and movement. *Nature* 519, 51–56 (2015).
4. Chen, T.-W., Li, N., Daie, K. & Svoboda, K. A Map of Anticipatory Activity in Mouse Motor Cortex. *Neuron* 94, 866-879.e4 (2017).

5. Pinto, L. & Dan, Y. *Cell-Type-Specific Activity in Prefrontal Cortex during Goal-Directed Behavior*. *Neuron* 87, 437–450 (2015).
6. Svoboda, K. & Li, N. *Neural mechanisms of movement planning: motor cortex and beyond*. *Curr. Opin. Neurobiol.* 49, 33–41 (2018).
7. Inagaki, H. K., Fontolan, L., Romani, S. & Svoboda, K. *Discrete attractor dynamics underlies persistent activity in the frontal cortex*. *Nature* 566, 212 (2019).
8. Liu, D. et al. *Medial prefrontal activity during delay period contributes to learning of a working memory task*. *Science* 346, 458–463 (2014).
9. Zhu, J. et al. *Transient Delay-Period Activity of Agranular Insular Cortex Controls Working Memory Maintenance in Learning Novel Tasks*. *Neuron* 105, 934-946.e5 (2020).

Response: Thank you much for kindly providing these references.

Various comments:

Row 173-74: 'This cannot be attributed to the mPFC neurons of Rxfp3-Cre mice carrying stronger target-related signals than those of Efr3a-Cre mice (Supplementary Fig. 6)'. This statement is quite confusing given what is stated before in this section. Do the authors mean that recorded PP neurons in Rxfp3-Cre mice do not only include IT neurons, and as a *population* PP neurons in Rxfp3-Cre mice do not carry 'stronger' target-related signals than the PP *population* in Efr3a-Cre mice? (And what is meant with stronger here?). This text would benefit from modification.

Response: We changed it to 'This cannot be attributed to the difference between Rxfp3-Cre and Efr3a-Cre mice because target decoding using untagged WS neurons was similar between the two animal groups' (L 176-178) to improve clarity.

It would be helpful for the readers if the authors in the section 'Delay-period activity differs between correct and error trials' (and anywhere else appropriate) point out that the PP neurons population represents random sampling of putative pyramidal neurons in the two mouse lines (i.e., is different from IT or PT populations). This to eliminate confusion and misunderstanding.

Response: As suggested, we added '(i.e., randomly sampled putative pyramidal neurons)' to indicate this clearly (L 204).

While we very much appreciate Fig S4, the present format prohibits inspection of the data – the plotting is just too small.

Response: We increased the spatial resolution of Fig. S4 to allow inspection of individual unit data upon zooming in (when using a digital file).

Please consistently state what parameters (neurons, sessions, trials etc) are indicated with given numbers (n), see eg Fig S7 figure legend. Alternatively, give the n directly after the relevant parameter instead of eg in the end of the sentence to avoid confusion, see e.g., row 260-61: '....using the untagged PP neurons recorded in the sessions with ≥ 2 error trials for both targets (n = 365). Much time is spent figuring out the n of parameters, and what parameters stated ns refer to.

Response: Done as suggested (L 263; L 378-379; Fig S7).

Row 167: “Also, decoding of correct target using delay-period neuronal ensemble activity improved as a function of ensemble size for the IT neurons, but remained similar to the level of chance (50%) across all tested PT neuron ensemble sizes (Fig. 3e).” Is the target, or correct reporting of target, decoded here (were not only correct trials included?)

Response: Right, only correct trials were used here and hence ‘decoding of correct target’ would be confusing. It’s been replaced with ‘decoding of target’.

Row 295; would be helpful if ‘trial outcome-dependent firing’ was defined; what is analyzed and during what phase.

Response: To indicate clearly trial outcome refers to correct versus error, we replaced ‘We then examined trial outcome-dependent firing of IT and PT neurons during the choice phase...’ with ‘We then compared IT and PT neuronal activity between correct and error trials during the choice phase (first 1 s since the first lick after delay offset) to examine trial outcome-dependent firing...’ (L 297-301).

Would be helpful if the peak procedure was shortly explained, what is specifically investigated and how.

Response: Done as suggested (L 362-365).

Abstract (row 26) and Discussion (row 430): “PT neurons convey far more temporal information” “Far” a quite unscientific and imprecise.

Response: ‘far more temporal information’ has been replaced with ‘more temporal information’.

Row 163: the “absolute target selectivity index” appears to not be included in the method section. We apologize if we are mistaken.

Response: We now mention absolute target selectivity index in Methods (L 829-830).

We would suggest to move the new section “IT and PT neurons are responsive to diverse task events” at an earlier position in the manuscript (before “IT, but not PT, neurons convey robust target-related signals” maybe) as this order would be more natural to the reader.

Response: Thanks for the suggestion, but we’d like to keep the current arrangement to focus more on delay-period activity of IT and PT neurons. The current flow wouldn’t be unnatural to the reader given the following statement in the introduction: ‘We are particularly interested in whether and how the different types of cortical projection neurons contribute to the maintenance of task-relevant information in the absence of external sensory cues.’

REVIEWERS' COMMENTS

Reviewer #3 (Remarks to the Author):

We have been through the revised manuscript and think that there now is a good balance between the data and its' description, and nice and helpful stringency in the terminology and concept.

We agree with the authors that a NS and WS classification for this dataset is better than the former classification.

"Finally, we'd like to refrain from showing the sliding window analysis figure for IT and PT neurons because the small sample size ($n = 26$) doesn't allow reliable estimation of target signals in 1-s windows." This comment raises some questions regarding the target encoding of IT neurons but we feel that we at this point must trust the authors' interpretation of their data.

We thank the authors for their collaborative spirit during the revisions and congratulate them on their interesting findings.